



Detecting the permafrost carbon feedback: Talik formation and increased cold-season
respiration as precursors to sink-to-source transitions
Nicholas C Parazoo[1], Nicholas.c.parazoo@jpl.nasa.gov
Charles D. Koven[2], cdkoven@lbl.gov
David M. Lawrence[3], dlawren@ucar.edu
Vladimir Romanovsky[4], veromanovsky@alaska.edu
Charles E. Miller[1], Charles.E.Miller@jpl.nasa.gov
[1]Jet Propulsion Laboratory, California Institute of Technology, Pasadena, California, 91109, USA
[2]Lawrence Berkeley National Laboratory, Berkeley, California, USA
[3]National Center for Atmospheric Research, Boulder, Colorado, USA
[4]Geophysical Institute UAF, Fairbanks, Alaska, 99775, USA



**Abstract**

Thaw and release of permafrost carbon (C) due to climate change is likely to offset increased vegetation C uptake in Northern High Latitude (NHL) terrestrial ecosystems. Models project that this permafrost C feedback may act as a slow leak, in which case detection and attribution of the feedback may be difficult. The formation of talik, a sub-surface layer of perennially thawed soil, can accelerate permafrost degradation and soil respiration, ultimately shifting the C balance of permafrost affected ecosystems from long-term C sinks to long-term C sources. It is imperative to understand and characterize mechanistic links between talik, permafrost thaw, and respiration of deep soil C to detect and quantify the permafrost C feedback. Here, we use the Community Land Model (CLM) version 4.5, a permafrost and biogeochemistry model, in comparison to long term deep borehole data along North American and Siberian transects, to investigate thaw driven C sources in NHL (> 55°N) from 2000-2300. Widespread talik at depth IS projected across most of the NHL permafrost region (14 million km$^2$) by 2300, correlated to increased cold season warming, earlier spring thaw, and growing active layers. Talik formation peaks in the 2050s in warm permafrost regions in the sub-Arctic. Comparison to borehole data suggests talik formation may even occur sooner. Accelerated decomposition of deep soil C following talik onset shifts the surface balance of photosynthetic uptake and litter respiration into long-term C sources across 3.2 million km$^2$ of permafrost. Talik driven sources occur predominantly in warm permafrost, but sink-to-source transition dates are delayed by decades to centuries due to high ecosystem productivity. In contrast, most of the cold permafrost region in the northern Arctic (3 million km$^2$) shifts to a net source by the end of the 21$^{st}$ century in the absence of talik due to the high decomposition rates of shallow, young C in organic rich soils coupled with low productivity. Our results provide important clues signaling imminent talik onset and C source transition including: (1) late cold season (Jan-Feb) soil warming at depth (~2 m), (2) increasing cold season emissions (Nov-Apr), (3) enhanced respiration of deep, old C in warm permafrost and young, shallow C in organic rich cold permafrost soils. Our results suggest a mosaic of processes that govern carbon source-to-sink transitions at high latitudes, and emphasize the urgency of monitoring soil thermal profiles, organic C age and content, cold season $CO_2$ emissions, and atmospheric $14CO_2$ as key indicators of the permafrost C feedback.





## 1. Introduction


The future trajectory of the Arctic Boreal Zone (ABZ) as a carbon (C) sink or source is of global
importance due to vast quantities of C in permafrost and frozen soils (Belshe et al. 2013). Cold
and waterlogged conditions in the ABZ have hindered soil organic material (SOM) from microbial
decomposition and led to long-term C accumulation at soil depths below 1 m (Ping et al. 2015).
Arctic warming, which stimulates plant growth as well as respiration in tundra ecosystems
(Euskirchen et al. 2012; Natali et al. 2012; Mack et al. 2004; Barichivich et al. 2013; Commane et
al. 2017), has driven a period of C cycle intensification over the last 50 years with greater C inputs
and outputs across high latitude ecosystems (Graven et al. 2013). Expert assessments of site-level
observations, inversion studies, and process models suggest that Arctic C balance is near neutral,
but large uncertainties allow for solutions ranging from small sources to moderate sinks;
however, most assessments favor an overall strengthening of the regional C sink, with
productivity gains exceeding respiration losses on average (McGuire et al. 2012).
The effect of continued warming on future northern high latitude (NHL) ecosystem C balance is
uncertain but appears to be increasingly dependent on responses to changes in cold season
emissions, soil moisture, shifts in vegetation community, and permafrost degradation (Abbott et
al. 2016). These vulnerabilities are likely driven by disproportionate warming during the cold
season (Fraser et al. 2014), which is projected to increase at twice the rate of summer warming
over the next century (Christensen et al. 2013). For example, winter warming during the long cold
season promotes increased soil respiration, offsetting C uptake during the short Arctic growing
season (Oechel et al. 2014; Euskirchen et al. 2016; Commane et al. 2017), and shifting tundra
ecosystems from C sink to source (Webb et al. 2016). Winter warming also promotes earlier and
more rapid snow melt and landscape thawing (Goulden 1998; Schuur et al. 2015). This can impact
seasonal C balance through increased hydrological export of SOM by Arctic rivers (Olefeldt &
Roulet 2014), which is projected to increase by 75% by end of century (Abbott et al. 2016). Early
snow melt can also cause increased exposure of the land surface to solar absorption (Lawrence
et al. 2012) resulting in increased evapotranspiration and summer drought risk (Zhang et al.
2011), which decreases terrestrial biomass through reduced plant growth and increased intensity





and frequency of boreal fire emissions and fire disturbance (Yi et al. 2014; Veravebeke et al.,
2017). ABZ fire-driven C losses are expected to increase four-fold by 2100 (Abbott et al. 2016).
On longer time scales, permafrost degradation and resulting C losses from deep, old C is expected
to be the dominant factor affecting future Arctic C balance (McGuire et al. 2012; Lawrence et al.
2015; Schuur et al. 2015). In addition to these effects, warmer temperatures and longer non-
frozen (NF) seasons caused by earlier spring thaw and later autumn freezing can promote
accelerated deepening and increased duration of the active layer (layer of soil near the surface
which is unfrozen in summer and frozen in winter) and thawing permafrost. This can initiate
formation of a talik zone (perennially thawed sub-surface soils) during active layer adjustment to
new thermal regimes (Jorgenson et al. 2010). Talik as well as longer, deeper active layer thaw
stimulate respiration of soil C (Romanovsky & Osterkamp 2000; Lawrence et al. 2008), making
the ~1000 Pg C in near surface permafrost vulnerable to decomposition (Tamocai et al. 2009;
Harden et al. 2012).
Climate models used in the Coupled Model Intercomparison Project Phase 5 (CMIP5) consistently
project widespread loss of permafrost in the future due to climate warming [Slater and Lawrence,
2013], though the ESMs that participated in the CMIP5 also project NHL terrestrial C uptake
rather than losses due to warming (Ciais et al., 2013). This projection conflicts with expectations
from field studies (Schuur et al., 2009; Natali et al., 2014), but newer approaches, such as
explicitly representing the vertical structure of soil respiration and its coupling to deep soil
thermal changes, lead to changes in the model-projected response from a net C gain with
warming to a net loss, and hence a positive carbon-climate feedback (Koven et al. 2011).
Permafrost C emissions are likely to occur gradually over decades to centuries, and therefore are
unlikely to cause abrupt and easily detected signals in the global C cycle or climate (Schuur et al.
2015). We use the coupled permafrost and biogeochemistry Community Land Model Version 4.5
(CLM4.5) to investigate in detail the subsurface thermal processes driving C emissions from
shallow (0-3m) and deep (>3m) permafrost C stocks and to project the rate of NHL permafrost C
feedbacks ( > 55°N) over the 21$^{st}$ century. Using CLM4.5 in the framework of an observing system
simulation experiment (e.g., Parazoo et al., 2016), we ask how we might be able to (1) identify
potential thresholds in soil thaw, (2) detect the specific changes in soil thermal regimes that lead



to changes in ecosystem C balance, and (3) project future C sources following talik onset. We
hypothesize that talik formation in permafrost triggers accelerated respiration of deep soil C and,
ultimately, NHL ecosystem transition to long-term C sources.
Comparison to observed thaw at selected tundra and forested ecosystems along north-south
transects in Siberia and North America in the 20[th] and early 21[st] century provides a reference to
evaluate historical thaw patterns and projected thaw rates. The remainder of our paper is
organized as follows: Section 2.1 describes our methods to simulate and analyze soil thaw and C
balance in CLM4.5; Section 2.2 describes borehole datasets used to analyze CLM4.5 soil thermal
regime; Section 3.1 presents results of talik formation in CLM4.5 and comparison of simulated
thaw profiles to borehole data in North America; Section 3.2 evaluates projected thaw rates
against long-term borehole data in Siberia; Section 3.3 identifies timing and location of C source
onset and discusses formation mechanisms in the presence and absence of talik; Section 3.4
presents a projection of future C sources at talik locations; Sections 4 discusses the main findings.

**2. Methods**
*2.1 Simulations*
CLM4.5 provides an accurate characterization of the physical and hydrological state of
permafrost needed to evaluate permafrost vulnerability and identify key processes (Swenson et
al. 2012; Lawrence et al. 2008). CLM4.5 includes a basic set of permafrost processes to allow
projection of permafrost carbon–climate feedbacks, including snow schemes, vertically resolved
SOM dynamics and soil hydrology, coupled hydraulic and thermal properties in frozen and
unfrozen soils allowing realistic seasonal evolution of the active layer, and interaction with
shallow (0-3m) and deep (>3m) permafrost C (Swenson et al. 2012; Oleson et al. 2013; Koven et
al. 2013, 2015; Lawrence et al. 2008). The soil grid includes 30 vertical levels that has a high-
resolution exponential grid in the interval 0–0.5 m and fixed 20-cm layer thickness in the range
of 0.5–3.5 m to maintain resolution through the base of the active layer and upper permafrost,
and reverts to exponentially increasing layer thickness in the range 3.5–45 m to allow for large
thermal inertia at depth.



We use CLM4.5 configured as described in two recent permafrost studies (Lawrence et al. 2015;
Koven et al. 2015) using time-varying meteorology, N deposition, $CO_2$ concentration, and land
use change to capture physiological (i.e., $CO_2$ fertilization) and climate effects of increasing $CO_2$
over the period 2006-2300. We use an anomaly forcing method to repeatedly force CLM4.5
with observed meteorological from the CRUNCEP dataset for the period 1996–2005 (data
available at dods.ipsl.jussieu.fr/igcmg/IGCM/BC/OOL/OL/CRU-NCEP/) and monthly anomalies
added based on a single ensemble member from a CCSM4 Representative Concentration
Pathway 8.5 (RCP8.5) simulation. The period from 1996 to 2015 represents a base
climatological period used for calculating monthly anomalies, with a 20-y record chosen to
minimize large anomalies in the first few years. This process is repeated for all variables and all
times from 2006 to 2300 (constantly cycling through the same 1996–2005 observed data).
We caution that we are using only a single ensemble member from CCSM4, and hence our
results represent one realization from one model forced with one climate scenario.  This results
in uncertainties from the historical climate/weather forcing, the structure and parameterization
of the model, and climate scenarios (both across models and across emissions scenarios).
Simulations are carried out on a global domain at a grid resolution of 1.25° longitude x 0.9375°
latitude and saved as monthly averages. Simulation output is collected into decadal averages
from 2011-2300 (e.g., 2011-2020 averages for the 2010s, 2021-2030 for the 2020s, etc). Our
method to link C balance changes to permafrost thermal state relies on identifying the timing of
two key processes: (1) talik formation, and (2) C source transition. Talik formation represents a
critical threshold of permafrost thaw. The C source transition represents a shift of ecosystem C
balance from a neutral or weak C sink to a long-term source driven by onset of permafrost thaw
and respiration of deep SOM (Koven et al. 2015). Using the hypothesis that talik formation
triggers a transition to long-term C sources, we quantify the extent of talik formation and rate
of transition to C source once talik has formed in permafrost-affected NHL ecosystems.
Following Koven et al. (2015), we define the timing of C source transition from net annual sink
to net source as the first decade when annual net biome production (NBP) decreases below -25
g C m$^{-2}$ y$^{-1}$ and remains a source (NBP < 0 g C m$^{-2}$ y$^{-1}$) through 2300. Here, we use the sign
convention of NBP < 0 to represent net C flux from land to atmosphere (e.g., source). The





timing of talik formation is defined as the first decade when soil temperature ($T_s$) for any layer
between 0 and 40 m exceeds -0.5°C for all months in a calendar year (Jan-Dec), assuming that
soils start off as permafrost at the beginning of our simulations in 2006. We use a negative
freezing point threshold to account for availability of liquid water below 0°C due to freezing
point depression. We note the real threshold temperature at which liquid water remains
available varies depending on the soil salinity or mineral content, the latter effect of which is
included in the actual respiration calculations used by CLM. Here we use -0.5°C as the
freeze/thaw cutoff, and examine cutoffs at 0.5°C increments from 0°C to -2.0°C.
We introduce the thawed volume-time integral, or "thaw volume", as a metric to better
understand thaw dynamics and help identify thaw instability thresholds. We integrate
permafrost in both time (month of year) and depth (soil layer from the surface to 40 m) into a
logical function that is one for thawed layers ($T_s$ > -0.5°C), zero for frozen layers, and multiply
each thawed layer by layer thickness to convert to units of meter months. This conversion
accounts for non-uniform layer thicknesses, providing a consistent metric for comparing
simulated and observed thaw.
Our analysis focuses on NHL grid points within the ABZ north of 55°N. We analyze talik
formation and C source transitions in the context of the simulated initial state of SOM, and
published maps of permafrost conditions from NSIDC
(https://nsidc.org/data/docs/fgdc/ggd318_map_circumarctic/) and described in Brown et al.
(2001). Permafrost extent is classified as continuous (90-100%), discontinuous (50-90%),
sporadic (10-50%).
*2.2 Observations*
We compare simulated patterns of active layer dynamics and soil thaw to patterns observed
from contemporary and historical borehole measurements of permafrost temperature profiles.
We focus on sites in western North America and eastern Siberia with daily continuous
observations year-round (Jan-Dec) over multiple consecutive years. The primary focus of data in
North America (2004-2013) is to evaluate seasonal progression of soil thaw and talik formation
near the surface (0-3 m). Siberian data, which have a longer record on average (1950-1994), are



used to evaluate long term trends in soil thaw at 0.0 - 3.6 m depth. Site locations are shown in
Fig. 1.
Siberian data are based on measurements along the East Siberian Transect (EST)
(https://arcticdata.io/metacat/metacat/doi:10.5065/D6Z036BQ/default). The EST consists of 13
sites that cover a southwest-to-northeast transect in east Siberia [60.7°N, 114.9°E to 68.3°N,
145°E] during the period 1882-1994 (Romanovsky et al. 2007). For this study, we focus on the 9
sites which report measurements as monthly averages at regular depths of 0.2, 0.4, 0.8, 1.6,
and 3.2 m. Unfortunately, data gaps of years to decades exist on a site-by-site basis, and many
years do not report the full annual cycle over multiple layers. We therefore only analyze years
with at least 10 months $yr^{-1}$ of reported monthly mean soil temperature at each layer, and 55
months across the 5 layers (out of 60 possible layer-months per year). Based on these
requirements, we find that 6 of 9 sites yield at least 6 years of data over multiple decades, and
are well suited for examining historical thaw trends. For comparison to projected trends in
CLM4.5, we recalculate observed trends using the inter-site average from all 9 sites at 3 unique
locations: northern Siberia (67°N, 144°E), southwest Siberia (61°N, 115°E), and southeast
Siberia (59°N, 131°E). Site information is shown in more detail in Table 1.
North American transect data are taken from the global terrestrial network for permafrost
(GTNP) borehole database (http://gtnpdatabase.org/boreholes): (1) Borehole 1108 at Mould
Bay in Canada [119°W, 76°N] from 2004-2012; (2) Borehole 33 in Barrow along the northern
coast of Alaska [156°W, 71.3°N] from 2006-2013; and (3) Borehole 848 in Gakona in southeast
Alaska [145°W, 62.39°N] from 2009-2013. Mould Bay is a continuous permafrost tundra site
with measurements at 63 depths from 0 - 3 m. Barrow is a continuous permafrost tundra site
with measurements at 35 depths from 0 - 15 m. Gakona is a continuous permafrost forest
tundra site with measurements at 36 depths from 0 - 30 m. All datasets are reported as daily
averages. For each site, we aggregate from daily to monthly averages requiring at least 20 days
$month^{-1}$ at each layer and for each year. Measurements are reported at multiple depths and
high vertical resolution (up to 0.1 m in shallow layers) but are generally non-uniform in depth
(multiple layers missing, different layers reported for each site). Given these inconsistencies



and records <= 8 years, we use these data for qualitative analysis of seasonal and vertical
patterns in permafrost thaw. Site information is shown in more detail in Table 2.
**3. Results**
*3.1 Simulated Talik Onset in the 21$^{st}$ Century*
Our simulations show widespread talik formation throughout Siberia and northern North
America over the period 2010-2300 (Fig. 1A), impacting ~14.5 million km$^2$ of land in NHL's (55°-
80°N) assuming a freeze/thaw threshold of -0.5°C. 10.6 million km$^2$ of land in Europe,
southwest Asia, and N. America (below 60°N) either formed talik prior to the start of our
simulation in 2010 in regions already experiencing degraded permafrost (e.g., Fig. 1D,
permafrost extent < 90% in southwest Siberia and southern N. America), or did not have
permafrost to begin with. A small amount of land along northern coastal regions (~1.6 million
km$^2$) show no talik formation prior to 2200.
The long-term trend and decadal variability of talik formation are quantitatively and
qualitatively similar for freeze/thaw thresholds at or below -0.5°C (Fig. 1B). Peak formation
generally occurs over the period 2050-2150, accelerating rapidly early in the 21$^{st}$ century, and
leveling off in the late 22$^{nd}$ century. The timing and location of talik formation correlates with
the annual mean temperature of permafrost at 3 m (T$_{soil-3m}$) (Fig. 1C) and observed permafrost
state (Fig. 1D, from Brown et al. 2001) at the start of our simulation; we see earlier talik
formations in sub-Arctic regions (< 66N) with warm simulated permafrost (T$_{soil-3m}$ > 0°C) and
permafrost extent less than 90%, and later formation in northern regions with cold permafrost
(T$_{soil-3m}$ < 0°C) and continuous permafrost. Talik formation progresses northward from the sub-
Arctic to the Arctic over time, starting in the warm/discontinuous permafrost zone in the 21$^{st}$
century then to the cold/continuous permafrost zone the 22$^{nd}$ century. This suggests a shift in
permafrost state across the pan-Arctic from continuous to discontinuous over the next 2
centuries.
Our simulations demonstrate consistent patterns of changing thaw volume leading up to and
following initial talik formation, independent of the decade of talik onset. Time series of thaw
volume as a function of decade relative to talik onset (Fig. 2A) show a steady rise in thaw



volume of 1-2 m months yr$^{-1}$ in the decades prior to talik formation, with thaw limited primarily
to shallow soils (< 1.5 m) and summer/early fall. Thaw volume accelerates to 10-20 m months
yr$^{-1}$ within 1-4 decades of talik onset, coinciding with thaw penetration at depth (~2 meters on
average, Fig. 2B) and deeper into the cold season (~Jan-Apr). Thaw penetration into the Jan-Apr
period occurs for the first time at 2.6 ± 0.9 decades prior to talik onset (vertical grey lines in Fig.
2A). At talik onset, thaw volume jumps from mean values of 60 ± 10.7 m months yr$^{-1}$ to 377 ±
44 m months yr$^{-1}$ at a mean depth of 4.1 meters. Thaw volume levels out within one decade
following initial talik formation and accelerated thaw of all soil layers; this leveling is an artifact
of the maximum depth of soils in CLM4.5 (equal to 45.1 meters), and represents the complete
transition from permafrost to seasonally-frozen ground in the model. The transition to deep
cold season thaw and rapidly increasing thaw volume represent key threshold signaling
imminent talik onset.
Onset of surface thaw in the uppermost soils during the spring freeze/thaw transition provides
another reliable predictor for talik onset. In particular, we find consistent dates and trends of
spring thaw in the surface soil layer in the decades leading up to talik onset (Fig. 2C), shifting by
about 1 week over 4 decades from Day of Year (DOY) 134 ± 2.8 (~mid May) to DOY 127 ± 3.5
during talik formation (~early May).
Changes in total column soil water and sub-surface drainage following talik onset may provide
clues a posteriori that talik is already present. Lawrence et al. (2015) show that deepening of
the active layer and thawing of permafrost allows water to drain deeper into the soil column,
which dries out near surface soils. Our simulations show a similar drying pattern in shallow
layers (~0-1 m depth) in the 4 decades prior to talik onset (Fig. 2D). This does not significantly
impact total water storage as it is primarily a redistribution of water within the column;
however, there are significant changes in water balance following talik onset, including rapid
increase in sub-surface drainage and decrease in volumetric soil moisture, as discussed below.
The time evolution of soil vertical thermal and hydrological structure for the subset of grid cells
that form talik in the 2090s is shown in more detail in Fig. 3. Here, we have subtracted the
thermal and hydrological profiles in the 2040s to show relative change. The 4 decades prior to
talik onset are shown in Fig. 3A-D (2050s – 2080s), the decade of talik onset in Fig. 3E (2090s),





and the 4 decades following talik onset in Fig. 3F-I (2100s – 2130s). CLM4.5 represents the
process of soil thawing as passage of a "thaw front" in space and time through soil layers,
penetrating and warming colder, deeper layers, and bringing the frozen soil environment at
depth closer to thermodynamic equilibrium with the warming atmosphere. At 4 decades prior
to talik onset (Fig. 3A), our simulated thawed layer exhibits a tilted time-depth profile with
earlier thaw and longer thaw duration (~4-5 months) in the near surface (< 1 m) compared to
later thaw and reduced thaw duration (1-2 months) at maximum thaw depth (~ 2 m). In the 3
decades leading up to talik onset, we find more pronounced tilting of the thawed layer with
time and depth, with gradual deepening to 3-4 m and penetration of thaw period into Jan-Feb.
Our simulations indicate an increased rate of heat transfer and thawing at depth following talik
onset, leading to rapid subsequent thawing, drying, and decrease in the thickness of the
seasonally frozen layer above talik (Fig. 3 E-I). This rapid thawing is depicted in Fig 2A as the
large jump in thaw volume, and in Fig. 2D as enhanced drying and drainage, with drying peaking
at 3.5-4.5 m depth. In our simulations, talik onset effectively pulls the "bath plug" that was the
ice filled pore space at depth, with year round ice-free conditions allowing soil water to
percolate and be diverted to sub-surface drainage (Lawrence et al., 2015). We note that
bedrock soil is not hydrologically active in CLM4.5, and thus the rate of thawing and drainage in
response to permafrost thaw may be underestimated in deeper CLM4.5 layers near bedrock
due to reduced heat capacity.
Our simulated pattern of phase lag for heat transfer to depth mimics observed thaw profiles in
N. America (Fig. 4), which are sensitive to latitude and ecosystem, but with more "vertical"
time-depth tilt in CLM4.5 compared to observations. Borehole data shows shallow (~0.5 m) and
seasonally short (~3-4 months from Jun-Sep) thaw at the northernmost tundra site in the
Canadian Archipelago (Fig. 4A; 76°N, Mould Bay), shallow but longer thaw (5 months from Jun-
Oct) moving slightly south to North Slope Alaska (Fig. 4B; 71.3°N, Barrow), and deep (~3 m) and
seasonally long (May-Feb) thaw at the low latitude continental boreal site in southeast Alaska
(Fig. 4C; 62.4°N, Gakona). CLM4.5 shows reduced depth and seasonal duration of thaw when
sampled at these specific geographical points, although the north-south gradient of increasing
thaw moving south is preserved (Fig. 4D-F). Given the challenging task of comparing point



locations with grid cell means, we also examine the mean behavior of CLM4.5 at locations
where soil temperature at depth is similar to that observed. Accounting for permafrost
temperature at 3 meters (by sampling all locations with $T_{soil-3m}$ within 0.5°C of the observed
temperature) better reproduces thaw depth, but with reduced seasonal duration throughout
the soil column (Fig. 4G-I). These results suggest the current ensemble CLM4.5 run
overestimates the rate of soil refreeze in early fall.
Based on the pattern of January and February thaw/freeze dynamics observed at Gakona in the
2010s and the time lag of 1-3 decades from this occurrence to talik onset in our simulations, we
project that Gakona will form talik as early as the 2020s, assuming the atmosphere continues to
warm as prescribed in CLM4.5. Talik onset in CLM4.5 is variable in this region with earliest onset
by mid-century (~2050s, Fig. 1A); however, our comparison to observations suggests that
simulated thaw rates in this region and for similar permafrost temperatures are
underestimated, and that earliest onset may occur sooner than predicted. Overall, we find that
simulated patterns of permafrost thermal state change are consistent with available
observations, but that the exact thaw rates are uncertain. We keep these uncertainties in mind
as we examine patterns of change and talik formation simulated into 2300.
*3.2 Evaluation of Simulated Thaw Rates and Talik Onset Against Siberian Borehole Data*
The Siberian borehole locations have similar permafrost extent (> 50%) to the North American
locations according to the Circumpolar Permafrost Map (Brown, 2001) and similar mean annual
air temperature (~ -13.6°C) in the 2000s according to CLM4.5. However, air temperature is
more seasonal in Siberia, including colder winters (4°C colder) and warmer summers (6°C
warmer). Spring thaw for the Siberian sites occurs two weeks earlier on average than for the
North American sites in the 2000s, but follows the same pattern of later thaw date moving
north along the borehole transect.
Next we examine thaw trends observed from borehole soil temperature data in Siberia in the
20[th] century and evaluate patterns of CLM4.5 projected trends in the 21[st] century. We note
several caveats in these comparisons: (1) model simulations are based on only one realization
(i.e., model ensemble member) of historic and future warming and projected permafrost thaw,



(2) availability and access of long term records in Siberia is limited, and (3) there is significant
variability in space and time in simulated and observed thaw rates, making direct comparisons
challenging. These comparisons thus serve primarily as a first benchmark for future model
analysis and development.
We focus first on site-specific long-term trends by analyzing the 5 Siberian borehole sites which
recorded at least 5 years of data spanning multiple decades: Drughina, Lensk, Macha, Uchur,
and Chaingda. Records at these locations show a decrease in thaw volume with an average of
0.19 m months yr$^{-1}$ from 1955 – 1990 (Table 1, Fig 5). All sites except Drughina show negative
trends, with larger trends in southern locations, ranging from 0.51 m months yr$^{-1}$ from 1957-
1990 at Chaingda in southern Siberia, to a statistically insignificant trend of -0.083 months yr$^{-1}$
from 1969-1990 at Drughina in northeastern Siberia, suggesting a more or less constant
thermal state at this site. Further examination indicates that active layer thickness at Drughina
actually decreased to 0.8 meters from 1989-1990 compared to 1.2 meters in the 1970s.
Drughina also shows smaller average thaw volume magnitude compared to other sites,
consistent with shallower thaw (layer thickness increases exponentially with depth along the
Siberian transect). Together, these findings indicate that active layer thickness is decreasing at
Drughina.
The average trend at 3 long term stations in southwest Siberia clustered at [60.7°N, 114.9°E] is -
0.18 m months yr$^{-1}$, which is a factor of 3 weaker compared to Chaingda station (thaw rate =
0.51 m months yr$^{-1}$; $p < 0.05$) slightly south and 15° to the east (59.0°N 130.6°E). Each of these
permafrost sites exhibits talik at various times between 1957 and 1990 (vertical dashed line),
with earlier talik onset to the west (1957 at Lensk, 1970 at Macha, 1974 at Uchur) and later talik
onset to the east (1989 at Chaingda). The presence of talik reflects an increase in the depth and
duration of thaw late into the cold season, rather than a physical decrease in soil thaw as
appears to be the case at Drughina. We acknowledge the difficulty in identifying talik onset due
to discontinuities in the dataset and limited vertical information; however, we note that the 15-
30 year gap between talik formation in the western site cluster vs Chaingda 15° east is
geographically consistent with model simulations of later talik formation in eastern Siberia in
the 21$^{st}$ century (Fig. 1A) and thus may represent a gradual expansion of warming into the east.





In general, permafrost appears to be degrading more rapidly at the southern locations
compared to the northern location.
We recompute observed thaw trends at regional clusters using combined records at the 2 sites
in northern Siberia, 6 sites in southwest Siberia, and 1 site in southeast Siberia (Table 1) and
compare these to thaw projections in CLM4.5 (Fig. 6). The simulated trend in thaw volume
shows a change in sign at northern locations (blue), acceleration of thaw at southwest sites
(orange), and reduction of thaw at the southeast sites (brown). Despite the change in sign in
northern Siberia representing a possible shift from net permafrost accretion to net thaw in the
mid 21$^{st}$ century, thaw projections indicate continued stability of permafrost through the early
22$^{st}$ century. Our simulations show a shift to accelerated soil thaw beginning in the early 2080s,
marked by onset of deep soil thaw late in the cold season (January, denoted by cyan circle).
Thaw projections in the southern locations show more abrupt shifts in the magnitude of thaw
volume in the 21$^{st}$ century. Projected thaw volume is generally on track with borehole data at
the western locations, but deviate strongly from the single record at Chaingda. Despite
observed talik onset as early as the 1970s in the southwest (Macha and Uchar) and late 1980s
in the southeast (Chaingda), simulated talik onset is not projected to occur until the 2030s at
the western sites (orange) and the 2080s at the eastern sites (brown). The strong discrepancy
between observed and simulated thaw and talik onset in southern Siberia warrants close
monitoring and continued investigation of this region through sustained borehole
measurements and additional model realizations of potential future warming.
*3.3 Carbon Cycle Responses to Changing Ground Thermal Regime*
Fig. 7A plots the decade in which NHL ecosystems are projected to transition to long-term C
sources over the next 3 centuries (2010-2300). A total of 6.8 million km$^2$ of land is projected to
transition, peaking in the late 21$^{st}$ century, with most regions transitioning prior to 2150 (4.8
million km$^2$ or 70%, Fig. 7B, solid black). At first, pan-Arctic C fluxes remain neutral on average
over the early- to mid- 21$^{st}$ century (2010-2070) as increasing productivity and C sinks dominate
large scale C balance (Fig. 7B, purple). The spatially integrated pan-Arctic C balance increasingly
favors net C source dominance over the next 100 years, peaking at 0.7-0.8 Pg C in the mid- to



late- 22nd century, followed by gradual decline to 0.5 Pg C by 2300. The cumulative C source
over the entire simulated period (2010-2300) is 11.6 Pg C.
Most regions (6.2 million $km^2$ of land, or 91%) identified as a C source also form talik at some
time during the simulation (Fig. 7C). However, the geographic pattern of C sink-to-source
transition date is reversed compared to that of talik formation, with earlier transitions at higher
latitudes (the processes driving these patterns are discussed in detail below). Overall, the lag
relationship between talik onset and C source transition exhibits a tri-modal distribution (Fig.
7D), with peaks at negative time lag (C source leads talik onset, Median Lag = -5 to -6 decades),
neutral time lag (C source synchronized with talik onset; Median Lag = -2 to 1 decade), and
positive time lag (C source lags talik; Median Lag = 12 decades; red shading in Fig. 7C). Roughly
half of these regions (3.2 million $km^2$) show neutral or positive time lag (lag ≥ 0). This pattern,
characteristic of the sub-Arctic (< 65°N), represents the vast majority of C source transitions
after 2150 (Fig. 7B, dotted). The remaining regions (3.0 million $km^2$) in the Arctic and high Arctic
(> 65°N) show negative time lag and account for most of late 21st century sources (Fig. 7B,
dashed). C sources in the regions not identified as talik (0.63 million $km^2$) either show talik
presence at the start of our simulation, or are projected to transition in the absence of
permafrost or in regions of severely degraded permafrost (Fig. 7C, dash dotted).
Here, we investigate biological and soil thermal processes driving these relationships, focusing
first on regions where C source transition leads talik onset (blue shading in Fig. 7C). In these,
thaw volume is low (< 50 m months $yr^{-1}$) and shows a weak relationship to NBP (NBP increases
much faster than thaw volume) prior to C source onset (indicated by large green circle in Fig.
8A). By the time thaw volume reaches 300 m months $yr^{-1}$ and talik formation occurs, these
regions are already very strong sources (NBP > 150 g C $m^{-2}$ $yr^{-1}$). This suggests that C sources in
these regions are not driven by respiration of old C from deep soil thaw, and thus alternative
explanations are needed.
Closer examination of thermal and moisture dynamics in shallow soils reveals three potential
indicators of C source transition: (1) seasonal duration of thaw, (2) depth of thaw, and (3) soil
drying. For example, vertical profiles of soil temperature and moisture (Fig. 9) in regions which
transition to C sources in the 2090s show deeper seasonal penetration of soil thaw, a jump in



active layer growth, and enhanced year round soil drying during the C source transition decade
(Fig. 9D). A broader analysis of soil thaw statistics over all regions and periods indicates that C
source transitions are most common when active layers grow to 1-2 meter depth and thaw
duration penetrates to Oct or Nov for the first time (Fig. 10).
Further examination of ecosystem biogeochemistry also shows high initial C stocks in these
regions (red shading in Fig. 7E). The median initial state of soil organic matter (SOM), 109 kg C
$m^{-2}$, is nearly a factor of 2 larger than the median value in regions where C source lags talik
onset (SOM = 59 kg C $m^{-2}$). These regions also show 40% less gross primary production (median
GPP = 755 vs 1296 g C $m^{-2}$ $yr^{-1}$) and higher over saturation prior to C source onset (water filled
pore space at 0.5 m depth at 10, 5, and 2 decades prior = 0.63, 0.59, and 0.57 mm3 $mm^{-3}$ for
cold permafrost, vs a near constant value of 0.57 mm3 $mm^{-3}$ in warm permafrost). The total
area of land in which SOM exceeds 100 kg C $m^{-2}$ represents 2/3 of all land where C sources lead
talik onset (2.0 million $km^2$), and peaks at a negative time lag of -5 to -6 decades (Fig. 7D, green
bars), which perfectly aligns with the peak distribution of negative time lags. These results
indicate peat like conditions characterized by saturated soils, high C stocks, and low annual
productivity which allow low thaw volumes (active layer depth < 2 m and peak thaw month of
October, on average) and rapid soil drying to produce early C losses in colder environments in
the absence of talik.
In regions where C source transitions lag talik onset (red shading in Fig. 7C), NBP is strongly
sensitive to changes in thaw volume until C source onset occurs (Fig. 8B), and talik formation
occurs when these regions are weak sinks (NBP < 0 g C $m^{-2}$ $yr^{-1}$). In general, C sources in these
regions are more sensitive to C emissions from deep soil thaw. However, as noted above,
neutral and positive time lags show a bimodal distribution peaking near 0 and 15 decades, and
thus additional explanations are needed. Further examination shows high fire activity in these
regions at the time of C source onset (red shading in Fig. 7F). The regions where fire C emissions
exceed 25 g C $m^{-2}$ $yr^{-1}$, representing our threshold for C source transition, are exclusively boreal
ecosystems, account for 1/3 of all land with negative lags (~1.1 million $km^2$), and align perfectly
with the peak distribution of positive time lags (Fig. 7D, red bars). NBP is less sensitive to thaw
volume in regions where fire dominates the C balance, which are strong C sinks at talik onset



(Fig. 8C), where soil C respiration is 13% less than non-fire regions (median SOM-HR = 331 vs
382 g C m$^{-2}$ yr$^{-1}$), and productivity is 25% more (median GPP = 1548 vs 1216 g C m$^{-2}$ yr$^{-1}$). Fire
regions are also 28% drier on average in the surface layer than non fire regions (volumetric soil
moisture = 0.28 vs 0.39 mm3 mm-3 in summer (May-Sep) in the upper 10 cm of soil). These
results suggest that soil thermal processes and talik formation are significant factors driving C
source transition in regions with reduced productivity, but fire activity, spurred by soil drying,
drives C source transition in higher productivity regions.
Figure 11 presents histograms of total permafrost area (Fig. 11A) and mean GPP (Fig. 11B) as a
function of decades since talik onset. We removed regions with high initial soil organic matter
(SOM > 100 kg m$^{-2}$, green bars in Fig. 7D) and fire C emissions (Fire > 25 g C m$^{-2}$ yr$^{-1}$, red bars in
Fig. 7D) during the C source transition decade. This screening yields a more normal distribution
of the decadal time lags between talik onset and C source transition (Fig. 11A), with a mean lag
of 1 decade form talik onset to C source. The high standard deviation of lags (± 8 decades)
reflects the skewed distribution of GPP (Fig. 11B); very low productivity in cold permafrost
increases the likelihood that soil thaw will lead to C source transition, while very high
productivity in warm permafrost decreases this likelihood.
Independent of the presence of talik, a key effect of an increasing number of thaw months is an
increasing rate of respiration from soil C pools. Warming and $CO_2$ fertilization increase the rate
of photosynthetic C uptake, increasing soil respiration mainly from younger near-surface C
pools; whereas deeper thawing affects both young and old C pools, so that the depth of thaw
dictates the timing and dominant C age of the net respiration flux. Fig. 12 illustrates this with a
comparison of decadal respiration trends for SOM (SOMHR) and litter (LITHR) C pools for C
source transitions in the mid 21$^{st}$ century, for scenarios where C source leads talik onset (blue
line, cold permafrost) and lags talik (red lines, warm permafrost). Here, we examine total
respiration (SOMHR+LITHR) and respiration difference (SOMHR=LITHR) from soil and litter C
pools.
GPP and total respiration show nearly linear increases (~15% per decade) for each permafrost
regime surrounding the decade of C source transition with peak fluxes in the growing season
(Fig. 12 A - D). Total respiration in cold permafrost is systematically larger than in warm



permafrost in the growing season (May – Sep) and smaller in the cold season (Oct – Apr). In
particular, total respiration is effectively zero for the late cold season (Jan – Apr) in cold
permafrost and significantly positive in warm permafrost over the same period. The respiration
difference also increases surrounding the C source transition (Fig. 12 E - F), but with 2 key
differences from total respiration: (1) the decadal increase is exponential, starting from a value
near zero just 3 decades prior to C source transition, and (2) peak respiration difference occurs
in late summer / early fall. Because litter respiration in the model is mainly drawing from C
pools with short turnover times, the litter respiration flux equilibrates rapidly to changes in
productivity and thus its change primarily reflects changes to inputs rather than decomposition
rates.  Conversely, soil C pools, which have much longer turnover times, equilibrate much more
slowly to the productivity changes and thus primarily reflect changes to the turnover times. The
trend in soil vs litter respiration explains almost the entire trend in net ecosystem C balance
from neutral to net source (Fig. 12 G – H). Furthermore, warm permafrost shows sustained
dominance of soil respiration during the entire cold season. These results are consistent with an
increasing thaw effect on C budgets during C source transitions, but where shallow thaw of
young soil C dominates in cold permafrost, and where talik formation and deep thaw of old soil
C dominate warm permafrost.
These results suggest that where talik forms, soil respiration increases throughout the year as
talik and perennial thaw mobilize deeper old soil C to respiration. In the absence of talik in
colder environments, soil respiration increases primarily in the NF season due to increased
availability of thawed shallow soil C. The lower GPP in colder regions suggests that increased
availability of substrate for respiration due to plant growth and soil C accumulation has less
impact on C source transition in our simulations than soil thaw dynamics and the initial state of
soil C. Thus, cold permafrost locations become C sources due only to thaw-season dynamics
while warmer permafrost locations transition to C sources due largely to changes in cold season
dynamics.

**4  Discussion**



Talik formation is widespread in our simulations, affecting ~14.5 million km$^2$ of NHL land in the
21$^{st}$ century. Simulations of the vertical thermal structure of soil thaw leading to talik in CLM4.5
qualitatively reproduce deep soil temperature data from borehole measurements in Siberia and
western North America, although rates of thaw at these and similar permafrost locations are
underestimated. Space-for-time comparisons along the north-south borehole transect in Alaska
and the Canadian Archipelago show a pattern of deepening and seasonal expansion of thaw
moving from the coldest location of the transect in northern Canada (Mould Bay) to the
warmest location in southeast Alaska (Gakona). Gakona shows the characteristic late cold
season thaw penetration into February at 2-3 meters depth which in our simulations signals
imminent talik onset (in the case of Gakona, as soon as the 2020s). Likewise, projected soil
thaw trends in east Siberia are in line with long term borehole measurements along the East
Siberian Transect, but the rate of talik formation here is also underestimated.
These comparisons indicate stable permafrost conditions in the colder sites in Siberia and N.
America through the 21$^{st}$ century, where thaw is generally slow, seasonally short, and stable.
This suggests talik formation in the northern Arctic is decades to centuries away, but potentially
sooner than the early 22$^{nd}$ century as projected by the CLM4.5 simulation. Our analysis finds
more unstable permafrost conditions to the south, with observed talik in the late 20$^{th}$ century
although simulated talik is delayed until the early 21$^{st}$ century.
Due to the potential for early 21$^{st}$ century talik and discrepancy between observed and
simulated trends in warm permafrost, continued model investigation of factors controlling the
rate of soil thaw is critically needed. In particular, large scale drying as projected in CLM4.5 near
the surface (Lawrence et al., 2015) may be restricting heat penetration and active layer growth
in the growing season, especially in organic rich soils which have very low thermal conductivity
(O'Donnell et al., 2009; Lawrence et al., 2011; 2012). Experiments demonstrating the sensitivity
of talik to soil drying within the active layer across soil hydrology schemes in previous (CLM4),
current (CLM4.5), and newly available (CLM5) versions of CLM could provide key insight on soil
thermal dynamics in frozen or partially frozen conditions.
Our simulations show robustly a pattern of accelerated soil C respiration following talik onset,
which shifts the surface C balance of photosynthetic uptake and litter respiration from net C





sinks to long term net sources across 3.2 million km$^2$ of NHL land by 2300. The pattern of C
source transition following talik formation is most evident in warm permafrost in the sub-Arctic,
suggesting increased microbial decomposition with warming soils. We also find evidence of talik
driven soil drying near the surface associated with increased active layer thickness and higher
available water storage, which can lead to enhanced decomposition rates by causing soils to be
less frequently saturated/anoxic (Lawrence et al., 2015). At the same time, these regions show
high ecosystem productivity which increases roughly in proportion to respiration, and thus
appears to be driven by combination of warming and increased nitrogen availability resulting
from permafrost thaw (Mack et al., 2004; Natali et al., 2012; Koven et al., 2015). As such, the
transition time to sustained net ecosystem C source is delayed by 1-2 centuries following talik
onset as productivity continues to outpace respiration as currently observed (Belshe et al.,
2013; Mack et al., 2004), with C balance transitions peaking in the mid- to late 22$^{nd}$ century. In
nearly 1/3 of these regions, an estimated 2 million km$^2$ of land, fires are a primary mechanism
triggering C source onset, rather than talik. Consequently, in regions of very high productivity,
talik appears to serve more as an indirect driver of long term C sources through accelerated soil
drying, rather than as a direct driver through accelerated respiration of deep soil C.
We identify an equally large region of land in the high Arctic, representing ~3.0 million km$^2$,
which is projected to transition to long term C sources much sooner than the sub-Arctic and in
the absence of talik. This region, distributed across northern Siberia and North America,
resembles peatlands and is characterized by cold permafrost, high soil C stocks and soil
moisture, and low productivity. Thawing in this cold northern permafrost is limited to young,
shallow soils with significantly reduced contributions from deeper, older C than warm
permafrost, but with a factor of 2 higher C stocks. These C rich soils become increasingly
vulnerable to decomposition as they are exposed to increased warming and drying as active
layers deepen and persist deeper into the cold season. The transition to long term C sources in
this region peak is expected to peak between 2050 and 2100, nearly a century prior to talik
driven sources in warm permafrost, and decades to centuries prior to talik onset, which
eventually amplifies C sources in this region.





These results have important implications for designing an Arctic monitoring system to
simultaneously detect changes in the soil thermal state and C state. In particular, C
observations should not be limited to warm permafrost regions of the sub-Arctic, since cold
northern permafrost regions are projected to become C sources even without forming talik.
Our analysis of the seasonal dynamics and vertical structure of permafrost thaw and soil C
emissions provides a general strategy for concurrent observing warm and cold permafrost
based on time of year and depth of thaw.
Observing warm permafrost will require year round measurements of ground thermal state to
detect precursors to talik onset including thaw penetration at depth (~2-3 m) and late into the
cold season (~Jan-Feb), as well as sustained cold season C flux observations to detect changes in
C balance associated decomposition and respiration of deep, old soil C. Continued monitoring
of these depths will require sustained long term measurements from deep boreholes, and
increasing reliance on remote sensing technologies such as Electromagnetic Imaging (EMI). In
particular, EMI surveys along the continuous/discontinuous permafrost transition zones during
the cold season from November – March are likely to provide key thermal state diagnostics.
Systematic radiocarbon ($^{14}$C) measurements, which can be used to partition respiration into
autotrophic and heterotrophic young and old soil components (Hicks Pries et al. 2015), would
provide a valuable tool to help disentangle and track future C emissions from deep permafrost,
especially during the long cold season when talik enables the microbial decomposition of deep
old C and is the primary source of C emissions.
Observing cold permafrost in the high Arctic is both more urgent, due to earlier shifts in C
balance, and more complicated, due to challenging observing conditions (remote, cold, and
dark) and less detectable signals in thermal state (e.g., talik) and C age (e.g., depleted in
radiocarbon) change. Our results suggest sustained observation of year round soil thermal and
hydrological profiles (soil drying; depth and duration of thaw at 1-2 meter depth) using
boreholes and EMI surveys, and cold season net $CO_2$ exchange (Sep – Oct) using atmospheric
$CO_2$ sensors and eddy covariance towers, can help detect changes in soil thaw and soil vs litter
respiration driving annual C balance changes. We also recommend an observing network





focused on regions rich in soil organic matter, where our simulations indicate increased
sensitivity of soil decomposition to warming.

**5    Conclusion**
Greening trends driven by high latitude warming and $CO_2$ fertilization have led to amplification
of the contemporary C cycle, characterized by increasing photosynthetic C uptake during the
short growing season and increasing respiration of recent labile soil C during the cold season
(Mack et al., 2004; Piao et al., 2008; Randerson et al., 1999; Graven et al., 2013; Forkel et al.,
2016; Wenzel et al., 2016; Webb et al., 2016). Our simulations of C-climate feedbacks with
interactive terrestrial biogeochemistry and soil thaw dynamics indicate this trend continues
mostly unabated in NHL ecosystems. However, sustained warming over the next 300 years
drives accelerated permafrost degradation and soil respiration, leading to widespread shifts in
the C balance of Arctic ecosystems toward long term net C source by the end of the $23^{rd}$
century. 6.8 million $km^2$ of land impacted in Siberia and North America will produce an
integrated C source of 11.6 Pg C from 2010-2300, peaking at 0.7-0.8 Pg C $yr^{-1}$ in the mid- to late-
$22^{nd}$ century. Our projected permafrost C feedback is comparable to the contemporary land
use/land use change contribution to the annual C cycle.
Our main results emphasize an emergence of cold season processes driven by amplified winter
warming, earlier spring thaw, longer NF seasons, and increased depth and seasonal duration of
soil thaw. Our simulations are consistent with soil thaw patterns observed from borehole time
series in Siberian and North American transects during the late $20^{th}$ and early $21^{st}$ centuries.
Patterns of deeper and longer thaw drive widespread talik, and exposes Arctic soils to increased
warming and drying, which accelerates decomposition and respiration of deep, old C, and shifts
ecosystem C balance to a state increasingly dominated by soil respiration.
The timing with which Arctic ecosystems transition to long term net C sources depends on a
number of factors including talik onset, vegetation productivity, permafrost temperature, soil
drying, and organic matter. The timing is most sensitive to talik onset in warm permafrost
regions in the sub-Arctic, which account for a total of 3.2 million $km^2$ of land, representing ~50%



of our simulated permafrost region. These regions are also the most productive, which can
delay the transition to net C source by decades or even centuries. As such, warm permafrost
regions typically do not transition to net C sources until the mid-22$^{nd}$ century.
The cold permafrost region in the northern Arctic, which accounts for an additional 3.0 million
km$^2$ of land, transitions to net C source in the late 21$^{st}$ century, much earlier than warm
permafrost and in the absence of talik. High decomposition rates, driven by warming and drying
of shallow, young C in organic rich soils, and low annual productivity make this region perhaps
the most vulnerable to C release and subject to further amplification with future talik onset.
This result is surprising given the region is dominated by tundra and underlain by deep, cold
permafrost that might be thought impervious to such changes.
Rather than thinking of the permafrost feedback as being primarily driven by a single coherent
geographic front driven by talik formation along the retreating boundary of the permafrost
zone, this analysis suggests multiple modes of permafrost thaw with a mosaic of processes
acting in different locations. Active-layer deepening leads to C sink-to-source transitions in
some regions, talik-driven permafrost loss in others, fire-driven changes in other places, and
thaw-led hydrologic change in yet others.  Our results reveal a complex interplay of amplified
contemporary and old C cycling that will require detailed monitoring of soil thermal properties
(cold season thaw depth, talik formation), soil organic matter content, soil C age profiles,
systematic $CO_2$ flux, and atmospheric $^{14}CO_2$ measurements to detect and attribute future C
sources. Further investigation of soil thermal properties and thaw patterns is required to
understand C balance shifts and potential further amplification of emissions from high northern
latitudes.

**Acknowledgements**
DML is supported by U.S. Department of Energy, Office of Biological and Environmental
Research grant DE-FC03-97ER62402/A0101. CDK is supported by the Director, Office of Science,
Office of Biological and Environmental Research of the US Department of Energy (DOE) under
Contract DE-AC02-05CH11231 as part of their Regional and Global Climate Modeling (BGC-





Feedbacks SFA), and Terrestrial Ecosystem Science Programs (NGEE-Arctic), and used resources
of the National Energy Research Scientific Computing Center, also supported by the Office of
Science of the US Department of Energy, under Contract DE-AC02-05CH11231. National Center
for Atmospheric Research (NCAR) is sponsored by the National Science Foundation (NSF). The
CESM project is supported by the NSF and the Office of Science (BER) of the US Department of
Energy. Computing resources were provided by the Climate Simulation Laboratory at NCAR's
Computational and Information Systems Laboratory, sponsored by NSF and other agencies.
Some of the research described in this paper was performed for CARVE, an Earth Ventures (EV-
1) investigation, under contract with NASA. A portion of this research was carried out at JPL,
California Institute of Technology, under contract with NASA. © 2017. All rights reserved

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





**Tables**
**Table 1**: Site information for long-term borehole temperature measurements along the East
Siberian Transect for the period 1957-1990. The 9 sites reported in this table, presented in a
north-to-south order, meet the criteria of at least one year of valid soil temperature data (>= 10
months per layer, $\geq$ 55 months across 5 layers). Talik is observed in 4 of 9 sites, 2 of which is
observed in the first year of valid reported data. Site-specific thaw trends are provided for sites
with at least 6 years of valid data. Regional trends are calculated from all available data for 3
regional locations.

| Site | Location | Date Range | Years with Valid Data | First Obs Talik | Site Trend (m mo yr$^{-1}$) | Region | Regional Trend (m mo yr$^{-1}$) |
|---|---|---|---|---|---|---|---|
| Drughina | 145.0°E, 68.3°N | 1969-1990 | 8 | N/A | -0.083 | N Siberia | -0.057 |
| Ustmoma | 143.1°E, 66.3°N | 1973-1975 | 3 | N/A | N/A | | |
| Chumpuruck | 114.9°E, 60.7°N | 1981-1984 | 4 | N/A | N/A | SW Siberia | 0.019 |
| Lensk | 114.9°E, 60.7°N | 1957-1990 | 11 | 1957 | 0.23 | | |
| Macha | 114.9°E, 60.7°N | 1970-1990 | 13 | 1970 | 0.070 | | |
| Oimyakon | 114.9°E, 60.7°N | 1966-1974 | 6 | N/A | 0.059 | | |
| Tongulakh | 114.9°E, 60.7°N | 1966-1966 | 1 | N/A | N/A | | |
| Uchur | 114.9°E, 60.7°N | 1966-1990 | 17 | 1974 | 0.24 | | |
| Chaingda | 130.6°E, 59.0°N | 1967-1990 | 8 | 1989 | 0.51 | SE Siberia | 0.51 |





**Table 2**: Site information for borehole temperature measurements at 3 sites along a north-to-
south transect in North America for the period 2004-2012. Climatological soil thermal state
presented on a site-to-site basis Fig. 5 are based on all available valid monthly data for each
site, with valid data requiring at least 20 days of reported data for each layer. Layer of Deepest
Thaw represents the deepest layer in which mean soil temperature exceeds freezing (> -0.5°C)
in at least 1 month. Month of Latest Thaw represents the latest month in which mean soil
temperature exceeds freezing. Here, we define May as the earliest possible month and April as
the latest possible month.

| Site | Location | Date Range | Depth / Number of Layers | Layer of Deepest Thaw | Month of Latest Thaw |
|---|---|---|---|---|---|
| Mould Bay, Canada | 119.0°W, 76.0°N | 2004-2012 | 3 m / 36 | 0.69 m | September |
| Barrow2, Alaska | 156.0°W, 71.3°N | 2006-2013 | 15 m / 63 | 0.58 m | October |
| Gakona1, Alaska | 145.0°W, 62.4°N | 2009-2013 | 30 m / 35 | 2.5 m | February |





**Figures**

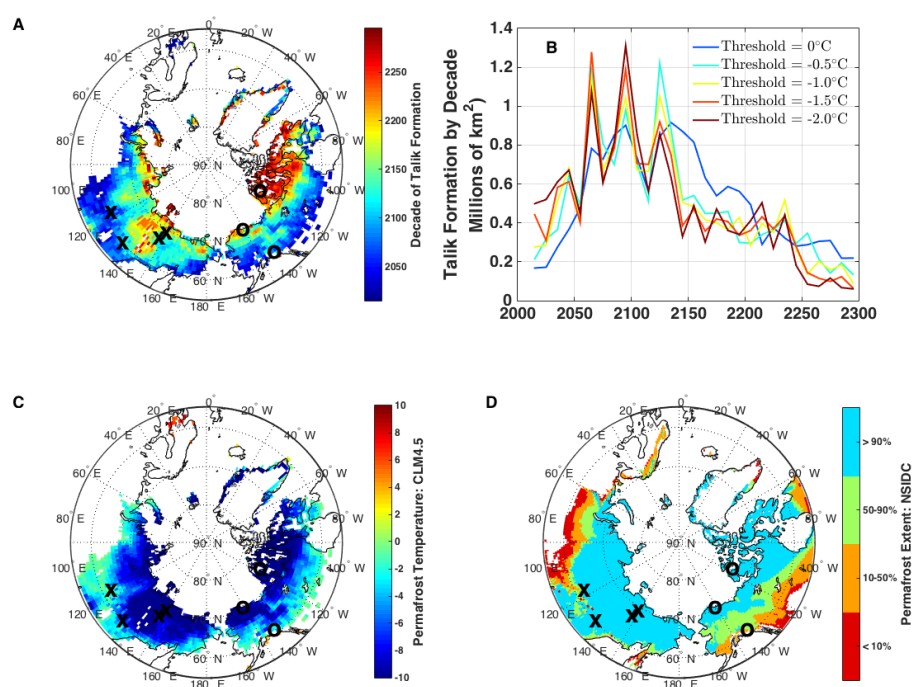


**Figure 1**. Decade of projected talik formation and correlation to initial state of simulated
permafrost temperature and observed permafrost extent. (A) Maps and (B) time series of the
simulated decade of talik formation are estimated from CLM4.5 as the first decade when the
mean temperature of a soil layer exceeds a freeze/thaw threshold of -0.5°C in every month. (C)
Initial permafrost temperature is defined as the annual mean soil temperature at 3 m depth
from 2006-2010. (D) Permafrost extent is taken from
(https://nsidc.org/data/docs/fgdc/ggd318_map_circumarctic/; Brown et al., 2001). Crosses in
A, C, D represent locations of Siberian borehole measurements along the East Siberian Transect
from 1955-1900 (Table 1). Circles represent locations of borehole measurements in Alaska and
Canada from 2002-2013 (Table 2). We note that peak transition time occurs around 2100.





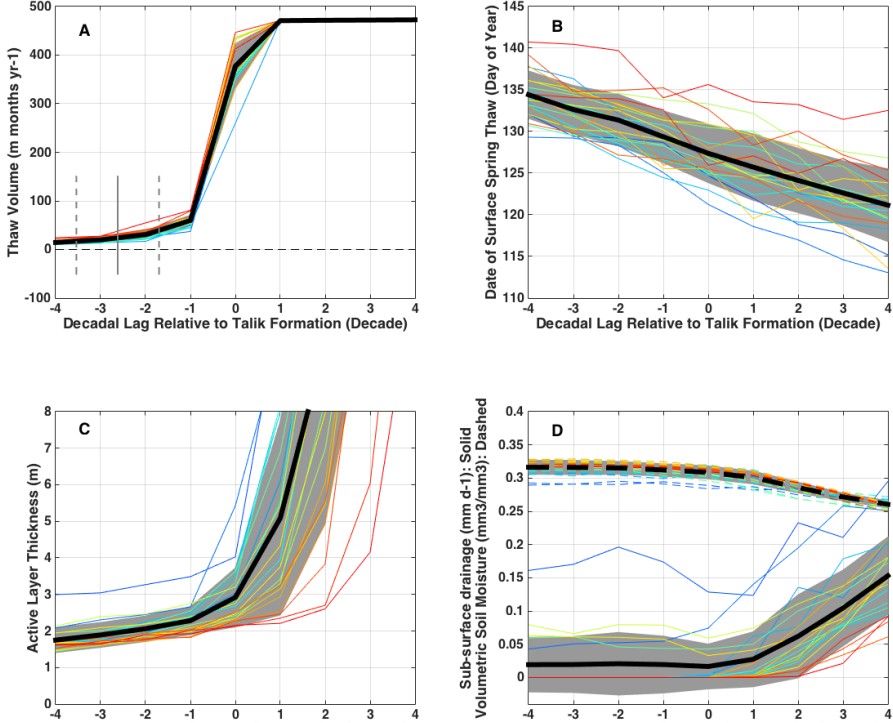

**Figure 2**. Patterns showing the progression of soil thaw in the decades surrounding talik onset.
Individual lines represent averages across the subset of talik forming regions for each decade
from the 2050s (darkest red) to the 2250s (darkest blue). (A) Integrated soil thaw volume,
where the vertical solid line represents the mean timing of initial thaw at depth and late into
the cold season (Jan-Apr). Note that the upper limit to the thaw volume metric in (A) is an
artifact of the arbitrary maximum soil depth of 45.1m in CLM4.5. Other panels show (B) Date of
spring surface thaw in the uppermost layer, (C) annual maximum active layer thickness, and (D)
annual sub-surface drainage (solid) and volumetric soil moisture averaged over the soil column
(dashed) and. Grey shaded areas show the standard deviation of results for individual talik
formation decades. Mean behavior exhibits a characteristic pattern: gradual increase in thaw
volume and active layer depth prior to talik onset, abrupt shift in thaw volume, active layer
depth, followed by stabilization to constant thaw volume as soil drying and sub-surface
drainage increases.




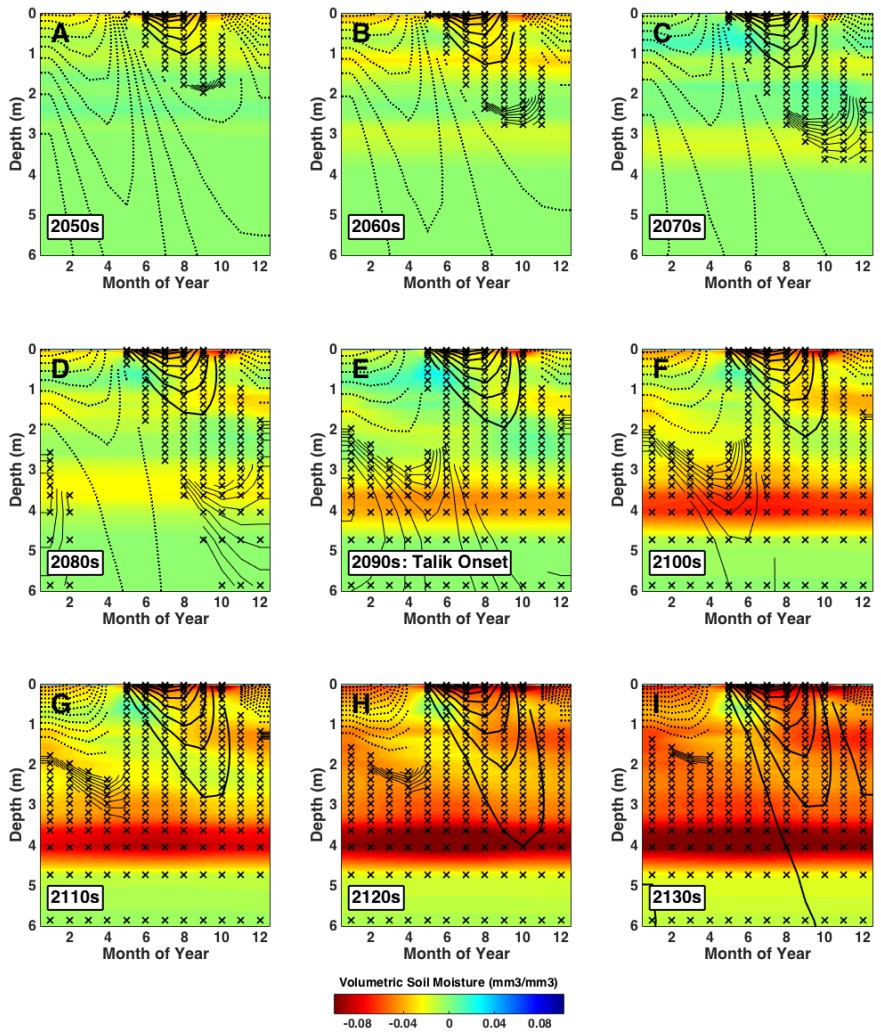


**Figure 3**. Evolution of simulated decadal thermal and hydrological state as functions of month
and depth averaged across talik forming regions in the 2090s. Each panel presents decadal
average seasonal profiles in the decades surrounding talik onset from the 2050s (A) to the
2130s (I). Contours are soil temperature in 0.5°C intervals, with solid (dashed) lines denoting
temperature above (below) a freeze/thaw threshold of -0.5°C. Stars indicate "thaw" months
where soil temperature exceeds -0.5°C. Color shading is volumetric soil moisture anomalies
relative to the 2040s, where red indicates drying. Note that soil depth on y-axis is plotted on a





non-linear scale. The soil thaw profile exhibits a shift from predominantly frozen and wet to
perpetually thawed and drying conditions at depth while remaining seasonally frozen near the
surface.





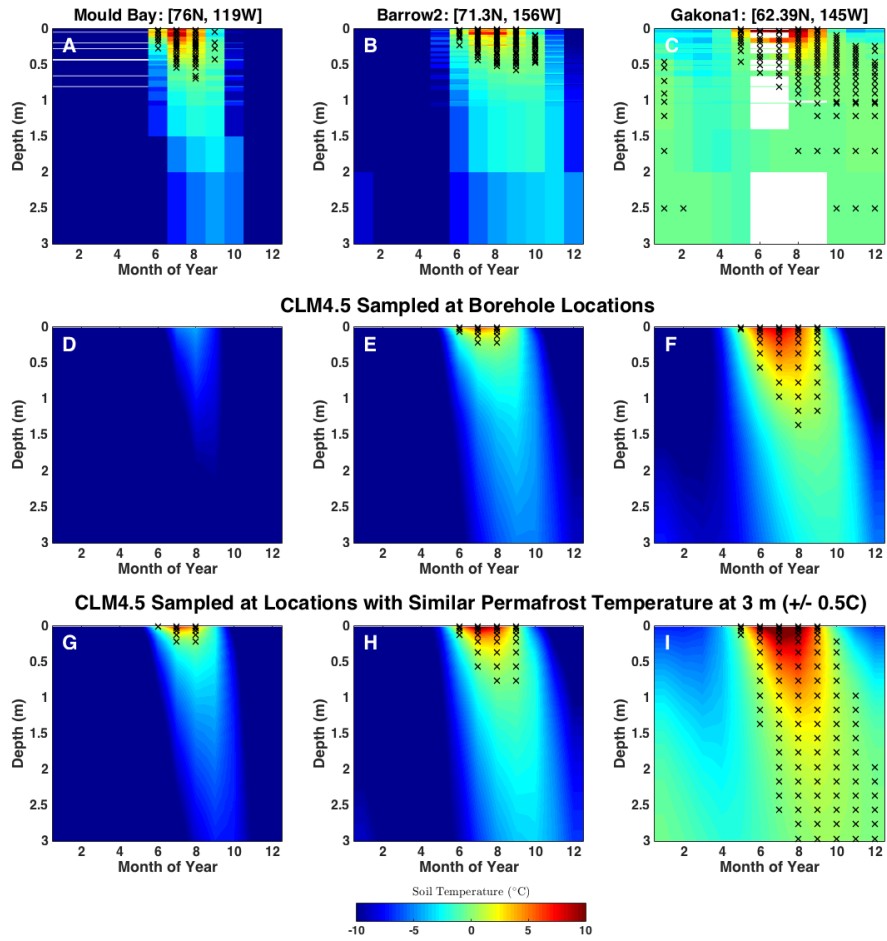


**Figure 4**. Observed and simulated early 21[st] century soil thermal state as a function of month

and depth for the North American Transect boreholes (black circles, Fig. 1). Top Row: Observed

multi-year means for Mould Bay, Canada (2004-2012), Barrow, Alaska (2006-2013), and

Gakona, Alaska (2009-2013). The color scale shows the mean temperature and the stars mark

the months when each layer is thawed (T > -0.5°C). Simulated soil thermal state from 2006-

2010 for borehole locations (Middle Row) and regions with 3 m permafrost temperature within

0.5°C of observed (Bottom Row) show similar north-to-south spatial gradient to observations,

especially for similar permafrost temperature. Note that the thaw state at Gakona, Alaska

persists at depths of 1-3 m into the deep cold season (Jan-Feb), perhaps signaling the threshold

for rapid talik formation (see Fig. 3D).






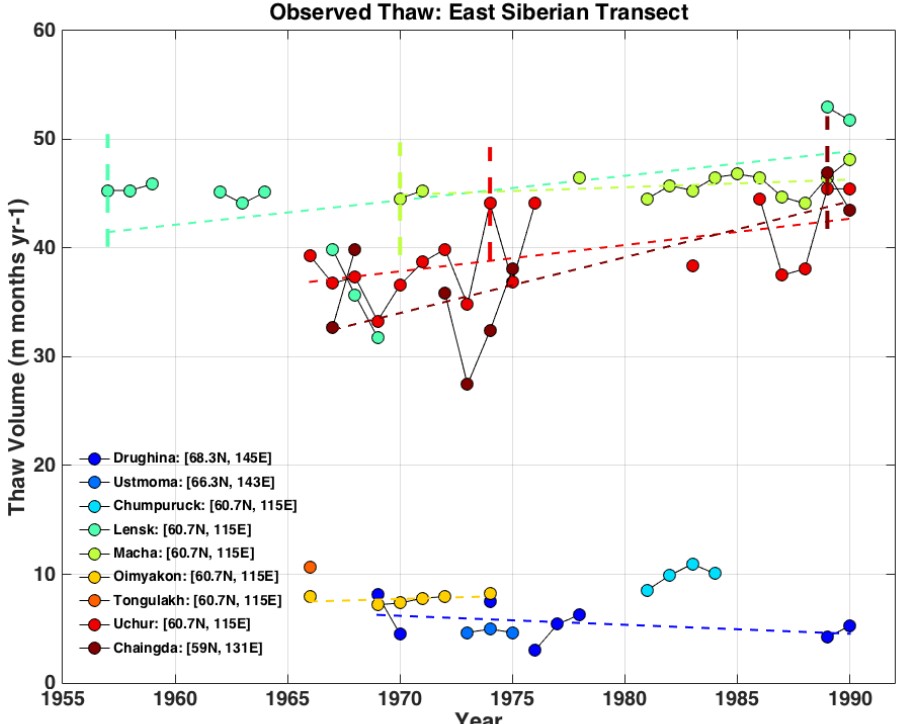


**Figure 5**. Soil thaw observation time series from borehole measurements of soil temperature at
sites along the East Siberian Transect over various periods from 1957 – 1990. Site coordinates
are provided in the legend and plotted as crosses on the map provided in Fig. 1. Thaw trends
are derived from estimates of thawed volume over a depth of 3.2 m for sites with > 5 years of
data over multiple decades: Drughina, Lensk, Macha, Uchur, and Chaingda. Trend values are
reported in Table 1. Vertical dashed lines mark the onset of talik formation at Lensk (1957),
Macha (1970), Uchur (1974), and Chaingda (1989). Sites in southern Siberia show significant
negative thaw volume trends over the 20[th] century, representing net increases in soil thaw. The
trend at Drughina is not statistically significant, indicating that soil thaw is unchanged in
northern Siberia.





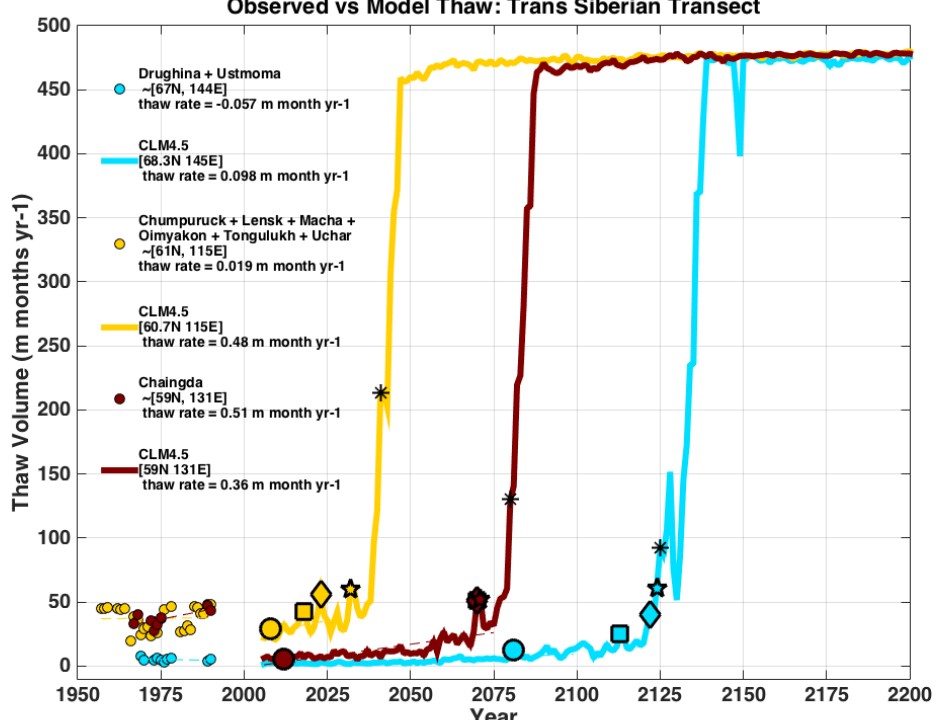


**Figure 6**. Comparison of 20th century observed (markers) and 21st century projected (solid lines)

soil thaw at sites along the East Siberian Transect (crosses in Fig. 1). Observed thaw from 1955-

1990 is based on soil thaw data in Fig. 5 and on the inter-site average at 3 locations: northern

Siberia (blue), southwest Siberia (orange), and southeast Siberia (brown). Simulated thaw from

2006-2200 is derived from CLM4.5 and sampled at the nearest grid cell of 3 above locations.

Markers represent thresholds for thaw onset in January (circle), February (square), March

(diamond), April (star), and talik onset (asterisk). Thaw trends are derived from estimates of soil

thaw volume. We note a key discrepancy between observed and simulated thaw volume:

Simulated thaw volume is integrated over depths from 0-40 meters; observed thaw volume is

integrated from 0-3.6 meters. The effect of this selection bias is a potential low bias in observed

thaw volume. In general, soil thaw is projected to remain stable in northern Siberia but become

increasingly unstable in southern Siberia.

883





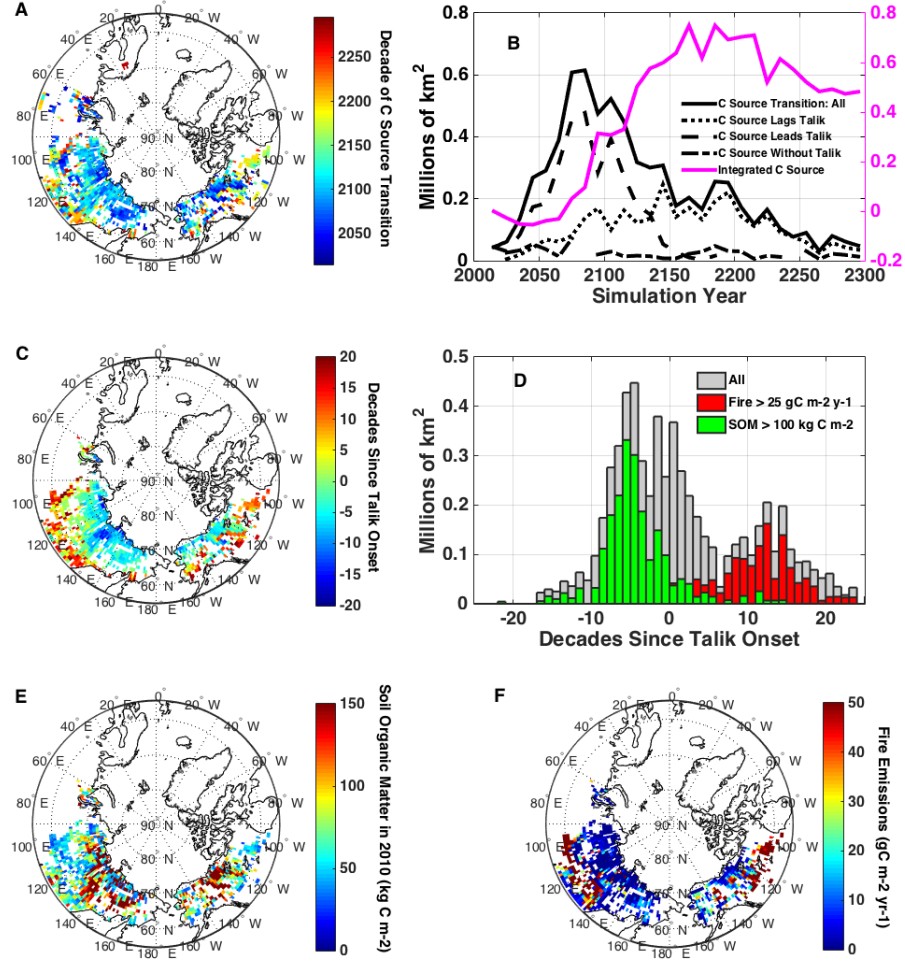

884

**Figure 7**. Projected decade when permafrost regions shift to long-term C sources over the
period 2010-2300, and relation to talik onset, soil C, and fire emissions. (A) Map of the decade
of transition to C source, reflected in the color code, showing earlier transitions in cold northern
permafrost. (B) The area of land that transitions peaks in the late 21$^{st}$ century, and is driven by
regions where the C source leads talik onset (dashed). The integrated C source from these
regions (magenta) peaks in the late 22$^{nd}$ century. (C) The decadal time lag from talik onset to C
source transition shows positive lags in warm southern permafrost (C source lags talik) and
negative lags in cold norther permafrost (C source leads talik). (D) Histogram shows trimodal



distribution of permafrost area as a function of decadal time lag, with negative lags related to
high soil organic matter (green bars and map in E), and large positive lags related to fires (red
bars and map in F) but delayed by high productivity (Fig. 11B). See text for details.




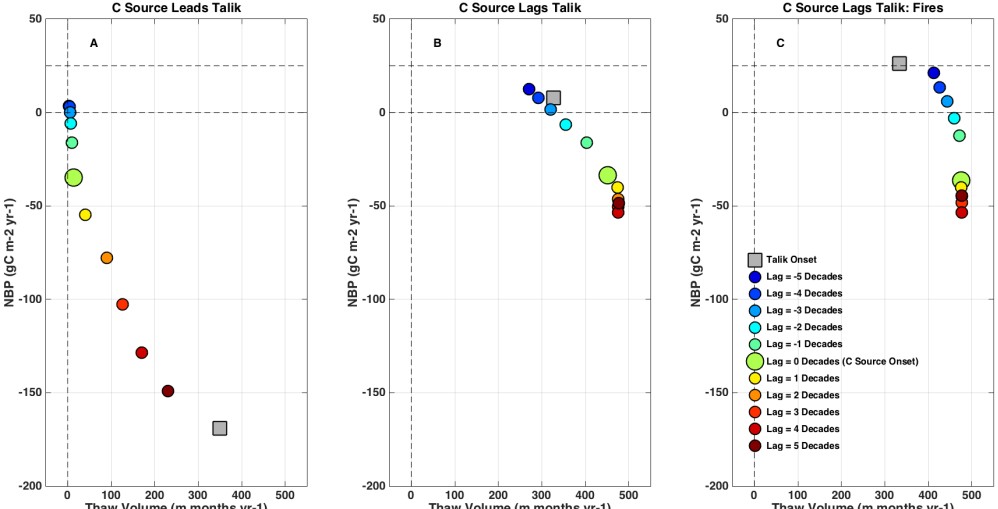


**Figure 8**. Net biome production (NBP) as a function of thaw volume. Symbols represent NBP

and thaw volume values averaged over regions which transition to long term C source from

2060-2140, binned into regions where the decade of C source transition (A) leads talik onset,

(B) lags talik onset, and (C) lags talik onset AND where fires exceed 25 g C m$^{-2}$ yr$^{-1}$. Colors

indicate decade relative to C Source transition, denoted by the large green marker, which

occurs when NBP exceeds 25 g C m$^{-2}$ yr$^{-1}$ (grey horizontal dashed line). The grey square marker

indicates the mean NBP and thaw volume values during talik onset. Cases where C source leads

talik (A) show small thaw volumes during C source transition, and amplified C sources during

talik onset. Cases where C source lags talik (B-C) show large thaw volumes during C source

transition, and C sinks during talik onset.



914

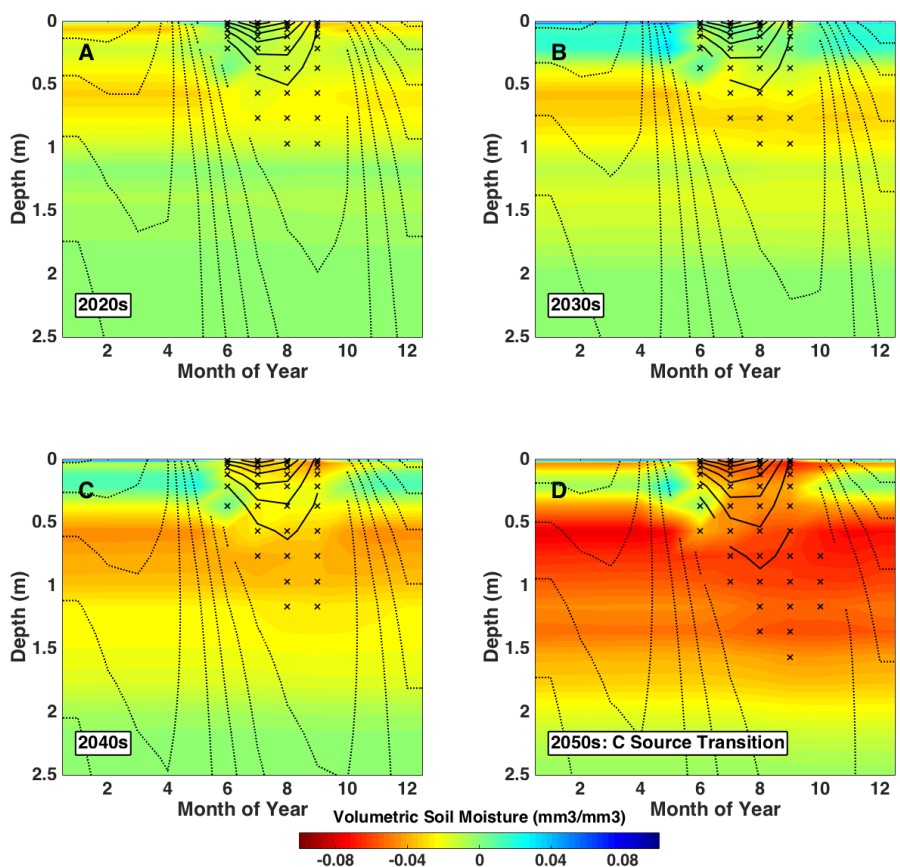

915

**Figure 9**. Evolution of simulated soil thermal and hydrological state, plotted as a function of

month and depth, for regions which transition to long term C sources in the 2060s but don't

form talik for another 3 decades (≥ 2090s). This represents cases where C Source leads talik

(e.g., Fig. 8B). Each panel presents decadal average seasonal profiles in the decades leading up

to C source transition. Shading and contour details are explained in Fig. 3. These profiles exhibit

shifts in thaw period (Oct), depth (> 1.5 m), and soil moisture (drying) in the transition decade.

922



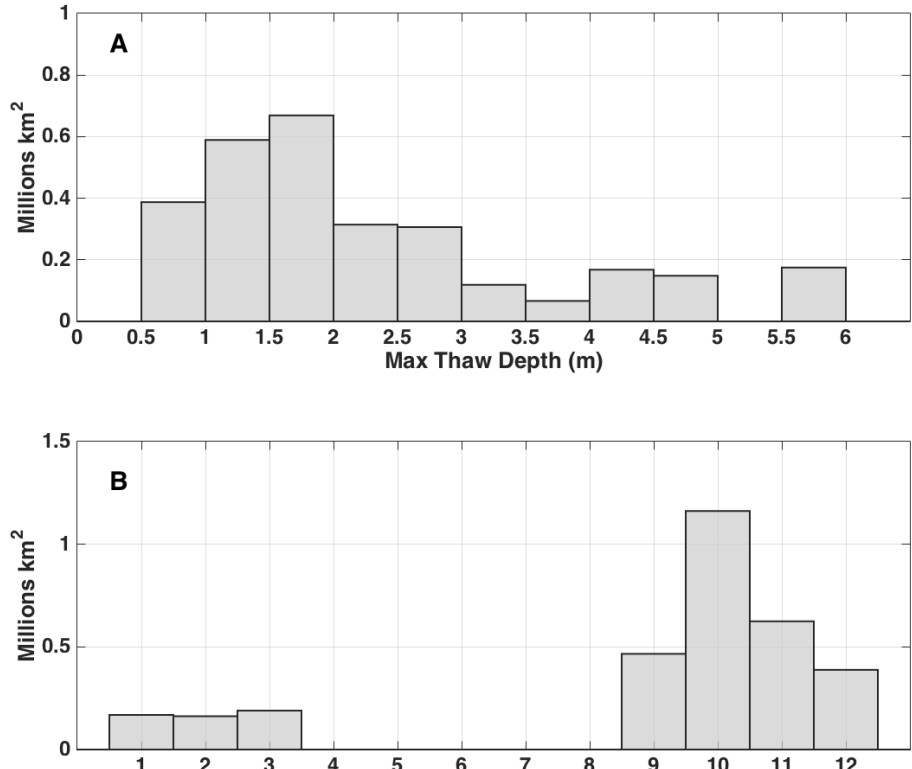

923

**Figure 10**. Histogram of total permafrost area as a function of (A) maximum thaw depth and (B)

maximum thaw month, for regions which transition to long term C sources prior to talik onset.

Here, thaw depth represents the depth of the active layer. This shows that most regions

transition to C sources when the active layer grows to a depth of 1-2 meters and reaches a peak

thaw month at depth (> 1 m) of October.

929





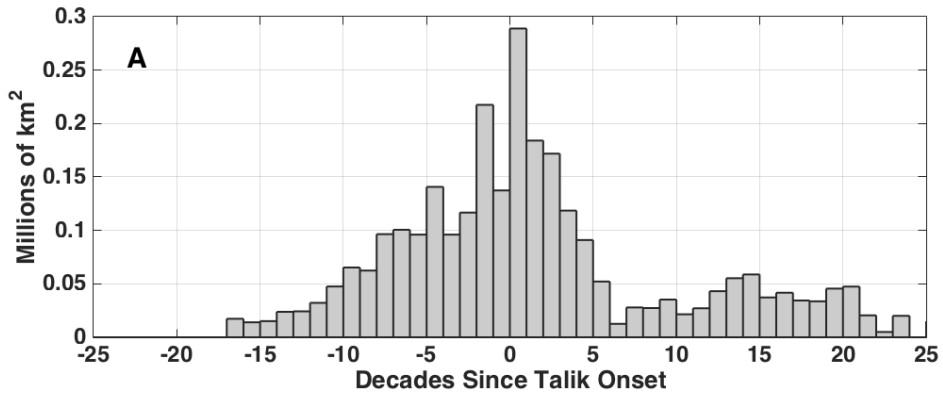

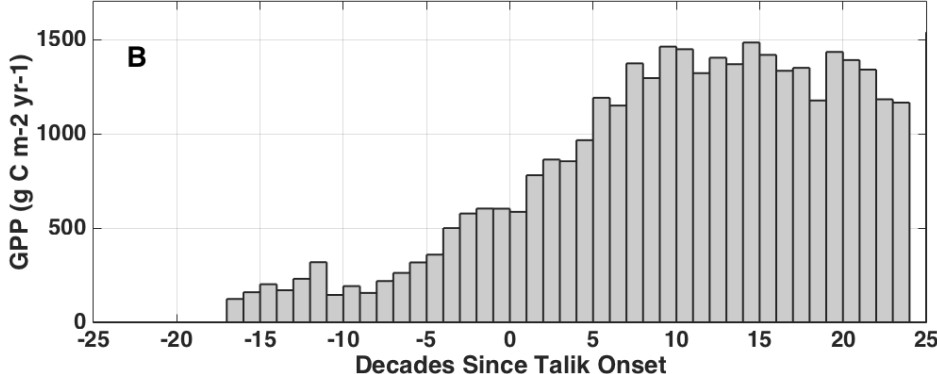

930

**Figure 11**. Histogram of total permafrost area (A) and mean GPP (B) as a function of decades

since talik onset. Regions with high initial soil organic matter (SOM > 100 kg m$^{-2}$) and fire C

emissions (Fire > 25 g C m$^{-2}$ yr$^{-1}$) during the C source transition decade are removed, such that

factors driving the trimodal distribution when all regions are excluded (Fig. 7D). In the absence

of these factors, most regions transition to long term C sources in 1 decade following talik

onset. The large standard deviation (8 decades) is related to gross primary production (GPP),

with long transition periods (right tail) correlated to high productivity (GPP > 1000 g C m$^{-2}$ yr$^{-1}$)

and negative transition (C source onset in the absence of talik) correlated to low productivity

(GPP < 250 g C m$^{-2}$ yr$^{-1}$)

940



941

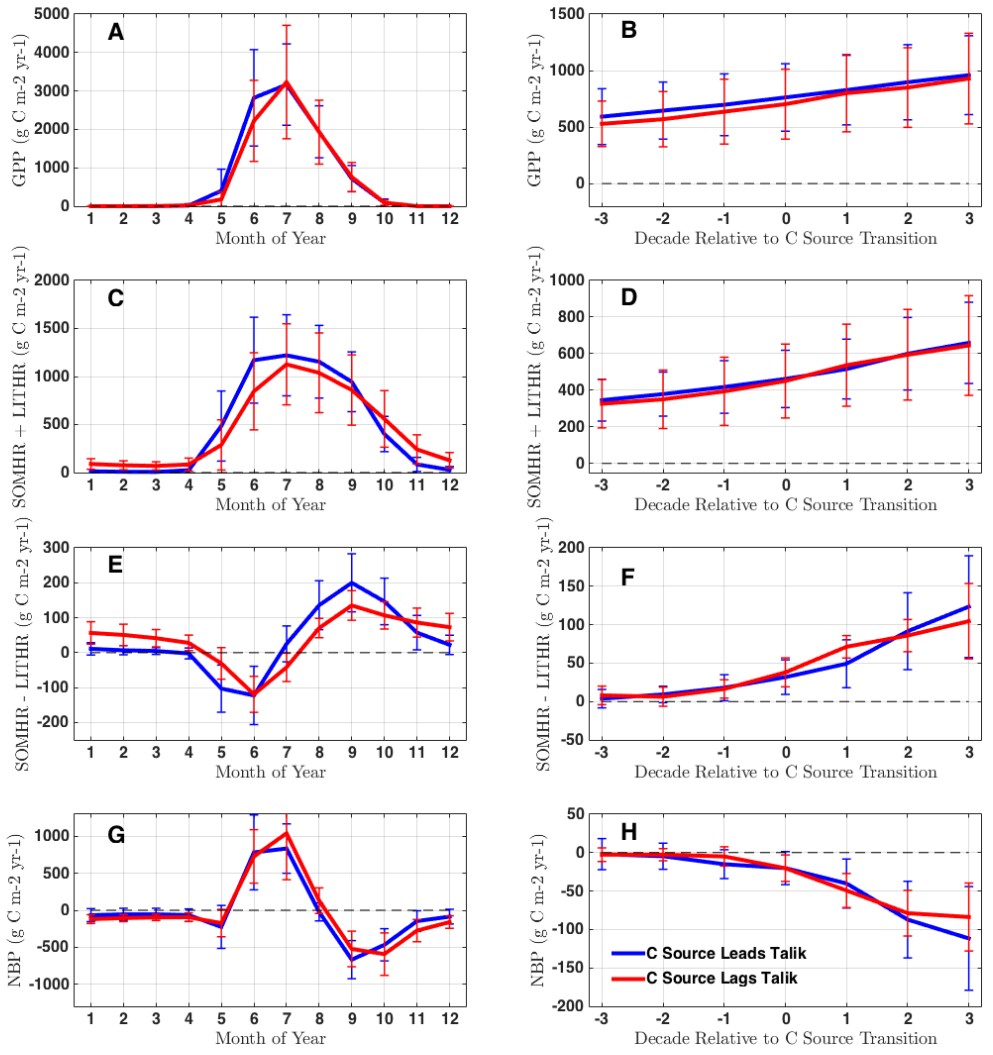

942

**Figure 12.** Time series of ecosystem C fluxes showing seasonal and decadal patterns during C

source transition. This present results for (A-B) Gross Primary Production (GPP), (C-D) Sum of

respiration from soils (SOMHR) and litter (LITHR), (E-F) Difference of respiration from soils and

litter, and (G-H) Net Biome Production (NBP). The left column show seasonal fluxes during the

decade of C source transition. The right column shows the evolution of decadal mean fluxes in

the 3 decades preceding and following C source transition. Regions where C source transition



leads talik (blue) show similar patterns to regions where transition lags talik (red). Specifically,
this shows a sudden jump in respiration (F) during C source transition in both cases, which
corresponds in time and magnitude to the jump in NBP (H). The primary difference between
regions is the seasonal distribution of SOMHR vs LITHR (E), which shows a large soil respiration
source throughout the cold season in cases where C sources lag talik. This indicates an annual
source of deep old C.
