# Peer review of "The Cryosphere Discuss., https://doi.org/10.5194/tc-2017-189 Manuscript under review for journal The Cryosphere Discussion started: 18 September 2017 © Author(s) 2017. CC BY 4.0 License."

_The Cryosphere, 2017_

## Referee Comment (RC1) · Anonymous Referee #1 · 13 Oct 2017

Review of "Detecting the permafrost carbon feedback: Talik formation and increased cold-season respiration as precursors to sink-to-source transitions"

The authors ran the Community Land Model (CLM) version 4.5 up to 2300, using RCP 8.5 forcing. They then perform an in depth analysis of permafrost-region dynamics in this simulation, including identifying key events: Talik formation (related the degradation of permafrost) and sink-to-source transition, i.e. the point at which the land surface changes from a net sink of carbon from the atmosphere, to a net source. It is interesting to note that this behaviour (starting as a sink and transitioning to a source) is identified

across a large fraction of the current permafrost zone. However, the total carbon source is apparently only 11.6 GtC by 2300, which is low compared with previous estimates.

The authors extensively analyse different variables such as thawed volume (a newly defined metric), active layer depth, primary production, respiration and fires, and how these influence talik formation and sink-to-source transition. They find three main drivers of sink-to-source transition: 1. Active layer thickening in cold, carbon-rich high Arctic permafrost 2. Talik formation leading to winter respiration in low Arctic, warmer soils 3. Fire driven carbon source in more productive regions which dry out, and a lot of vegetation is burned. They also showed some indicators of talik formation such as a rapid increase in thawed volume immediately preceding talik formation.

This is a very thorough analysis and a well written paper that will make a great publication in The Cryosphere, after some small revisions. In general I would like to see a bit more analysis about the size of the carbon sources, not just the timing of transition. This could comprise a bit of discussion of the cumulative carbon source (11.6 Gt), and the significance of this - compared to previous estimates, and the time trajectory of the cumulative source (i.e. when does the Arctic as a whole become a net source?). And then if they could break that down to say which of the different types of source (driven by AL, talik or fires) has the bigger contribution to the total source or if these are all comparable magnitude, that would be add some value to the paper. It's fine saying that we should monitor the high Arctic systems as they will become a source soonest, but if this source is likely to be very small, there would not be so much point?

I also suggest considering the soil types at the boreholes. It would hopefully be possible to get an idea of this from a site description or by asking the PI. For example if these are peaty soils that might explain the very slow progression of freeze-thaw compared with CLM (also relatedly, the water content).

Finally the paper is rather long and I would suggest reducing in length where possible. I have indicated a couple of points below.

Line-by-line comments

Introduction: L62: 'Shifts in vegetation community' is mentioned in the introduction as being an important factor, but is this considered here? Are you running with dynamic vegetation? If you are, this should mentioned and if not, this omission should be discussed later on. L70: Same for soil organic matter export by rivers.

L86-87: Include more recent references for total permafrost carbon quantity, such as Hugelius et al 2014, Biogeosciences (https://www.biogeosciences.net/11/6573/2014/bg-11-6573-2014.pdf), and there is also a new paper by Jackson et al coming out in November with revised estimates, this will be in Annual review of Ecology, Evolution and Systematics (http://www.annualreviews.org/doi/abs/10.1146/annurev-ecolsys-112414-054234).

Methods: L138-142 The standard RCP 8.5 only goes until 2100 so presumably some extension is used here? Could you mention what this looks like - for example, does it stabilise at some point or does the global temperature and $CO_2$ just keep increasing? I also think it would be useful for comparing with the permafrost thaw results, to see a plot of the global temperature across the three centuries of future simulation. I suggest adding this at least as a supplementary figure (as there are already a lot of figures in the main manuscript).

L152-154 "The C source transition represents a shift of ecosystem C balance from a neutral or weak C sink to a long-term source driven by onset of permafrost thaw and respiration of deep SOM" - here you suggest that the deep SOM alone is driving the transition, whereas your analysis suggests that it is driven by different things depending on region. Maybe you can qualify this sentence a bit?

L201-204: "For comparison to projected trends in CLM4.5, we recalculate observed trends using the inter-site average from all 9 sites at 3 unique locations: northern Siberia (67°N, 144°E), southwest Siberia (61°N, 115°E), and southeast Siberia (59°N, 131°E)." This is not entirely clear. You were talking about using 6 sites and now it says

9, but then you end up with 3? Can you make it more clear? Did you combine sites into groups based on approximate locations. . .?

Results: L228 Do you mean 2300?

L266 "Our simulations show a similar drying pattern in shallow layers ($\sim$0-1 m depth) in the 4 decades prior to talik onset (Fig. 2D)." The shallow drying does not appear to be shown on Figure 2D, only the total column soil moisture?

L281-2 "we find more pronounced tilting of the thawed layer with time and depth" This is not obvious to me from the plot. I might just suggest deleting this.

L290-2 "the rate of thawing and drainage in response to permafrost thaw may be underestimated in deeper CLM4.5 layers near bedrock due to reduced heat capacity." Sorry if I am missing something here but it doesn't seem to me like reduced heat capacity would reduce the rate of thawing, but rather than it would thaw more quickly because less heat is needed to thaw? Can you check this? Thanks.

L313-315 "however, our comparison to observations suggests that simulated thaw rates in this region and for similar permafrost temperatures are underestimated". This is not totally clear. Which comparison with obs? Are you referring to the comparison against thaw rates in Siberia which comes in the following section? Or are you inferring this from your comparison against borehole temperatures?

L335-336 "5 Siberian borehole sites which recorded at least 5 years of data spanning multiple decades:" Earlier you were talking about having 6 sites (or 9, or 3) and here it is 5. Please just clarify this a bit!

L337 "Records at these locations show a decrease in thaw volume" Do you mean an increase? On the next line it also refers to 'negative trends', which doesn't seem to fit with the plots/results (or the expectations!). Maybe these things should read 'increase in thaw volume' and 'positive trends'?

L345-346 "(layer thickness increases exponentially with depth along the Siberian transect)" This is not totally clear. Do you mean that active layer thickness increases expo-nentially with latitude. . .? Is that data shown somewhere? (It doesn't necessarily need to be, maybe just write 'data not shown' if it isn't)

L348-355. I'm not sure how much this is adding overall. It gets a bit confusing. Where you say "(vertical dashed line)", I would change to '(vertical dashed lines on Figure 5)', assuming this is what you're referring to? Anyway, it gets confusing when talking about groups of sites and it being hard to identify those groups. I would maybe condense these lines to something along the lines of "There is considerably spatial variability in thaw trends, for example site X is this far from site Y [relatively close] but with Z difference in trends [relatively large]. Talik formation occurs at several sites, at differ-ent times between 1957 and 1990 (shown by vertical dashed lines on Figure 5). We acknowledge the difficulty. . ."

L364-366 "The simulated trend in thaw volume shows a change in sign at northern lo-cations (blue), acceleration of thaw at southwest sites (orange), and reduction of thaw at the southeast sites (brown)". This sentence suggests that the thaw volume reduces at the southeast sites, I guess because the thaw volume in CLM is less than the obser-vations, but I would be inclined to interpret this instead as: the CLM simulation always had a too-small thaw volume, and there was never any 'decrease in thaw volume' in the simulation. But it is not possible to tell from the plot - Why did you not include the historical CLM simulation on the plot so it would overlap with the observed period? I also wouldn't say there is a "change of sign" at the northern locations. I guess you refer to the fact that the thaw volume is slightly decreasing at the northern sites historically, and increasing in the future? But as you say this is a very small trend so I would proba-bly instead interpret this as a relatively stable site that shifts to degradation towards the end of the century? I also find figure 6 a bit confusing with the symbols and what they represent. So apparently the circle represents 'thaw onset in January' but this happens considerably sooner in the simulation than 'thaw onset in March', which doesn't make sense to me? Thaw onset should get earlier each year? In the main text it implies this

is actually referring to deep thaw lasting throughout the winter (but this is not implied in the figure caption!). Maybe it would just be best not to include the symbols at all. This would make the plot simpler to interpret. Lines 369 "Our simulations show a shift to accelerated soil thaw beginning in the early 2080s". This sounds like you are referring to the whole simulation and not just this one particular point - can you make this more clear?

L382 "A total of 6.8 million km2 of land is projected to transition". This is not clear in the abstract which reads like it's only around 3 million km2.

L388 "followed by gradual decline to 0.5 Pg C by 2300". Does this suggest the temperature has stabilised and things are moving back towards equilibrium or is it more complicated than that? Could you comment? Including the supplementary plots of temperature trajectories that I suggested earlier might clear this one up.

L405-412. Here you are talking about NBP as positive, increasing, but in the plot (and according to your stated sign convention), a source is represented by negative NBP, decreasing. Please make this paragraph consistent.

L437 - Again wrong sign convention for NBP?

L437-438 "In general, C sources in these regions are more sensitive to C emissions from deep soil thaw" Have you actually quantified how much of the C is coming from deep soil...?

Figure 10: I think this could also be a supplementary figure or removed altogether, maybe giving slightly more detail on the numbers where it's mentioned in the text.

Figure 11 / Lines 453-461. I am struggling to interpret the upper plot on this Figure, and I am wondering whether this part adds much to the analysis. Since the analysis is already long I might suggest removing this paragraph and figure.

I suggest doing a bit more quantification of the contribution to total carbon sources. If there is a total of 11PgC emitted by 2300, what fraction of that comes from the three

different 'categories' of points in the trimodal distribution? I think it would be really useful to know which are the important carbon sources - or whether they are all similar. (I have made this comment again above)

L472 "GPP and total respiration show nearly linear increases ($\sim$15% per decade)" Minor point but an increase of 15% per decade would be exponential, not linear.

L485-487 "The trend in soil vs litter respiration explains almost the entire trend in net ecosystem C balance from neutral to net source (Fig. 12 G – H)." I'm not sure what you mean by this. I would have thought the trend in NBP is determined by the difference between GPP and total respiration, rather the difference between soil and litter respiration? And how do you draw these conclusions from the plots? Please could you make this clearer! Thanks.

L494 NF was only defined at the beginning of the paper and never used again until this point, which means that I had forgotten what it meant by now. You don't use this abbreviation much so I suggest you don't need it.

Discussion: L526-529 "Experiments demonstrating the sensitivity of talik to soil drying within the active layer across soil hydrology schemes in previous (CLM4), current (CLM4.5), and newly available (CLM5) versions of CLM could provide key insight on soil thermal dynamics in frozen or partially frozen conditions." This comes out of the blue a bit, and I am also not clear about why this would be useful? Comparing different hydrology schemes could be useful if one is more realistic than the others or includes different processes? But it is not clear that this would be the case in different CLM versions? It might be better to look at a purpose-built permafrost model, for example, or a model that resolves discontinuous permafrost.

L538-540 "thus appears to be driven by combination of warming and increased nitrogen availability resulting from permafrost thaw". I would suggest changing "appears to be" to "may be". . . you haven't looked at nitrogen here, and it is not totally clear/agreed that permafrost thaw will increase nitrogen availability.

L605 "Our main results emphasize an emergence of cold season processes" Not sure what you mean by 'an emergence of cold season processes'...? Rephrase?

L629-630 "Active-layer deepening leads to C sink-to-source transitions in some regions, talik-driven permafrost loss in others,..." This should be rephrased, "C sink-to-source transitions are caused by active layer deepening in some regions, talik-driven permafrost loss in others,..." Otherwise the sentence doesn't quite make sense, it sounds like active-layer deepening is causing all of the other things!

Hope you find these comments helpful. Best wishes!

---

## Referee Comment (RC2) · Anonymous Referee #2 · 13 Oct 2017

**Overall impression**

In the current study, Parazoo and colleagues have used the land surface model CLM4.5 to simulate permafrost state and changes from 2010 to 2300 under strong warming following the RCP8.5 scenario. They have investigated how permafrost degradation evolves in space and time, with a special focus on how talik formation affects thaw dynamics. Further, the authors have used their model experiment to analyse how future C fluxes in permafrost regions will evolve and when a carbon sink to source transition is likely to occur along different regions of the permafrost domain.

The presented analyses are helpful for increasing our process understanding of how individual factors explain inferred differences in simulated carbon fluxes between cold and warmer permafrost regions. E.g. the authors find that cold permafrost locations become C sources due to altered thaw-season dynamics while transitions of warm permafrost regions are mainly affected by changes in cold season dynamics. Further, the authors discuss how the presented results of this study can help finding an (optimal) design for monitoring the thermal and carbon state and changes in permafrost regions. The paper is well structured and written, and model analyses were performed elaborately. Adding new insights into permafrost degradation and carbon dynamics, this study can be considered of broad interest to the readership of The Cryosphere.

**General comments**

1) Initial SOC storages The initialized soil C stocks play a key role in affecting simulated future C release and the timing for a sink to source transition. In the study presented, no information is given how these C stocks were initialized for the simulation setting used here (besides referring to two previous CLM4.5 studies). Information should be provided in a sub-section on how SOC stocks were initialized, and on how these stocks compare to observed data (e.g. NCSCD - in terms of total storages and with regard to CLM4.5 inferred high (peaty) SOC storages at northern grid cells). As talik formation down to some meters are analysed in this study, I wonder how deep SOC is initialized in the soil column? Can e.g. soil thaw deeper than 3 meters further increase the pool of thawed carbon available for decomposition? A further key factor not discussed in the manuscript concerns assumptions about SOM lability made in the model. As especially uncertainty in slowly decomposing SOM is very high, I wonder how different settings of a humus timescale parameter (or different partitioning between labile and less labile pools) would affect inferred sink to source transition times? If feasible, this would be worth exploring by an extra sensitivity run, or at least by discussing qualitatively the impact of uncertainty in assumed SOM decomposition timescales on the findings discussed here.
**Talik formation**

The study focuses on talik formation as a key process which leads to abrupt permafrost degradation. The discussion (and the model simulation) is done for non-lake environments. Talik formation through thermokarst lake initialisation is not considered and mentioned. Yet this process is known to lead to rapid thaw through pronounced sublake talik formation, which can strongly affect carbon release from thawed sub-lake sediments. Although it is questionable to which extent future Arctic landscapes will be affected by thermokarst formation, this process should be discussed in the context of future permafrost degradation and permafrost carbon release.

**Vegetation distribution**

As discussed extensively in the presented paper, the Arctic land carbon balance is determined by changes in the net flux of vegetation carbon uptake and respiration losses. To what extent is CLM4.5 able capturing high latitude vegetation distribution/patterns? Some short discussion on simulated high latitude vegetation in CLM4.5 would be interesting to include.

**Cumulated C fluxes**

I guess you want to discuss C source numbers as PgC per year? (L387). Please specify in the text and in Fig.7B y-axis label. How does the cumulative C source from 2010-2300 of 11.6 PgC relate to shown C release rates in Fig. 7B? If I interpret numbers shown correctly, these suggest much larger cumulative release. Given published work on total C release from future permafrost degradation under RCP8.5 in other studies (suggesting much larger release), this number (total C release) should be discussed in the context of existing estimates.

**Specific comments**

Results presented in this study are inferred for the RCP8.5 scenario. This should be made more clear in the text (when discussing future evolution of permafrost and carbon
fluxes), and in some of the figure legends.

Please check for consistent/correct use of NBP sink/source definition (e.g. L410) Fig.8: Did you intend to put the dashed horizontal NBP threshold line at -25 gCm-2yr-1 - in accordance with your definition?

I wonder whether a discussion of a bi-modal distribution (Fig. 7D) seems more likely than a tri-modal distribution.

Maybe a shifting of some figures (e.g. Fig 10., Fig.11) to an appendix section would be good?

L338/339 you mean positive trends?

L 365 and following Given the small (statistically insignificant?) trend at Drughina, probably discussing "unchanged" conditions (instead of discussing a "change in sign") is more appropriate.

L 362 and following Please check colour specifications (in my printed version lines are yellow instead of orange, "blue" and "cyan" are used to refer to the same line, ...)

L494 define "NF" - or better avoid abbreviation

Fig.12 legend how were error bars inferred / what do they describe?

Spelling

L29 IS -> is

L458 form -> from

L470 = -> -

TCD

---

## Author Comment (AC1) · 20 Nov 2017

Reviewer 1

Review of "Detecting the permafrost carbon feedback: Talik formation and increased cold-season respiration as precursors to sink-to-source transitions"

The authors ran the Community Land Model (CLM) version 4.5 up to 2300, using RCP 8.5 forcing. They then perform an in depth analysis of permafrost-region dynamics in this simulation, including identifying key events: Talik formation (related the degradation

of permafrost) and sink-to-source transition, i.e. the point at which the land surface changes from a net sink of carbon from the atmosphere, to a net source. It is interesting to note that this behaviour (starting as a sink and transitioning to a source) is identified across a large fraction of the current permafrost zone. However, the total carbon source is apparently only 11.6 GtC by 2300, which is low compared with previous estimates.

The authors extensively analyse different variables such as thawed volume (a newly defined metric), active layer depth, primary production, respiration and fires, and how these influence talik formation and sink-to-source transition. They find three main drivers of sink-to-source transition: 1. Active layer thickening in cold, carbon-rich high Arctic permafrost 2. Talik formation leading to winter respiration in low Arctic, warmer soils 3. Fire driven carbon source in more productive regions which dry out, and a lot of vegetation is burned. They also showed some indicators of talik formation such as a rapid increase in thawed volume immediately preceding talik formation.

This is a very thorough analysis and a well written paper that will make a great publication in The Cryosphere, after some small revisions. In general I would like to see a bit more analysis about the size of the carbon sources, not just the timing of transition. This could comprise a bit of discussion of the cumulative carbon source (11.6 Gt), and the significance of this - compared to previous estimates, and the time trajectory of the cumulative source (i.e. when does the Arctic as a whole become a net source?). And then if they could break that down to say which of the different types of source (driven by AL, talik or fires) has the bigger contribution to the total source or if these are all comparable magnitude, that would be add some value to the paper. It's fine saying that we should monitor the high Arctic systems as they will become a source soonest, but if this source is likely to be very small, there would not be so much point?

Response:

We thank the reviewer for the astute observation of the small source and excellent recommendation to enhance our discussion. We found an error in our C source budget

calculation with the new total being 120 Pg C by 2300. We have corrected this error, and included more extensive analysis and discussion of cumulative carbon emissions and sources throughout the paper (see responses below)."

Reviewer: I also suggest considering the soil types at the boreholes. It would hopefully be possible to get an idea of this from a site description or by asking the PI. For example if these are peaty soils that might explain the very slow progression of freeze-thaw compared with CLM (also relatedly, the water content).

Response:

This is a good suggestion. It's difficult to pin down in CLM if mineral soil texture affects thaw rates compared to borehole observations due to the many possible explanation for differences in thaw rates: soil organic content, lateral water flow, surface slope and aspect, ground ice. However, we have looked into soil texture effects at the Alaskan boreholes and find that higher rates of observed soil thawing may be related to 2 factors: (1) relatively dry upper soil at the Gakona and Mould Bay sites, and (2) low surface organic layer and high conductivity of the Barrow2 and Mould Bay soils.

We revised the site descriptions in methods as follows (line 245):

"Mould Bay is a continuous permafrost tundra site with measurements at 63 depths from 0 - 3 m. Mould Bay has almost no organic layer (about 2 cm) and then sandy silt with high thermal conductivity. Barrow is a continuous permafrost tundra site with measurements at 35 depths from 0 - 15 m. The soil at Barrow is represented by silt with a bit of mix with some organics and almost no organic layer on top. Conductivity of the upper layer is $\sim$1 W mK-1 for unfrozen and $\geq$ 2 W mK-1 for frozen soil. Gakona is a continuous permafrost forest tundra site with measurements at 36 depths from 0 - 30 m. Gakona has a thick organic layer of moss (0 to 5 cm), dead moss (from 5 to 13 cm), and peat (from 13 to 50 cm), then silty clay at depth."

We offer an explanation for high observed thaw rates in Section 3.1 (line 359):

"Overall, we find that simulated patterns of permafrost thermal state change are consistent with available observations, but that the exact thaw rates are uncertain. Although there are many possible explanations for differences in observed and simulated thaw rates, we can attribute high observed thaw rates in part to a combination of (1) relatively dry upper soil at Gakona and Mould Bay, and (2) low surface organic layer and high conductivity of the Barrow and Mould Bay soils. We keep these uncertainties in mind as we examine patterns of change and talik formation simulated into 2300."

We offer some advice for future experiments in the discussion (line 586):

"Controlled experiments demonstrating the sensitivity of talik to parameters that control soil drying such ice impedance or baseflow scalars (e.g. Lawrence et al., 2015), and the effect of organic content and mineral soil texture (Lawrence and Slater, 2008), could provide key insight on soil thermal dynamics in frozen or partially frozen conditions."

We also include an additional column in Table 2 for soil characteristics indicating surface organic layer content and soil type:

Soil Features: Surface organic layer / Soil Type Mould Bay, Canada: Organic layer (∼2 cm)/ Sandy silt Barrow2, Alaska: Low organic layer / Sandy silt Gakona1, Alaska: Thick organic layer (50 cm) / Silty clay

Reviewer: Finally the paper is rather long and I would suggest reducing in length where possible. I have indicated a couple of points below.

Response:

We removed figures 10 and 11 as suggested by the reviewers, but added a new figure for cumulative carbon sources to address the primary reviewer concerns.

Line-by-line comments

Reviewer: Introduction: L62: 'Shifts in vegetation community' is mentioned in the introduction as being an important factor, but is this considered here? Are you running with

dynamic vegetation? If you are, this should mentioned and if not, this omission should be dis- cussed later on. L70: Same for soil organic matter export by rivers.

Response:

Since these processes are not considered in this analysis, we added a qualifier statement later on in the discussion (Line 590):

"Other factors affecting soil hydrology and carbon cycling not considered in our CLM4.5 simulations include high spatial resolution in discontinuous permafrost, shifts in vegetation community, lateral flow representation, thermokarst activity and other thaw-related changes to the ground surface, surface slope and aspect, soil heterogeneity, and potentially several other factors (see Jorgenson and Osterkamp (2005) for discussion of some of the many complexities to be considered)."

Reviewer: L86-87: Include more recent references for total permafrost car- bon quantity, such as Hugelius et al 2014, Biogeosciences (https://www.biogeosciences.net/11/6573/2014/bg-11-6573-2014.pdf), and there is also a new paper by Jackson et al coming out in November with revised estimates, this will be in Annual review of Ecology, Evolution and Systematics (http://www.annualreviews.org/doi/abs/10.1146/annurev-ecolsys-112414-054234).

Response:

References and carbon totals are updated as follows (Line 91):

"Talik as well as longer, deeper active layer thaw stimulate respiration of soil C (Romanovsky and Osterkamp, 2000; Lawrence et al., 2008), making the ∼1035 Pg soil organic carbon in near surface permafrost (0-3 m) and ∼350 Pg soil organic carbon in deep permafrost (> 3 m) vulnerable to decomposition (Hugelius et al., 2014; Jackson et al., 2017)."

Reviewer: Methods: L138-142 The standard RCP 8.5 only goes until 2100 so presumably some extension is used here? Could you mention what this looks like - for

example, does it stabilise at some point or does the global temperature and CO2 just keep increasing? I also think it would be useful for comparing with the permafrost thaw results, to see a plot of the global temperature across the three centuries of future simulation. I suggest adding this at least as a supplementary figure (as there are already a lot of figures in the main manuscript).

Response:

We used the ECP8.5 scenario for the period 2100-2300. We added a time series for air temperature in Figure 1A (shown below) and modified our methodology description as follows (Line 165):

"We use an anomaly forcing method to repeatedly force CLM4.5 with observed meteorological from the CRUNCEP dataset for the period 1996–2005 (data available at dods.ipsl.jussieu.fr/igcmg/IGCM/BC/OOL/OL/CRU-NCEP/) and monthly anomalies added based on a single ensemble member from a CCSM4 Representative Concentration Pathway 8.5 (RCP8.5) simulation for the years 2006-2100 and Extended Concentration Pathway 8.5 (ECP8.5) for the years 2100-2300. Land air temperature for the period 2006-2300, shown in Fig. 1A., is projected to increase steadily in our simulation, with a slight decrease in the rate of warming"

Reviewer: L152-154 "The C source transition represents a shift of ecosystem C balance from a neutral or weak C sink to a long-term source driven by onset of permafrost thaw and respiration of deep SOM" - here you suggest that the deep SOM alone is driving the transition, whereas your analysis suggests that it is driven by different things depending on region. Maybe you can qualify this sentence a bit?

Response:

Great point! We have qualified this sentence as follows (L185):

"The C source transition represents a shift of ecosystem C balance from a neutral or weak C sink to a long-term source as C balance shifts to increasing dominance of

C source processes including permafrost thaw and fires (Koven, Lawrence and Riley, 2015)."

Reviewer: L 201-204: "For comparison to projected trends in CLM4.5, we recalculate observed trends using the inter-site average from all 9 sites at 3 unique locations: northern Siberia (67°N, 144°E), southwest Siberia (61°N, 115°E), and southeast Siberia (59°N, 131°E)." This is not entirely clear. You were talking about using 6 sites and now it says 9, but then you end up with 3? Can you make it more clear? Did you combine sites into groups based on approximate locations. . .?

Response:

We clarified our analysis as follows (L231):

"To assess observed thaw trends from 1955-1990, we analyze individual sites which report at least 10 months yr-1 of reported monthly mean soil temperature at each layer, and 55 months across the 5 layers (out of 60 possible layer-months per year). Based on these requirements, we find that 6 of 9 sites yield at least 6 years of data over multiple decades, and are well suited for examining historical thaw trends. For comparison of observed trends to historical and projected trends from 1950-2300, we analyze clusters of sites by combining the 9 sites into 3 groups based on approximate locations, and calculate observed trends using the inter-site average at each location. We use 2 sites in northern Siberia (67°N, 144°E), 6 sites in southwest Siberia (61°N, 115°E), and 1 site in southeast Siberia (59°N, 131°E). Site information is shown in more detail in Table 1."

Reviewer: Results: L228 Do you mean 2300?

Response:

Yes, thank you for identifying this mistake!

Reviewer: L266 "Our simulations show a similar drying pattern in shallow layers (âĹij0-1 m depth) in the 4 decades prior to talik onset (Fig. 2D)." The shallow drying does not

appear to be shown on Figure 2D, only the total column soil moisture?

Response:

We added a line for shallow soil moisture (see attached revision of Fig 2), and revised the text as follows (L07):

"Our simulations show a similar, but very slight, drying pattern in shallow layers in the 4 decades prior to talik onset (1.3% loss of soil moisture over 0-1 m depth; Fig. 2D), accounting for about half of total water storage loss in the column. More significant changes in water balance occur following talik onset, including more rapid drying in shallow layers ($\sim$10% over 4 decades) and in the column ($\sim$16%), and a substantial increase in sub-surface drainage, as discussed below."

Reviewer: L281-2 "we find more pronounced tilting of the thawed layer with time and depth" This is not obvious to me from the plot. I might just suggest deleting this.

Response:

"Pronounced tilting" is a misleading description of the thaw pattern. We revised this description as follows (L323):

"In the 3 decades leading up to talik onset, we find gradual deepening of the thawed layer to 3-4 m and penetration of thaw period into Jan-Feb."

Reviewer: L290-2 "the rate of thawing and drainage in response to permafrost thaw may be under- estimated in deeper CLM4.5 layers near bedrock due to reduced heat capacity." Sorry if I am missing something here but it doesn't seem to me like reduced heat capacity would reduce the rate of thawing, but rather than it would thaw more quickly because less heat is needed to thaw? Can you check this? Thanks.

Response:

Yes, thanks for catching this inaccurate statement. We clarified as follows (L332).

"We note that bedrock soil is not hydrologically active in CLM4.5, and thus the rate of thawing and drainage in response to permafrost thaw may be overestimated in deeper CLM4.5 layers near bedrock due to reduced heat capacity."

Reviewer: L313-315 "however, our comparison to observations suggests that simulated thaw rates in this region and for similar permafrost temperatures are underestimated". This is not totally clear. Which comparison with obs? Are you referring to the comparison against thaw rates in Siberia which comes in the following section? Or are you inferring this from your comparison against borehole temperatures?

Response:

We clarified the discussion as follows (L355):

"Talik onset in CLM4.5 is variable in the region containing Gakona (southeast Alaska) with earliest onset by mid-century (∼2050s, Fig. 1A); however, our comparison to borehole temperature data at Gakona suggests that simulated thaw rates in southwest Alaska and across pan-Arctic regions with similar permafrost temperatures are underestimated, and that earliest onset may occur sooner than predicted."

Reviewer: L335-336 "5 Siberian borehole sites which recorded at least 5 years of data spanning multiple decades:" Earlier you were talking about having 6 sites (or 9, or 3) and here it is 5. Please just clarify this a bit!

Response:

Should be 6, thank you! We corrected as follows (L383):

"We focus first on site-specific long-term historical trends by analyzing the 6 Siberian borehole sites which recorded at least 5 years of temperature data spanning multiple decades: Drughina, Lensk, Macha, Oimyakon, Uchur, and Chaingda."

Reviewer: L337 "Records at these locations show a decrease in thaw volume" Do you mean an increase? On the next line it also refers to 'negative trends', which doesn't

seem to fit with the plots/results (or the expectations!). Maybe these things should read 'increase in thaw volume' and 'positive trends'?

Response:

Yes, this should read "increase in thaw volume" and "positive trend". We corrected as follows (385):

"Records at these locations show an increase in thaw volume with an average positive trend of 0.19 m months yr-1 from 1955 – 1990 (Table 1, Fig 5). All sites except Drughina show positive trends"

Reviewer: L345-346 "(layer thickness increases exponentially with depth along the Siberian tran- C4 sect)" This is not totally clear. Do you mean that active layer thickness increases expo- nentially with latitude. . .? Is that data shown somewhere? (It doesn't necessarily need to be, maybe just write 'data not shown' if it isn't)

Response:

It's not clear why we added this statement, and it doesn't appear to affect the analysis, so we removed it and modified the text as follows (L390):

"Further examination indicates that active layer thickness at Drughina actually de- creased to 0.8 meters from 1989-1990 compared to 1.2 meters in the 1970s (data not shown). Drughina also shows smaller average thaw volume magnitude compared to other sites, consistent with shallower thaw. Together, these findings indicate that active layer thickness is decreasing at Drughina."

Reviewer: L348-355. I'm not sure how much this is adding overall. It gets a bit confus- ing. Where you say "(vertical dashed line)", I would change to '(vertical dashed lines on Figure 5)', assuming this is what you're referring to? Anyway, it gets confusing when talking about groups of sites and it being hard to identify those groups. I would maybe condense these lines to something along the lines of "There is considerably spatial variability in thaw trends, for example site X is this far from site Y [relatively close] but

with Z difference in trends [relatively large]. Talik formation occurs at several sites, at differ- ent times between 1957 and 1990 (shown by vertical dashed lines on Figure 5). We acknowledge the difficulty. . ."

Response:

This section sets up empirical evidence for increasing permafrost degradation from west to east in Siberia. We therefore clarify and condensed this section as follows (L395):

"There is considerable spatial variability in thaw volume and trends, but in general thaw trends increase from west (0.18 m months yr-1) to east (0.51 m months yr-1). Talik forms at several sites, at different times between 1957 and 1990 (shown by vertical dashed lines on Fig. 5), with earlier talik to the west consistent with higher mean initial thaw volumes."

Reviewer: L364-366 "The simulated trend in thaw volume shows a change in sign at northern locations (blue), acceleration of thaw at southwest sites (orange), and reduc- tion of thaw at the southeast sites (brown)". This sentence suggests that the thaw volume reduces at the southeast sites, I guess because the thaw volume in CLM is less than the observations, but I would be inclined to interpret this instead as: the CLM simulation always had a too-small thaw volume, and there was never any 'decrease in thaw volume' in the simulation. But it is not possible to tell from the plot - Why did you not include the historical CLM simulation on the plot so it would overlap with the ob- served period? I also wouldn't say there is a "change of sign" at the northern locations. I guess you refer to the fact that the thaw volume is slightly decreasing at the north- ern sites historically, and increasing in the future? But as you say this is a very small trend so I would probably instead interpret this as a relatively stable site that shifts to degradation towards the end of the century? I also find figure 6 a bit confusing with the symbols and what they represent. So apparently the circle represents 'thaw onset in January' but this happens considerably sooner in the simulation than 'thaw onset in

March', which doesn't make sense to me? Thaw onset should get earlier each year? In the main text it implies this is actually referring to deep thaw lasting throughout the winter (but this is not implied in the figure caption!). Maybe it would just be best not to include the symbols at all. This would make the plot simpler to interpret. Lines 369 "Our simulations show a shift to accelerated soil thaw beginning in the early 2080s". This sounds like you are referring to the whole simulation and not just this one particular point - can you make this more clear?

Response:

We apologize for the confusion this section created. We added historical thaw simulations from CLM4.5 so the simulated record is continuous from 1950-2300. We also removed the thaw onset symbols, which were meant to represent the progression of thaw later into the cold season (from early winter (Oct-Dec) to deep winter (Jan-Apr)) prior to talik onset. The revised figure 6 is attached, The revised text is shown below (405):

"We recompute observed thaw trends at regional clusters using combined records at the 2 sites in northern Siberia (blue), 6 sites in southwest Siberia (orange), and 1 site in southeast Siberia (brown, Table 1) and compare to historical and projected thaw volume trends in CLM4.5 (Fig. 6). Northern locations show a consistent pattern of low thaw volume (< 10 m month yr-1) and negligible thaw trend ($\sim$0 m month yr-1) in the historical simulations and observed record from 1950-2000. Thaw projections in northern Siberia indicate continued stability of permafrost through the early 22st century, followed by a shift to accelerated soil thaw in the early 2120, marked by onset of deep soil thaw late in the cold season."

"Southern locations show a systematic underestimate of mean thaw volume (< 20 m month yr-1) compared to observations ($\sim$40 m month yr-1) from 1950-2000. Simulated thaw trends are negligible prior to 2000, but these likely represent an underestimate given low simulated thaw volumes and significant positive observed trends in

both southeast and southwest Siberia beginning in the 1960s following talik onset (Fig. 5). Thaw projections show more abrupt shifts in thaw volume in the early 21st century in the southwest (∼2025) and in the mid 21st century (∼2050) in the southeast. The strong discrepancy between observed and simulated thaw and talik onset in southern Siberia warrants close monitoring and continued investigation of this region through sustained borehole measurements and additional model realizations of potential future warming."

Reviewer: L382 "A total of 6.8 million km2 of land is projected to transition". This is not clear in the abstract which reads like it's only around 3 million km2.

Response:

6.8 million km2 refers to all NHL regions, within and outside the permafrost zone. This is clarified in Section 3.3 (L425).

"A total of 6.8 million km2 of land is projected to transition, peaking in the late 21st century, with most regions transitioning prior to 2150 (4.8 million km2 or 70%, Fig. 7B, solid black). C source transitions which occur in the permafrost zone, accounting for 6.2 million km2 of land (91% of all C source transitions), also form talik at some time from 2006-2300 (Fig. 7C). The remaining C source transitions (0.6 million km2, or 9%) occur outside the permafrost zone, primarily in eastern Europe."

We also rephrase the abstract to be more consistent with this section (L29):

"Widespread talik at depth is projected across most of the NHL permafrost region (14 million km2) by 2300, 6.2 million km2 of which is projected to become a long term C source, emitting 10 Pg C by 2100, 50 Pg C by 2200, and 120 Pg C by 2300, with few signs of slowing. Roughly half of the projected C source region occurs in predominantly warm sub-Arctic permafrost following talik onset. This region emits only 20 Pg C by 2300, but the CLM4.5 estimate may be biased low by not accounting for deep C in yedoma. Accelerated decomposition of deep soil C following talik onset shifts the

ecosystem C balance away from surface dominant processes (photosynthesis and lit-
ter respiration), but sink-to-source transition dates are delayed by 20-200 years by high
ecosystem productivity, such that talik peaks early (∼2050s, borehole data suggests
sooner) and C source transition peaks late (∼2150-2200). The remaining C source
region is in cold northern Arctic permafrost, which shifts to a net source early (late 21st
century), emits 5 times more C (95 Pg C) by 2300, and prior to talik formation due to
the high decomposition rates of shallow, young C in organic rich soils coupled with low
productivity."

Reviewer: L388 "followed by gradual decline to 0.5 Pg C by 2300". Does this suggest
the temperature has stabilised and things are moving back towards equilibrium or is it
more complicated than that? Could you comment? Including the supplementary plots
of temperature trajectories that I suggested earlier might clear this one up.

Response:

Temperatures continue to rise based on ECP8.5 which suggests it is more complicated.
We clarify this section, including more detailed analysis of C source magnitudes and
categories, as follows (L431):

"Net C emissions from C source transition regions are a substantial fraction of the
total NHL C budget over the next 3 centuries (Fig. 8). The cumulative pan-Arctic C
source increases slowly over the 21st century, reaching 10 Pg C by 2100 with RCP8.5
warming, then increases more rapidly to 70 Pg C by 2200 and 120 Pg by 2300 with
sustained ECP8.5 warming (Fig. 8, solid black). This pan-Arctic source represents
86% of cumulative emissions in 2300 from the larger NHL talik region (crosses), despite
the 2 fold smaller land area, and exceeds the talik region through 2200 due to mitigating
widespread vegetation C gains (Koven et al., 2015). Cumulative emissions over all NHL
land regions (diamonds, > 55N) increase in similar fashion to the talik region, reaching
120 Pg C by 2200 and 220 by 2300, with no sign of slowing."

Reviewer: L405-412. Here you are talking about NBP as positive, increasing, but in

the plot (and according to your stated sign convention), a source is represented by negative NBP, decreasing. Please make this paragraph consistent.

Response:

Corrected as follows (L458):

"In these regions, thaw volume is low (< 50 m months yr-1) and shows a weak relationship to NBP (NBP decreases much faster than thaw volume) prior to C source onset (indicated by large green circle in Fig. 8A)."

Reviewer: L437 - Again wrong sign convention for NBP?

Response:

Corrected as follows (489):

"and talik formation occurs when these regions are weak sinks (NBP > 0 g C m-2 yr-1)."

We have also revised Figure 8 (now Figure 9, attached) with arrows indicating C source or sink for clarification:

Reviewer: L437-438 "In general, C sources in these regions are more sensitive to C emissions from deep soil thaw" Have you actually quantified how much of the C is coming from deep soil. . .?

Response:

Unfortunately, we can't quantify C from deep soils since vertical resolved C flux output is not available. We have revised the statement to reflect an inferred contribution from deep soils (L490):

"In general, C source onset under high thaw volume indicates these regions are more sensitive to C emissions from deep soil thaw."

Reviewer: Figure 10: I think this could also be a supplementary figure or removed altogether, maybe giving slightly more detail on the numbers where it's mentioned in

the text.

Response:

We have removed Figure 10 and revised the text as follows (L470):

"A broader analysis of soil thaw statistics over all regions and periods indicates that most C source transitions ($\sim$2.3 million km2, or 77% of land where C source leads talik) occur at active layer depths below 3 m and thaw season penetration into November."

Reviewer: Figure 11 / Lines 453-461. I am struggling to interpret the upper plot on this Figure, and I am wondering whether this part adds much to the analysis. Since the analysis is already long I might suggest removing this paragraph and figure.

Response:

Agreed. The main point is the difference in GPP between warm and cold permafrost regions, which is discussed in previous sections. We have removed this paragraph and figure.

Reviewer: I suggest doing a bit more quantification of the contribution to total carbon sources. If there is a total of 11PgC emitted by 2300, what fraction of that comes from the three different 'categories' of points in the trimodal distribution? I think it would be really useful to know which are the important carbon sources - or whether they are all similar. (I have made this comment again above)

Response:

We have added a new figure (Figure 8, attached) quantifying the different carbon sources (below). We also found that our cumulative C emission estimate was off by a factor of 10 (didn't convert from year to decade), bringing our new C emission to 120 PgC by 2300.

We also revised the text as follows:

Abstract, to include total and regional contributions:

"Widespread talik at depth is projected across most of the NHL permafrost region (14 million km2) by 2300, 6.2 million km2 of which is projected to become a long term C source, emitting 10 Pg C by 2100, 50 Pg C by 2200, and 120 Pg C by 2300, with few signs of slowing. Roughly half of the projected C source region occurs in predominantly warm sub-Arctic permafrost following talik onset. This region emits only 20 Pg C by 2300, but the CLM4.5 estimate may be biased low by not accounting for deep C in yedoma. Accelerated decomposition of deep soil C following talik onset shifts the ecosystem C balance away from surface dominant processes (photosynthesis and litter respiration), but sink-to-source transition dates are delayed by 20-200 years by high ecosystem productivity, such that talik peaks early (∼2050s, borehole data suggests sooner) and C source transition peaks late (∼2150-2200). The remaining C source region is in cold northern Arctic permafrost, which shifts to a net source early (late 21st century), emitting 80 Pg C by 2300, and prior to talik formation due to the high decomposition rates of shallow, young C in organic rich soils coupled with low productivity."

The opening of section 3.3 to include total emissions and different categories:

"Fig. 7A plots the decade in which NHL ecosystems are projected to transition to long-term C sources over the next 3 centuries (2010-2300). A total of 6.8 million km2 of land is projected to transition, peaking in the late 21st century, with most regions transitioning prior to 2150 (4.8 million km2 or 70%, Fig. 7B, solid black). C source transitions which occur in the permafrost zone, accounting for 6.2 million km2 of land (91% of all C source transitions), also form talik at some time from 2006-2300 (Fig. 7C). The remaining C source transitions (0.6 million km2, or 9%) occur outside the permafrost zone, primarily in eastern Europe."

"Net C emissions from C source transition regions are a substantial fraction of the total NHL C budget over the next 3 centuries (Fig. 8). The cumulative pan-Arctic C source increases slowly over the 21st century, reaching 10 Pg C by 2100 with RCP8.5

warming, then increases more rapidly to 70 Pg C by 2200 and 120 Pg by 2300 with sustained ECP8.5 warming (Fig. 8, solid black). This pan-Arctic source represents 86% of cumulative emissions in 2300 from the larger NHL talik region (crosses), despite the 2 fold smaller land area, and exceeds the talik region through 2200 due to mitigating widespread vegetation C gains (Koven et al., 2015). Cumulative emissions over all NHL land regions (diamonds, > 55N) increase in similar fashion to the talik region, reaching 120 Pg C by 2200 and 220 by 2300, with no sign of slowing."

"The geographic pattern of C sink-to-source transition date is reversed compared to that of talik formation, with earlier transitions at higher latitudes (the processes driving these patterns are discussed in detail below). Overall, the lag relationship between talik onset and C source transition exhibits a tri-modal distribution (Fig. 7D), with peaks at negative time lag (C source leads talik onset, Median Lag = -5 to -6 decades), neutral time lag (C source synchronized with talik onset; Median Lag = -2 to 1 decade), and positive time lag (C source lags talik; Median Lag = 12 decades; red shading in Fig. 7C). Roughly half of these regions (3.2 million km2) show neutral or positive time lag (lag $\geq$ 0). This pattern, characteristic of the sub-Arctic (< 65°N), represents the vast majority of C source transitions after 2150 (Fig. 7B, dotted), but only accounts for 17% of cumulative emissions (20 Pg C by 2300, Fig. 8, dotted). The remaining regions (3.0 million km2) in the Arctic and high Arctic (> 65°N) show negative time lag and account for most of late 21st century sources (Fig. 7B, dashed) and cumulative emissions (95 Pg C by 2300, or 79%; Fig. 8, dashed). C sources in regions not identified as talik (0.63 million km2) either show talik presence at the start of our simulation, or are projected to transition in the absence of permafrost or in regions of severely degraded permafrost (Fig. 7C, dash dotted). This region contributes only 5 Pg C (4%) of cumulative C emissions in 2300."

The following statements on line 479 referring to high SOM emissions:

"The total area of land in which SOM exceeds 100 kg C m-2 represents 2/3 of all land where C sources lead talik onset (2.0 million km2), and peaks at a negative time lag of

[Figure]

-5 to -6 decades (Fig. 7D, green bars), which perfectly aligns with the peak distribution of negative time lags. Cumulative C emissions from regions of SOM > 100 kg C m-2 are also 2/3 of total C emissions (80 Pg C; Fig 8, green)."

Line 495 for fire emissions:

"The regions where fire C emissions exceed 25 g C m-2 yr-1, representing our threshold for C source transition, are exclusively boreal ecosystems, account for 1/3 of all land with negative lags ($\sim$1.1 million km2), and align perfectly with the peak distribution of positive time lags (Fig. 7D, red bars) and cumulative C emissions (20 Pg C in 2300, Fig. 8, red)."

Lines 608 and 635 in the discussion:

"About half of this region (3.2 million km2) shows a pattern of accelerated soil C respiration following talik onset, which shifts the surface C balance of photosynthetic uptake and litter respiration from net C sinks to long term net sources totaling 20 Pg C across 3.2 million km2 of NHL land by 2300."

"We identify an equally large region of land in the high Arctic, representing $\sim$3.0 million km2, which is projected to transition to a long term C source much sooner than the sub-Arctic in the absence of talik, and emit 5 times as much carbon by 2300 ($\sim$95 Pg C)."

And line 688 in the conclusions

Reviewer: "6.8 million km2 of land impacted in Siberia and North America will produce an integrated C source of 90 Pg C by 2100 and 120 Pg C by 2200." L472 "GPP and total respiration show nearly linear increases ($\sim$15% per decade)" Minor point but an increase of 15% per decade would be exponential, not linear.

Response:

Revised as follows (L527):

"GPP and combined respiration increase by ∼15% per decade for each permafrost regime surrounding the decade of C source transition with peak fluxes in the growing season (Fig. 11 A - D)."

Reviewer: L485-487 "The trend in soil vs litter respiration explains almost the entire trend in net ecosystem C balance from neutral to net source (Fig. 12 G – H)." I'm not sure what you mean by this. I would have thought the trend in NBP is determined by the difference between GPP and total respiration, rather the difference between soil and litter respiration? And how do you draw these conclusions from the plots? Please could you make this clearer! Thanks.

Response:

We have revised as follows (L541):

"The trend in the respiration difference in warm and cold permafrost, which increase by similar amounts (∼100 g C m-2 yr-1), thus reflects an increasing dominance of respiration from younger and older soil C pools, respectively. These trends are identical to the corresponding NBP trends, which decrease by 100 g C m-2 yr-1 over the same period from neutral to net source (Fig. 12 G – H), such that the differences between GPP and respiration driving the NBP trends are explained almost entirely by the increasing fraction of soil vs litter respiration."

Reviewer: L494 NF was only defined at the beginning of the paper and never used again until this point, which means that I had forgotten what it meant by now. You don't use this abbreviation much so I suggest you don't need it.

Response:

NF has been changed to non-frozen.

Reviewer: Discussion: L526-529 "Experiments demonstrating the sensitivity of talik to soil drying within the active layer across soil hydrology schemes in previous (CLM4), current (CLM4.5), and newly available (CLM5) versions of CLM could provide key insight on soil thermal dynamics in frozen or partially frozen conditions." This comes out of the blue a bit, and I am also not clear about why this would be useful? Comparing different hydrology schemes could be useful if one is more realistic than the others or includes different processes? But it is not clear that this would be the case in different CLM versions? It might be better to look at a purpose-built permafrost model, for example, or a model that resolves discontinuous permafrost.

Response:

We agree it's not clear how exactly this would be useful since the model differences in soil hydrology and snow are not systematically differences. We don't think a purpose-built permafrost model is likely to be any better since the processes represented are typically the same as in CLM. We agree that resolving discontinuous permafrost would be good, but it's unclear how to do that other than with increased resolution, but this would miss other important processes such as lateral flow. Something more controlled like a parameter sensitivity study with variable ice impedance or baseflow scalar, such as examined in Lawrence et al., 2015, is likely to provide the best insight. We therefore replace our comment with a more general comment about uncertainties, and suggest more controlled experiments similar to Lawrence (L586):

"Controlled experiments demonstrating the sensitivity of talik to parameters that control soil drying such ice impedance or baseflow scalars (e.g. Lawrence et al., 2015), and the effect of organic content and mineral soil texture (Lawrence and Slater, 2008), could provide key insight on soil thermal dynamics in frozen or partially frozen conditions. Other factors affecting soil hydrology and carbon cycling not considered in our CLM4.5 simulations include high spatial resolution in discontinuous permafrost, shifts in vegetation community, lateral flow representation, thermokarst activity and other thaw-related changes to the ground surface, surface slope and aspect, soil heterogeneity, and potentially several other factors (see Jorgenson and Osterkamp (2005) for discussion of some of the many complexities to be considered)."

Reviewer: L538-540 "thus appears to be driven by combination of warming and increased nitrogen availability resulting from permafrost thaw". I would suggest changing "appears to be" to "may be". . . you haven't looked at nitrogen here, and it is not totally clear/agreed that permafrost thaw will increase nitrogen availability.

Response:

We have revised as suggested

Reviewer: L605 "Our main results emphasize an emergence of cold season processes" Not sure what you mean by 'an emergence of cold season processes'. . .? Rephrase?

Response:

Rephrased as follows (L692):

"Our main results emphasize an increasingly important impact of NHL cold season warming on earlier spring thaw, longer non-frozen seasons, and increased depth and seasonal duration of soil thaw."

Reviewer: L629-630 "Active-layer deepening leads to C sink-to-source transitions in some re- gions, talik-driven permafrost loss in others,. . ." This should be rephrased,

Response:

Reviewer: "C sink-to- source transitions are caused by active layer deepening in some regions, talik-driven permafrost loss in others,..." Otherwise the sentence doesn't quite make sense, it sounds like active-layer deepening is causing all of the other things!

Rephrased as suggested.

Reviewer: Hope you find these comments helpful. Best wishes!

Response:

Thank you, much improved now!

[Figure]

Fig. 1. Figure 1. Decade of projected talik formation and correlation to initial state of simulated permafrost temperature and observed permafrost extent. (A) Time series and (B) map of the simulated decade o

[Figure]

**Fig. 2.** Figure 2. Patterns showing the progression of soil thaw in the decades surrounding talik onset. Individual lines represent averages across the subset of talik forming regions for each decade from the

**Observed vs Model Thaw: Trans Siberian Transect**

*Figure: plot of Thaw Volume (m months yr$^{-1}$) versus Year (1950–2200). Legend entries include:*

CLM4.5 Historical: 1950-2005
[68.3N 145E]
thaw rate = 0.0048 m month yr-1

CLM4.5 Future: 2006-2300

Borehole: Drughina + Ustmoma
~[67N, 144E]
thaw rate = -0.057 m month yr-1

CLM4.5 Historical: 1950-2005
[60.7N 115E]
thaw rate = 0.079 m month yr-1

CLM4.5 Future: 2006-2300

Borehole: Chumpuruck + Lensk +
Macha + Oimyakon +
Tongulukh + Uchar
~[61N, 115E]
thaw rate = 0.019 m month yr-1

CLM4.5 Historical: 1950-2005
[59N 131E]
thaw rate = -0.013 m month yr-1

CLM4.5 Future: 2006-2300

Borehole: Chaingda
~[59N, 131E]
thaw rate = 0.51 m month yr-1

**Fig. 3.** Figure 6. Comparison of observed soil thaw to historical and future simulations at sites along the East Siberian Transect (crosses in Fig. 1). Observed thaw (filled circles) from 1955-1990 is based on

[Figure]

Cumulative Net Biome Production (Pg C) versus Simulation Year

Legend:
- NHL Land: > 55°N (28 Million km²)
- NHL Land: Talik Region (14.5 Million km²)
- C Source Transition: All (6.8 Million km²)
- C Source Transition: C Source Lags Talik (3.0 Million km²)
- C Source Transition: C Source Leads Talik (3.2 Million km²)
- C Source Transition: C Source Without Talik (0.6 Million km²)
- C Source Transition: SOM > 100 kg C m⁻² (2.0 Million km²)
- C Source Transition: Fire > 25 g C m⁻² y⁻¹ (1.7 Million km²)
- C Source Transition: Fire < 25 g C m⁻² y⁻¹ & SOM < 100 kg C m⁻² (2.8 Million km²)

**Fig. 4.** Figure 8. Cumulative net biome production (NBP) over northern high latitude (NHL) regions (> 55°N) from 2010 to 2300, where NBP < 0 is a net source. NHL regions are divided into the following categori

[Figure]

**Fig. 5.** Figure 9. Net biome production (NBP) as a function of thaw volume. Symbols represent NBP and thaw volume values averaged over regions which transition to long term C source from 2060-2140, binned into

---

## Author Comment (AC2) · 20 Nov 2017

Reviewer 2

Overall impression

In the current study, Parazoo and colleagues have used the land surface model CLM4.5 to simulate permafrost state and changes from 2010 to 2300 under strong warming following the RCP8.5 scenario. They have investigated how permafrost degradation evolves in space and time, with a special focus on how talik formation affects thaw

dynamics. Further, the authors have used their model experiment to analyse how future C fluxes in permafrost regions will evolve and when a carbon sink to source transition is likely to occur along different regions of the permafrost domain.

The presented analyses are helpful for increasing our process understanding of how individual factors explain inferred differences in simulated carbon fluxes between cold and warmer permafrost regions. E.g. the authors find that cold permafrost locations become C sources due to altered thaw-season dynamics while transitions of warm permafrost regions are mainly affected by changes in cold season dynamics. Further, the authors discuss how the presented results of this study can help finding an (optimal) design for monitoring the thermal and carbon state and changes in permafrost regions. The paper is well structured and written, and model analyses were performed elaborately. Adding new insights into permafrost degradation and carbon dynamics, this study can be considered of broad interest to the readership of The Cryosphere.

General comments

1) Initial SOC storages The initialized soil C stocks play a key role in affecting simulated future C release and the timing for a sink to source transition. In the study presented, no information is given how these C stocks were initialized for the simulation setting used here (besides referring to two previous CLM4.5 studies). Information should be provided in a sub-section on how SOC stocks were initialized, and on how these stocks compare to observed data (e.g. NCSCD - in terms of total storages and with regard to CLM4.5 inferred high (peaty) SOC storages at northern grid cells). As talik formation down to some meters are analysed in this study, I wonder how deep SOC is initialized in the soil column? Can e.g. soil thaw deeper than 3 meters further increase the pool of thawed carbon available for decomposition? A further key factor not discussed in the manuscript concerns assumptions about SOM lability made in the model. As especially uncertainty in slowly decomposing SOM is very high, I wonder how different settings of a humus timescale parameter (or different partitioning between labile and less labile pools) would affect inferred sink to source transition times? If feasible, this

would be worth exploring by an extra sensitivity run, or at least by discussing quali-
tatively the impact of uncertainty in assumed SOM decomposition timescales on the
findings discussed here.

Response:

We thank the reviewer for the recommendation to include the very important discussion
of soil C stocks and decomposition controls in CLM. Deep soil decomposition experi-
ments in CLM have been run in previous studies by the co-authors, to decouple chang-
ing dynamics of surface soils from deep soils, to determine the potential contribution
from permafrost layers, and to calibrate initial soil C stocks against observations.

We have elaborated on these details in the methods (L139) as follows:

"CLM is spunup to C equilibrium for the year 1850 by repeatedly cycling through 20
years of pre-industrial climate forcing with $CO_2$ and N-deposition set at 1850 levels.
C initialization is achieved via slow mixing by cryoturbation between the seasonally
thawed active layers and deeper permafrost layers (Koven et al., 2009). Including
vertically resolved processes leads to a sign change in the projected high-latitude C
response to warming, from net C gains driven by increased vegetation productivity to
net C losses from enhanced SOM decomposition (Koven et al., 2011). The soil grid
includes 30 vertical levels that has a high-resolution exponential grid in the interval 0–
0.5 m and fixed 20-cm layer thickness in the range of 0.5–3.5 m to maintain resolution
through the base of the active layer and upper permafrost, and reverts to exponentially
increasing layer thickness in the range 3.5–45 m to allow for large thermal inertia at
depth. Soil C turnover in CLM4.5 is based on a vertical discretization of first-order
multipool SOM dynamics (Koven et al., 2013; Oleson et al., 2013) where decomposi-
tion rates as a function of soil depth are controlled by a parameter $Z_\tau$ (Koven et al.,
2015; Lawrence et al., 2015). This depth control of decomposition represents the net
impacts of unresolved depth dependent processes. In this study, we utilize $Z_\tau$=10 m,
which yields a weak additional depth dependence of decomposition beyond the environmental controls and, as discussed and evaluated relative to $Z\tau$=1m and $Z\tau$=0.5m in Koven et al (2015), results in CLM permafrost-domain soil C stocks that are in closest agreement (1582 Pg for $Z\tau$=0.5 m, 1331 Pg for $Z\tau$=1m, and 1032 Pg for $Z\tau$=10m) with observed estimates (1060 Pg C to 3 m depth; Hugelius et al 2013). This reduction in initial C is due to higher decomposition rates at depth during the model initialization period. There is no C below 3.5 m, so additional thaw below 3 meters has a small impact on the C cycle. We note that the relationship applied in CLM4.5, which implies multiplicative impacts of limitations to decomposition, is commonly applied in land biogeochemical models, but is quite uncertain."

We also discuss implications of 3.5 m C limit in discussion (Line 626):

"Our estimate of C emissions following talik onset ($\sim$20 Pg C) is low compared to the cumulative emissions from all long term C source transitions (120 Pg C), but likely strongly underestimated. Soil C is not permitted below 3.5 m in CLM4.5, or in most analogous models, such that potential decomposition of the $\sim$350 Pg soil organic C in deep permafrost (> 3 m) is not accounted for (Hugelius et al., 2014; Jackson et al., 2017). This is significant for our simulations, which show frequent talik formation and accelerating thaw volumes below 3 m (e.g., Fig. 3). We therefore caution the reader in the interpretation of the timing and magnitude of permafrost C emissions following talik onset in our simulations, which represent a lower bound of potential emissions based on the current formulation of CLM4.5."

Reviewer:

Talik formation

The study focuses on talik formation as a key process which leads to abrupt permafrost degradation. The discussion (and the model simulation) is done for non-lake environments. Talik formation through thermokarst lake initialisation is not considered and mentioned. Yet this process is known to lead to rapid thaw through pronounced sublake talik formation, which can strongly affect carbon release from thawed sub-lake

sediments. Although it is questionable to which extent future Arctic landscapes will be affected by thermokarst formation, this process should be discussed in the context of future permafrost degradation and permafrost carbon release.

Response:

We agree that thermokarst is an important process which is not considered in our simulations. We added the following discussion of thermokarst effects to the intro (Line 87)

"More abrupt processes such as thermokarst lake initialization can also lead to rapid thaw through pronounced sub-lake talik formation (Jorgenson and Osterkamp, 2005). These processes can initiate formation of a talik zone (perennially thawed sub-surface soils) during active layer adjustment to new thermal regimes (Jorgenson et al., 2010) in lake and non-lake environments."

And methods (Line 135)

"More abrupt thaw processes affecting permafrost C dynamics and talik formation such as the effect of thermokarst or other thaw related landscape dynamics changes in wetland or lake distribution are not accounted for ( see Riley et al (2011) for more discussion)."

and discussion (Line 590):

"Other factors that are necessary for more accurate simulation of soil hydrology and carbon cycling but not considered in CLM4.5 and analogous models include: enhanced spatial resolution in complex topography and in discontinuous permafrost, sophisticated vegetation community definitions including trait based representation, shifts in vegetation community, lateral flow representation, thermokarst activity and other thaw-related changes to the ground surface, surface slope and aspect, soil heterogeneity, and potentially several other factors (see Jorgenson and Osterkamp (2005) for discussion of some of the many complexities to be considered)."

Reviewer:

Vegetation distribution: As discussed extensively in the presented paper, the Arctic land carbon balance is determined by changes in the net flux of vegetation carbon uptake and respiration losses. To what extent is CLM4.5 able capturing high latitude vegetation distribution/patterns? Some short discussion on simulated high latitude vegetation in CLM4.5 would be interesting to include.

Response:

The use of static vegetation distributions and simple vegetation community definitions is a major source of uncertainty in CLM4.5. We have acknowledged this model weakness in the response above.

Reviewer:

Cumulated C fluxes: I guess you want to discuss C source numbers as PgC per year? (L387). Please specify in the text and in Fig.7B y-axis label. How does the cumulative C source from 2010- 2300 of 11.6 PgC relate to shown C release rates in Fig. 7B? If I interpret numbers shown correctly, these suggest much larger cumulative release. Given published work on total C release from future permafrost degradation under RCP8.5 in other studies (suggesting much larger release), this number (total C release) should be discussed in the context of existing estimates.

Response:

We thank the reviewer for correctly pointing out our y-axis error and suggesting to compare our cumulative releases to existing estimates. We corrected the y-axis error by moving the C source numbers from Fig. 7B to Fig. 8 for more in depth analysis (Figures 7 and 8 attached). Our cumulated emission is 120 Pg C which is more in line with existing estimates. We summarize in the discussion as follows (Line 596):

"Our simulations show increasing C emissions over time across the talik region (Fig. 1B), as cumulative NBP becomes increasingly negative (NBP < 0 equals a net C

[Figure]

source), reaching a net source of 140 Pg C by 2300 (Fig. 8, crosses), consistent with previous estimates of net C balance across the larger pan-Arctic region from CLM4.5 (∼160 Pg C, Koven et al., 2015; Lawrence et al., 2015). Ecosystems which transition from net C sinks to net C sources represent less than half the total talik area (6.8 of 14.5 Million km2, Fig. 7A), but account for most (∼85%) of the cumulative emissions, reaching 10 Pg C in 2100, 70 Pg C in 2200, and 120 Pg C by 2300 (Fig. 8, solid black). Removing the effect of vegetation C gain (∼20 Pg C in 2100 and 40 Pg C in 2200 and 2300 according Koven et al., 2015), we estimate a cumulative permafrost emission for C source transition regions of 30 Pg C in 2100, 110 Pg C in 2200, and 160 Pg C in 2300. These numbers are on the low end but consistent with estimates of permafrost C emissions summarized by Schuur et al. (2015), which range from 37-174 Pg C by 2100 and 100-400 Pg C by 2300."

Reviewer:

Specific comments

Results presented in this study are inferred for the RCP8.5 scenario. This should be made more clear in the text (when discussing future evolution of permafrost and carbon fluxes), and in some of the figure legends.

Response:

We have added RCP8.5 and ECP8.5 information throughout the text:

L263 Section 3.1:

"Our simulations show widespread talik formation throughout Siberia and northern North America over the period 2010-2300 (Fig. 1B), impacting ∼14.5 million km2 of land in NHLs (55°-80°N) assuming a freeze/thaw threshold of -0.5°C and RCP8.5 and ECP8.5 warming scenarios."

L432 Section 3.3:

"The cumulative pan-Arctic C source increases slowly over the 21st century, reaching 10 Pg C by 2100 with RCP8.5 warming, then increases more rapidly to 70 Pg C by 2200 and 120 Pg by 2300 with sustained ECP8.5 warming (Fig. 8, solid black)."

We also included information in the first talik figure (Fig. 1, attached) and carbon figure (Fig. 7):

"These results assume a Representative Pathway 8.5 warming scenario through 2100 and an Extended Concentration Pathway 8.5 through 2300."

Reviewer:

Please check for consistent/correct use of NBP sink/source definition (e.g. L410) Fig.8: Did you intend to put the dashed horizontal NBP threshold line at -25 gCm-2yr-1 - in accordance with your definition?

Response:

The definition of NBP has been corrected in the text and in the figures.

Reviewer:

I wonder whether a discussion of a bi-modal distribution (Fig. 7D) seems more likely than a tri-modal distribution.

Response:

We thank the reviewer for this suggestion and understand the motivation, since 2 clearly defined peaks exist on either side of the talik onset lag = 0 line. However, we argue that a tri-modal distribution is appropriate here since each peak falls into a distinct category associated with a unique process: High SOM (left peak), High Fires (right peak), and low SOM and low fires (middle peak). We have added the following discussion on L507 to justify the trimodal distribution:

"The decadal time lag between talik onset and C source transition is more normally
distributed in the remaining region, represented by the residual grey bars visible in Fig. 7D, which occurs predominantly in cold northern permafrost in northwest Siberia where low SOM (< 100 Kg m-2) and fire emission (< 25 g C m-2 yr-1) prevail. This region has a mean lag of 1 decade from talik onset to C source, with high standard deviation of lags ($\pm$ 8 decades) reflecting a skewed distribution of GPP; low productivity in cold permafrost (GPP = 385 g C m-2 yr-1) increases the likelihood that soil thaw will lead to C source transition prior to talik onset, and high productivity in warm permafrost GPP = 1111 g C m-2 yr-1) increasing the likelihood of a transition after talik onset. Cumulative C emissions from this region are on the low end (27 Pg C by 2300; Fig. 8, blue) due to low soil C (SOM = 59 kg C m-2). "

Reviewer:

Maybe a shifting of some figures (e.g. Fig 10., Fig.11) to an appendix section would be good?

Response:

Based on comments from both reviewers that Figures 10 and 11 do not add much to the paper, they have been removed, and replaced with more details describing the results of these figures

Reviewer:

L338/339 you mean positive trends?

Response:

Yes, this has been corrected

Reviewer:

L 365 and following Given the small (statistically insignificant?) trend at Drughina, probably discussing "unchanged" conditions (instead of discussing a "change in sign") is more appropriate.

Response:

Thanks for the excellent suggestions. We have modified the text as follows (L410):

"Thaw projections in northern Siberia indicate an unchanged trend and continued stability of permafrost through the early 22st century, followed by a shift to accelerated soil thaw in the early 2120, marked by onset of deep soil thaw late in the cold season."

Reviewer:

L 362 and following Please check colour specifications (in my printed version lines are yellow instead of orange, "blue" and "cyan" are used to refer to the same line, ...)

Response:

Color specifications have been corrected.

Reviewer:

L494 define "NF" - or better avoid abbreviation

Response:

NF has been expanded to non-frozen

Reviewer:

Fig.12 legend how were error bars inferred / what do they describe? Spelling

Response:

We have added a statement about error bars in figure 12 (now figure 11), which represent the multi-decadal standard deviation of seasonal and C source transition time series from 2040 – 2270.

"Time series of ecosystem C fluxes showing seasonal and decadal patterns during C source transition. This present mean and standard deviations over the period 2040-2270 for (A-B) Gross Primary Production (GPP),"

Reviewer:

L29 IS -> is

Response:

Corrected, thank you

Reviewer:

L458 form -> from

Response:

This paragraph has been removed

Reviewer:

L470 = -> -

Response:

Corrected, thank you

[Figure]

**Fig. 1.** Figure 1. Decade of projected talik formation and correlation to initial state of simulated permafrost temperature and observed permafrost extent. (A) Time series and (B) map of the simulated decade o

[Figure]

**Fig. 2.** Figure 7. Projected decade when permafrost regions shift to long-term C sources over the period 2010-2300, and relation to talik onset, soil C, and fire emissions. (A) Map of the decade of transition

[Figure]

**Fig. 3.** Figure 8. Cumulative net biome production (NBP) over northern high latitude (NHL) regions (> 55°N) from 2010 to 2300. NBP < 0 represents a net C source. NHL regions are divided into the following cate

---

## Author Response (AR1)

**Point-by-point Response**

*** Note to Editor ***

**All line numbers refer to revised manuscript. Line numbers in "tracked changes" at the end of the "Authors Response" document differ.**

\*\*\*\*\*\*\*\*\*\*\*\*\*\*\*\*\*\*\*

**Reviewer 1**

Review of "Detecting the permafrost carbon feedback: Talik formation and increased cold-season respiration as precursors to sink-to-source transitions"

The authors ran the Community Land Model (CLM) version 4.5 up to 2300, using RCP 8.5 forcing. They then perform an in depth analysis of permafrost-region dynamics in this simulation, including identifying key events: Talik formation (related the degradation of permafrost) and sink-to-source transition, i.e. the point at which the land surface changes from a net sink of carbon from the atmosphere, to a net source. It is interesting to note that this behaviour (starting as a sink and transitioning to a source) is identified across a large fraction of the current permafrost zone. However, the total carbon source is apparently only 11.6 GtC by 2300, which is low compared with previous estimates.

The authors extensively analyse different variables such as thawed volume (a newly defined metric), active layer depth, primary production, respiration and fires, and how these influence talik formation and sink-to-source transition. They find three main drivers of sink-to-source transition: 1. Active layer thickening in cold, carbon-rich high Arctic permafrost 2. Talik formation leading to winter respiration in low Arctic, warmer soils 3. Fire driven carbon source in more productive regions which dry out, and a lot of vegetation is burned. They also showed some indicators of talik formation such as a rapid increase in thawed volume immediately preceding talik formation.

This is a very thorough analysis and a well written paper that will make a great publication in The Cryosphere, after some small revisions. In general I would like to see a bit more analysis about the size of the carbon sources, not just the timing of transition. This could comprise a bit of discussion of the cumulative carbon source (11.6 Gt), and the significance of this - compared to previous estimates, and the time trajectory of the cumulative source (i.e. when does the Arctic as a whole become a net source?). And then if they could break that down to say which of the different types of source (driven by AL, talik or fires) has the bigger contribution to the total source or if these are all comparable magnitude, that would be add some value to the paper. It's fine saying that we should monitor the high Arctic systems as they will become a source soonest, but if this source is likely to be very small, there would not be so much point?

We thank the reviewer for the astute observation of the small source and excellent recommendation to enhance our discussion. We found an error in our C source budget calculation with the new total being 120 Pg C by 2300. We have corrected this error, and included more extensive analysis and discussion of cumulative carbon emissions and sources throughout the paper (see responses below).

I also suggest considering the soil types at the boreholes. It would hopefully be possible to get an idea of this from a site description or by asking the PI. For example if these are peaty soils that might explain the very slow progression of freeze-thaw compared with CLM (also relatedly, the water content).

This is a good suggestion. It's difficult to pin down in CLM if mineral soil texture affects thaw rates compared to borehole observations due to the many possible explanation for differences in thaw rates: soil organic content, lateral water flow, surface slope and aspect, ground ice. However, we have looked into soil texture effects at the Alaskan boreholes and find that higher rates of observed soil thawing may be related to 2 factors: (1) relatively dry upper soil at the Gakona and Mould Bay sites, and (2) low surface organic layer and high conductivity of the Barrow2 and Mould Bay soils.

We revised the site descriptions in methods as follows (line 245):

"*Mould Bay is a continuous permafrost tundra site with measurements at 63 depths from 0 - 3 m.* **Mould Bay has almost no organic layer (about 2 cm) and then sandy silt with high thermal conductivity.** *Barrow is a continuous permafrost tundra site with measurements at 35 depths from 0 - 15 m.* **The soil at Barrow is represented by silt with a bit of mix with some organics and almost no organic layer on top. Conductivity of the upper layer is ~1 W mK$^{-1}$ for unfrozen and ≥ 2 W mK$^{-1}$ for frozen soil.** *Gakona is a continuous permafrost forest tundra site with measurements at 36 depths from 0 - 30 m.* **Gakona has a thick organic layer of moss (0 to 5 cm), dead moss (from 5 to 13 cm), and peat (from 13 to 50 cm), then silty clay at depth.**"

We offer an explanation for high observed thaw rates in Section 3.1 (line 359):

"*Overall, we find that simulated patterns of permafrost thermal state change are consistent with available observations, but that the exact thaw rates are uncertain.* **Although there are many possible explanations for differences in observed and simulated thaw rates, we can attribute high observed thaw rates in part to a combination of (1) relatively dry upper soil at Gakona and Mould Bay, and (2) low surface organic layer and high conductivity of the Barrow and Mould Bay soils.** *We keep these uncertainties in mind as we examine patterns of change and talik formation simulated into 2300.*"

We offer some advice for future experiments in the discussion (line 586):

"*Controlled experiments demonstrating the sensitivity of talik to parameters that control soil drying such ice impedance or baseflow scalars (e.g. Lawrence et al., 2015), and the effect of organic content and mineral soil texture (Lawrence and Slater, 2008), could provide key insight on soil thermal dynamics in frozen or partially frozen conditions.*"

We also include an additional column in Table 2 for soil characteristics:

| Site | Location | Date Range | **Soil Features: Surface organic layer / Soil Type** | Depth / Number of Layers | Layer of Deepest Thaw | Month of Latest Thaw |
|------|----------|------------|------------------------------------------------------|--------------------------|-----------------------|----------------------|
|      |          |            |                                                      |                          |                       |                      |

| Mould Bay, Canada | 119.0°W, 76.0°N | 2004-2012 | **Low organic layer (~2 cm)/ Sandy silt** | 3 m / 36 | 0.69 m | September |
| --- | --- | --- | --- | --- | --- | --- |
| Barrow2, Alaska | 156.0°W, 71.3°N | 2006-2013 | **Low organic layer / Sandy silt** | 15 m / 63 | 0.58 m | October |
| Gakona1, Alaska | 145.0°W, 62.4°N | 2009-2013 | **Thick organic layer (50 cm) / Silty clay** | 30 m / 35 | 2.5 m | February |

Finally the paper is rather long and I would suggest reducing in length where possible. I have indicated a couple of points below.

*We removed figures 10 and 11 as suggested by the reviewers, but added a new figure for cumulative carbon sources to address the primary reviewer concerns.*

**Line-by-line comments**

Introduction: L62: 'Shifts in vegetation community' is mentioned in the introduction as being an important factor, but is this considered here? Are you running with dynamic vegetation? If you are, this should mentioned and if not, this omission should be dis- cussed later on. L70: Same for soil organic matter export by rivers.

*Since these processes are not considered in this analysis, we added a qualifier statement later on in the discussion (Line 590):*

***"Other factors affecting soil hydrology and carbon cycling not considered in our CLM4.5 simulations include high spatial resolution in discontinuous permafrost, shifts in vegetation community, lateral flow representation, thermokarst activity and other thaw-related changes to the ground surface, surface slope and aspect, soil heterogeneity, and potentially several other factors (see Jorgenson and Osterkamp (2005) for discussion of some of the many complexities to be considered)."***

L86-87: Include more recent references for total permafrost car- bon quantity, such as Hugelius et al 2014, Biogeosciences (https://www.biogeosciences.net/11/6573/2014/bg-11-6573-2014.pdf), and there is also a new paper by Jackson et al coming out in November with revised estimates, this will be in Annual review of Ecology, Evolution and Systematics (http://www.annualreviews.org/doi/abs/10.1146/annurev-ecolsys-112414-054234).

*References and carbon totals are updated as follows (Line 91):*

*"Talik as well as longer, deeper active layer thaw stimulate respiration of soil C (Romanovsky and Osterkamp, 2000; Lawrence et al., 2008), making the **~1035 Pg soil organic carbon in near surface permafrost (0-3 m) and ~350 Pg soil organic carbon in deep permafrost (> 3 m) vulnerable to decomposition (Hugelius et al., 2014; Jackson et al., 2017)**."*

Methods: L138-142 The standard RCP 8.5 only goes until 2100 so presumably some extension is used here? Could you mention what this looks like - for example, does it stabilise at some point or does the global temperature and CO2 just keep increasing? I also think it would be useful for comparing with the permafrost thaw results, to see a plot of the global temperature across the three centuries of future simulation. I suggest adding this at least as a supplementary figure (as there are already a lot of figures in the main manuscript).

We used the ECP8.5 scenario for the period 2100-2300. We added a time series for air temperature in Figure 1A (shown below) and modified our methodology description as follows (Line 165):

"*We use an anomaly forcing method to repeatedly force CLM4.5 with observed meteorological from the CRUNCEP dataset for the period 1996–2005 (data available at dods.ipsl.jussieu.fr/igcmg/IGCM/BC/OOL/OL/CRU-NCEP/) and monthly anomalies added based on a single ensemble member from a CCSM4 Representative Concentration Pathway 8.5 (RCP8.5) simulation for the years 2006-2100* **and Extended Concentration Pathway 8.5 (ECP8.5) for the years 2100-2300. Land air temperature for the period 2006-2300, shown in Fig. 1A., is projected to increase steadily in our simulation, with a slight decrease in the rate of warming**"

[Figure]

**Figure 1**. Decade of projected talik formation and correlation to initial state of simulated permafrost temperature and observed permafrost extent. (A) Time series and (B) map of the simulated decade of talik formation are estimated from CLM4.5 as the first decade when the mean temperature of a soil layer exceeds a freeze/thaw threshold of -0.5°C in every month. Additional colors in A represent progression of talik onset for different freeze/thaw threshold. (C) Initial permafrost temperature is defined as the annual mean soil temperature at 3 m depth from 2006-2010. (D) Permafrost extent is taken from (https://nsidc.org/data/docs/fgdc/ggd318_map_circumarctic/; Brown et al., 2001). Crosses in A, C, D represent locations of Siberian borehole measurements along the East Siberian Transect from 1955-1900 (Table 1). Circles represent locations of borehole measurements in Alaska and Canada from 2002-2013 (Table 2). Dashed black line in A shows projected air temperature over the talik region. These results assume a Representative Pathway 8.5 warming scenario through 2100 and an Extended Concentration Pathway 8.5 through 2300. We note that peak talik formation occurs around 2100.

L152-154 "The C source transition represents a shift of ecosystem C balance from a neutral or weak C sink to a long-term source driven by onset of permafrost thaw and respiration of deep SOM" - here you suggest that the deep SOM alone is driving the transition, whereas your analysis suggests that it is driven by different things depending on region. Maybe you can qualify this sentence a bit?

Great point! We have qualified this sentence as follows (L185):

"*The C source transition represents a shift of ecosystem C balance from a neutral or weak C sink to a long-term source **as C balance shifts to increasing dominance of C source processes including permafrost thaw and fires** (Koven, Lawrence and Riley, 2015).*"

L 201-204: "For comparison to projected trends in CLM4.5, we recalculate observed trends using the inter-site average from all 9 sites at 3 unique locations: northern Siberia (67°N, 144°E), southwest Siberia (61°N, 115°E), and southeast Siberia (59°N, 131°E)." This is not entirely clear. You were talking about using 6 sites and now it says 9, but then you end up with 3? Can you make it more clear? Did you combine sites into groups based on approximate locations. . .?

We clarified our analysis as follows (L231):

"***To assess observed thaw trends from 1955-1990, we analyze individual sites which report** at least 10 months yr$^{-1}$ of reported monthly mean soil temperature at each layer, and 55 months across the 5 layers (out of 60 possible layer-months per year). Based on these requirements, we find that 6 of 9 sites yield at least 6 years of data over multiple decades, and are well suited for examining historical thaw trends. **For comparison of observed trends to historical and projected trends from 1950-2300, we analyze clusters of sites by combining the 9 sites into 3 groups based on approximate locations, and calculate** observed trends using the inter-site average at each location. **We use 2 sites in northern Siberia (67°N, 144°E), 6 sites in southwest Siberia (61°N, 115°E), and 1 site in southeast Siberia (59°N, 131°E).** Site information is shown in more detail in Table 1.*"

Results: L228 Do you mean 2300?

Yes, thank you for identifying this mistake!

L266 "Our simulations show a similar drying pattern in shallow layers (~0-1 m depth) in the 4 decades prior to talik onset (Fig. 2D)." The shallow drying does not appear to be shown on Figure 2D, only the total column soil moisture?

We added a line for shallow soil moisture (see attached revision of Fig 2), and revised the text as follows (L07):

"*Our simulations show a **similar, but very slight,** drying pattern in shallow layers in the 4 decades prior to talik onset (**1.3% loss of soil moisture over 0-1 m depth; Fig. 2D**), **accounting for about half of** total water storage **loss in** the column. **More** significant changes in water balance **occur** following talik onset, including **more** rapid **drying in shallow layers (~10% over 4 decades) and in the column (~16%), and a substantial** increase in sub-surface drainage, as discussed below.*"

[Figure]

**Figure 2**. Patterns showing the progression of soil thaw in the decades surrounding talik onset. Individual lines represent averages across the subset of talik forming regions for each decade from the 2050s (darkest red) to the 2250s (darkest blue). (A) Integrated soil thaw volume, where the vertical solid line represents the mean timing of initial thaw at depth and late into the cold season (Jan-Apr). Note that the upper limit to the thaw volume metric in (A) is an artifact of the arbitrary maximum soil depth of 45.1m in CLM4.5. Other panels show (B) Date of spring surface thaw in the uppermost layer, (C) annual maximum active layer thickness, and (D) annual sub-surface drainage (solid) and volumetric soil moisture averaged over the soil column (dashed) and. Grey shaded areas show the standard deviation of results for individual talik formation decades. Mean behavior exhibits a characteristic pattern: gradual increase in thaw volume and active layer depth prior to talik onset, abrupt shift in thaw volume, active layer depth, followed by stabilization to constant thaw volume as soil drying and sub-surface drainage increases.

L281-2 "we find more pronounced tilting of the thawed layer with time and depth" This is not obvious to me from the plot. I might just suggest deleting this.

"Pronounced tilting" is a misleading description of the thaw pattern. We revised this description as follows (L323):

"*In the 3 decades leading up to talik onset, we find **gradual deepening of the thawed layer** to 3-4 m and penetration of thaw period into Jan-Feb.*"

L290-2 "the rate of thawing and drainage in response to permafrost thaw may be under-estimated in deeper CLM4.5 layers near bedrock due to reduced heat capacity." Sorry if I am missing something here but it doesn't seem to me like reduced heat capacity would reduce the rate of thawing, but rather than it would thaw more quickly because less heat is needed to thaw? Can you check this? Thanks.

Yes, thanks for catching this inaccurate statement. We clarified as follows (L332).

"*We note that bedrock soil is not hydrologically active in CLM4.5, and thus the rate of thawing and drainage in response to permafrost thaw may be **overestimated** in deeper CLM4.5 layers near bedrock due to reduced heat capacity.*"

L313-315 "however, our comparison to observations suggests that simulated thaw rates in this region and for similar permafrost temperatures are underestimated". This is not totally clear. Which comparison with obs? Are you referring to the comparison against thaw rates in Siberia which comes in the following section? Or are you inferring this from your comparison against borehole temperatures?

We clarified the discussion as follows (L355):

"*Talik onset in CLM4.5 is variable in **the region containing Gakona (southeast Alaska)** with earliest onset by mid-century (~2050s, Fig. 1A); however, our comparison to **borehole temperature data at Gakona** suggests that simulated thaw rates **in southwest Alaska and across pan-Arctic regions with** similar permafrost temperatures are underestimated, and that earliest onset may occur sooner than predicted.*"

L335-336 "5 Siberian borehole sites which recorded at least 5 years of data spanning multiple decades:" Earlier you were talking about having 6 sites (or 9, or 3) and here it is 5. Please just clarify this a bit!

Should be 6, thank you! We corrected as follows (L383):

*"We focus first on site-specific long-term **historical** trends by analyzing the **6** Siberian borehole sites which recorded at least **5** years of **temperature** data spanning multiple decades: Drughina, Lensk, Macha, **Oimyakon**, Uchur, and Chaingda."*

L337 "Records at these locations show a decrease in thaw volume" Do you mean an increase? On the next line it also refers to 'negative trends', which doesn't seem to fit with the plots/results (or the expectations!). Maybe these things should read 'increase in thaw volume' and 'positive trends'?

Yes, this should read "increase in thaw volume" and "positive trend". We corrected as follows (385):

*"Records at these locations show an **increase** in thaw volume with an average positive trend of 0.19 m months yr$^{-1}$ from 1955 – 1990 (Table 1, Fig 5). All sites except Drughina show **positive trends**"*

L345-346 "(layer thickness increases exponentially with depth along the Siberian tran- C4

sect)" This is not totally clear. Do you mean that active layer thickness increases expo- nentially with latitude. . .? Is that data shown somewhere? (It doesn't necessarily need to be, maybe just write 'data not shown' if it isn't)

It's not clear why we added this statement, and it doesn't appear to affect the analysis, so we removed it and modified the text as follows (L390):

*"Further examination indicates that active layer thickness at Drughina actually decreased to 0.8 meters from 1989-1990 compared to 1.2 meters in the 1970s **(data not shown)**. Drughina also shows smaller average thaw volume magnitude compared to other sites, consistent with shallower thaw. Together, these findings indicate that active layer thickness is decreasing at Drughina."*

L348-355. I'm not sure how much this is adding overall. It gets a bit confusing. Where you say "(vertical dashed line)", I would change to '(vertical dashed lines on Figure 5)', assuming this is what you're referring to? Anyway, it gets confusing when talking about groups of sites and it being hard to identify those groups. I would maybe condense these lines to something along the lines of "There is considerably spatial variability in thaw trends, for example site X is this far from site Y [relatively close] but with Z difference in trends [relatively large]. Talik formation occurs at several sites, at differ- ent times between 1957 and 1990 (shown by vertical dashed lines on Figure 5). We acknowledge the difficulty. . ."

This section sets up empirical evidence for increasing permafrost degradation from west to east in Siberia. We therefore clarify and condensed this section as follows (L395):

**"There is considerable spatial variability in thaw volume and trends, but in general thaw trends increase from west (0.18 m months yr-1) to east (0.51 m months yr-1). Talik forms at several sites, at different times between 1957 and 1990 (shown by vertical dashed lines on Fig. 5), with earlier talik to the west consistent with higher mean initial thaw volumes."**

L364-366 "The simulated trend in thaw volume shows a change in sign at northern locations (blue), acceleration of thaw at southwest sites (orange), and reduction of thaw at the southeast sites (brown)". This sentence suggests that the thaw volume reduces at the southeast sites, I guess because the thaw volume in CLM is less than the observations, but I would be inclined to interpret this instead as: the CLM simulation always had a too-small thaw volume, and there was never any 'decrease in thaw volume' in the simulation. But it is not possible to tell from the plot - Why did you not include the historical CLM simulation on the plot so it would overlap with the observed period? I also wouldn't say there is a "change of sign" at the northern locations. I guess you refer to the fact that the thaw volume is slightly decreasing at the northern sites historically, and increasing in the future? But as you say this is a very small trend so I would probably instead interpret this as a relatively stable site that shifts to degradation towards the end of the century? I also find figure 6 a bit confusing with the symbols and what they represent. So apparently the circle represents 'thaw onset in January' but this happens considerably sooner in the simulation than 'thaw onset in March', which doesn't make sense to me? Thaw onset should get earlier each year? In the main text it implies this is actually referring to deep thaw lasting throughout the winter (but this is not implied in the figure caption!). Maybe it would just be best not to include the symbols at all. This would make the plot simpler to interpret. Lines 369 "Our simulations show a shift to accelerated soil thaw beginning in the early 2080s". This sounds like you are referring to the whole simulation and not just this one particular point - can you make this more clear?

We apologize for the confusion this section created. We added historical thaw simulations from CLM4.5 so the simulated record is continuous from 1950-2300. We also removed the thaw onset symbols, which were meant to represent the progression of thaw later into the cold season (from early winter (Oct-Dec) to deep winter (Jan-Apr)) prior to talik onset. The revised figure and text are shown below (405):

[revised manuscript text omitted]

L388 "followed by gradual decline to 0.5 Pg C by 2300". Does this suggest the temperature has stabilised and things are moving back towards equilibrium or is it more complicated than that? Could you comment? Including the supplementary plots of temperature trajectories that I suggested earlier might clear this one up.

Temperatures continue to rise based on ECP8.5 which suggests it is more complicated. We clarify this section, including more detailed analysis of C source magnitudes and categories, as follows (L431):

**"Net C emissions from C source transition regions are a substantial fraction of the total NHL C budget over the next 3 centuries (Fig. 8). The cumulative pan-Arctic C source increases slowly over the 21ˢᵗ century, reaching 10 Pg C by 2100 with RCP8.5 warming, then increases more rapidly to 70 Pg C by 2200 and 120 Pg by 2300 with sustained ECP8.5 warming (Fig. 8, solid black). This pan-Arctic source represents 86% of cumulative emissions in 2300 from the larger NHL talik region (crosses), despite the 2 fold smaller land area, and exceeds the talik region through 2200 due to mitigating widespread vegetation C gains (Koven et al., 2015). Cumulative emissions over all NHL land regions (diamonds, > 55N) increase in similar fashion to the talik region, reaching 120 Pg C by 2200 and 220 by 2300, with no sign of slowing."**

L405-412. Here you are talking about NBP as positive, increasing, but in the plot (and according to your stated sign convention), a source is represented by negative NBP, decreasing. Please make this paragraph consistent.

Corrected as follows (L458):

*"**In these regions**, thaw volume is low (< 50 m months yr⁻¹) and shows a weak relationship to NBP (NBP **decreases** much faster than thaw volume) prior to C source onset (indicated by large green circle in Fig. 8A)."*

L437 - Again wrong sign convention for NBP?

Corrected as follows (489):

*"and talik formation occurs when these regions are weak sinks (NBP > 0 g C m⁻² yr⁻¹)."*

We have also revised Figure 8 (now Figure 9) with arrows indicating C source or sink for clarification:

[Figure]

**Figure 9**. Net biome production (NBP) as a function of thaw volume. Symbols represent NBP and thaw volume values averaged over regions which transition to long term C source from 2060-2140, binned into regions where the decade of C source transition (A) leads talik onset, (B) lags talik onset, and (C) lags talik onset AND where fires exceed 25 g C m⁻² yr⁻¹. Colors indicate decade relative to C Source transition, denoted by the large green marker, which occurs when NBP exceeds -25 g C m⁻² yr⁻¹ (grey horizontal dashed line). The grey square marker indicates the mean NBP and thaw volume values during talik onset. Cases where C source leads talik (A) show small thaw volumes during C source transition, and amplified C sources during talik onset. Cases where C source lags talik (B-C) show large thaw volumes during C source transition, and C sinks during talik onset.

L437-438 "In general, C sources in these regions are more sensitive to C emissions from deep soil thaw" Have you actually quantified how much of the C is coming from deep soil. . .?

Unfortunately, we can't quantify C from deep soils since vertical resolved C flux output is not available. We have revised the statement to reflect an inferred contribution from deep soils (L490):

*"In general, **C source onset under high thaw volume indicates** these regions are more sensitive to C emissions from deep soil thaw."*

Figure 10: I think this could also be a supplementary figure or removed altogether, maybe giving slightly more detail on the numbers where it's mentioned in the text.

We have removed Figure 10 and revised the text as follows (L470):

"*A broader analysis of soil thaw statistics over all regions and periods indicates **that most C source transitions (~2.3 million km$^2$, or 77% of land where C source leads talik) occur at active layer depths below 3 m  and thaw season penetration into November**.*"

Figure 11 / Lines 453-461. I am struggling to interpret the upper plot on this Figure, and I am wondering whether this part adds much to the analysis. Since the analysis is already long I might suggest removing this paragraph and figure.

Agreed. The main point is the difference in GPP between warm and cold permafrost regions, which is discussed in previous sections. We have removed this paragraph and figure.

I suggest doing a bit more quantification of the contribution to total carbon sources. If there is a total of 11PgC emitted by 2300, what fraction of that comes from the three different 'categories' of points in the trimodal distribution? I think it would be really useful to know which are the important carbon sources - or whether they are all similar. (I have made this comment again above)

We have added a new figure (Figure 8) quantifying the different carbon sources (below). We also found that our cumulative C emission estimate was off by a factor of 10 (didn't convert from year to decade), bringing our new C emission to 120 PgC by 2300.

[revised manuscript text omitted]

Revised as follows (L527):

*"GPP and combined respiration **increase by ~15%** per decade for each permafrost regime surrounding the decade of C source transition with peak fluxes in the growing season (Fig. 11 A - D)."*

L485-487 "The trend in soil vs litter respiration explains almost the entire trend in net ecosystem C balance from neutral to net source (Fig. 12 G – H)." I'm not sure what you mean by this. I would have thought the trend in NBP is determined by the difference between GPP and total respiration, rather the difference between soil and litter respiration? And how do you draw these conclusions from the plots? Please could you make this clearer! Thanks.

We have revised as follows (L541):

*"**The trend in the respiration difference in warm and cold permafrost, which increase by similar amounts (~100 g C $m^{-2}$ $yr^{-1}$), thus reflects an increasing dominance of respiration from younger and older soil C pools, respectively. These trends are identical to the corresponding NBP trends, which decrease by 100 g C $m^{-2}$ $yr^{-1}$ over the same period from neutral to net**"*

*source (Fig. 12 G – H), such that the differences between GPP and respiration driving the NBP trends are explained almost entirely by the increasing fraction of soil vs litter respiration."*

L494 NF was only defined at the beginning of the paper and never used again until this point, which means that I had forgotten what it meant by now. You don't use this abbreviation much so I suggest you don't need it.

NF has been changed to non-frozen.

Discussion: L526-529 "Experiments demonstrating the sensitivity of talik to soil drying within the active layer across soil hydrology schemes in previous (CLM4), current (CLM4.5), and newly available (CLM5) versions of CLM could provide key insight on soil thermal dynamics in frozen or partially frozen conditions." This comes out of the blue a bit, and I am also not clear about why this would be useful? Comparing different hydrology schemes could be useful if one is more realistic than the others or includes different processes? But it is not clear that this would be the case in different CLM versions? It might be better to look at a purpose-built permafrost model, for example, or a model that resolves discontinuous permafrost.

We agree it's not clear how exactly this would be useful since the model differences in soil hydrology and snow are not systematically differences. We don't think a purpose-built permafrost model is likely to be any better since the processes represented are typically the same as in CLM. We agree that resolving discontinuous permafrost would be good, but it's unclear how to do that other than with increased resolution, but this would miss other important processes such as lateral flow. Something more controlled like a parameter sensitivity study with variable ice impedance or baseflow scalar, such as examined in Lawrence et al., 2015, is likely to provide the best insight. We therefore replace our comment with a more general comment about uncertainties, and suggest more controlled experiments similar to Lawrence (L586):

**"*Controlled experiments demonstrating the sensitivity of talik to parameters that control soil drying such ice impedance or baseflow scalars (e.g. Lawrence et al., 2015), and the effect of organic content and mineral soil texture (Lawrence and Slater, 2008), could provide key insight on soil thermal dynamics in frozen or partially frozen conditions. Other factors affecting soil hydrology and carbon cycling not considered in our CLM4.5 simulations include high spatial resolution in discontinuous permafrost, shifts in vegetation community, lateral flow representation, thermokarst activity and other thaw-related changes to the ground surface, surface slope and aspect, soil heterogeneity, and potentially several other factors (see Jorgenson and Osterkamp (2005) for discussion of some of the many complexities to be considered).*"**

L538-540 "thus appears to be driven by combination of warming and increased nitrogen availability resulting from permafrost thaw". I would suggest changing "appears to be" to "may be". . . you haven't looked at nitrogen here, and it is not totally clear/agreed that permafrost thaw will increase nitrogen availability.

We have revised as suggested

L605 "Our main results emphasize an emergence of cold season processes" Not sure what you mean by 'an emergence of cold season processes'. . .? Rephrase?

Rephrased as follows (L692):

*"Our main results emphasize **an increasingly important impact of NHL cold season warming on** earlier spring thaw, longer non-frozen seasons, and increased depth and seasonal duration of soil thaw."*

L629-630 "Active-layer deepening leads to C sink-to-source transitions in some re- gions, talik-driven permafrost loss in others,. . ." This should be rephrased, "C sink-to- source transitions are caused by active layer deepening in some regions, talik-driven permafrost loss in others,..." Otherwise the sentence doesn't quite make sense, it sounds like active-layer deepening is causing all of the other things!

Rephrased as suggested.

Hope you find these comments helpful. Best wishes!

Thank you, much improved now!

**Reviewer 2**

Overall impression

In the current study, Parazoo and colleagues have used the land surface model CLM4.5 to simulate permafrost state and changes from 2010 to 2300 under strong warming following the RCP8.5 scenario. They have investigated how permafrost degradation evolves in space and time, with a special focus on how talik formation affects thaw dynamics. Further, the authors have used their model experiment to analyse how future C fluxes in permafrost regions will evolve and when a carbon sink to source transition is likely to occur along different regions of the permafrost domain.

The presented analyses are helpful for increasing our process understanding of how individual factors explain inferred differences in simulated carbon fluxes between cold and warmer permafrost regions. E.g. the authors find that cold permafrost locations become C sources due to altered thaw-season dynamics while transitions of warm permafrost regions are mainly affected by changes in cold season dynamics. Further, the authors discuss how the presented results of this study can help finding an (optimal) design for monitoring the thermal and carbon state and changes in permafrost regions. The paper is well structured and written, and model analyses were performed elaborately. Adding new insights into permafrost degradation and carbon dynamics, this study can be considered of broad interest to the readership of The Cryosphere.

General comments

1) Initial SOC storages The initialized soil C stocks play a key role in affecting simulated future C release and the timing for a sink to source transition. In the study presented, no information is given how these C stocks were initialized for the simulation setting used here (besides referring to two previous CLM4.5 studies). Information should be provided in a sub-section on how SOC stocks were initialized, and on how these stocks compare to observed data (e.g. NCSCD - in terms of total storages and with regard to CLM4.5 inferred high (peaty) SOC storages at northern grid cells). As talik formation down to some meters are analysed in this study, I wonder how deep SOC is initialized in the soil column? Can e.g. soil thaw deeper than 3 meters further increase the pool of thawed carbon available for decomposition? A further key factor not discussed in the manuscript concerns assumptions about SOM lability made in the model. As especially uncertainty in slowly decomposing SOM is very high, I wonder how different settings of a humus timescale parameter (or different partitioning between labile and less labile pools) would affect inferred sink to source transition times? If feasible, this would be worth exploring by an extra sensitivity run, or at least by discussing qualitatively the impact of uncertainty in assumed SOM decomposition timescales on the findings discussed here.

We thank the reviewer for the recommendation to include the very important discussion of soil C stocks and decomposition controls in CLM. Deep soil decomposition experiments in CLM have been run in previous studies by the co-authors, to decouple changing dynamics of surface soils from deep soils, to determine the potential contribution from permafrost layers, and to calibrate initial soil C stocks against observations.

We have elaborated on these details in the methods (L139) as follows:

"**CLM is spunup to C equilibrium for the year 1850 by repeatedly cycling through 20 years of pre-industrial climate forcing with CO$_2$ and N-deposition set at 1850 levels. C initialization is achieved via slow mixing by cryoturbation between the seasonally thawed active layers and deeper permafrost layers (Koven et al., 2009). Including vertically resolved processes leads to a sign change in the projected high-latitude C response to warming, from net C gains driven by increased vegetation productivity to net C losses from enhanced SOM decomposition (Koven et al., 2011)**. *The soil grid includes 30 vertical levels that has a high-resolution exponential grid in the interval 0–0.5 m and fixed 20-cm layer thickness in the range of 0.5–3.5 m to maintain resolution through the base of the active layer and upper permafrost, and reverts to exponentially increasing layer thickness in the range 3.5–45 m to allow for large thermal inertia at depth.* **Soil C turnover in CLM4.5 is based on a vertical discretization of first-order multipool SOM dynamics (Koven et al., 2013; Oleson et al., 2013) where decomposition rates as a function of soil depth are controlled by a parameter $Z\tau$ (Koven et al., 2015; Lawrence et al., 2015). This depth control of decomposition represents the net impacts of unresolved depth dependent processes. In this study, we utilize $Z\tau=10$ m, which yields a weak additional depth dependence of decomposition beyond the environmental controls and, as discussed and evaluated relative to $Z\tau=1m$ and $Z\tau=0.5m$ in Koven et al (2015), results in CLM permafrost-domain soil C stocks that are in closest agreement (1582 Pg for $Z\tau=0.5$ m, 1331 Pg for $Z\tau=1m$, and 1032 Pg for $Z\tau=10m$) with observed estimates (1060 Pg C to 3 m depth; Hugelius et al 2013). This reduction in initial C is due to higher decomposition rates at depth during the model initialization period. There is no C below 3.5 m, so additional thaw below 3 meters has a small impact on the C cycle. We note that the relationship applied in CLM4.5, which implies multiplicative impacts of limitations to decomposition, is commonly applied in land biogeochemical models, but is quite uncertain**."

We also discuss implications of 3.5 m C limit in discussion (Line 626):

"**Our estimate of C emissions following talik onset (~20 Pg C) is low compared to the cumulative emissions from all long term C source transitions (120 Pg C), but likely strongly underestimated. Soil C is not permitted below 3.5 m in CLM4.5, or in most analogous models, such that potential decomposition of the ~350 Pg soil organic C in deep permafrost (> 3 m) is not accounted for (Hugelius et al., 2014; Jackson et al., 2017). This is significant for our simulations, which show frequent talik formation and accelerating thaw volumes below 3 m (e.g., Fig. 3). We therefore caution the reader in the interpretation of the timing and**

*magnitude of permafrost C emissions following talik onset in our simulations, which represent a lower bound of potential emissions based on the current formulation of CLM4.5."*

Talik formation

The study focuses on talik formation as a key process which leads to abrupt permafrost degradation. The discussion (and the model simulation) is done for non-lake environments. Talik formation through thermokarst lake initialisation is not considered and mentioned. Yet this process is known to lead to rapid thaw through pronounced sub- lake talik formation, which can strongly affect carbon release from thawed sub-lake sediments. Although it is questionable to which extent future Arctic landscapes will be affected by thermokarst formation, this process should be discussed in the context of future permafrost degradation and permafrost carbon release.

We agree that thermokarst is an important process which is not considered in our simulations. We added the following discussion of thermokarst effects to the intro (Line 87)

"***More abrupt processes such as thermokarst lake initialization can also lead to rapid thaw through pronounced sub-lake talik formation (Jorgenson and Osterkamp, 2005).*** *These processes can initiate formation of a talik zone (perennially thawed sub-surface soils) during active layer adjustment to new thermal regimes (Jorgenson et al., 2010)* ***in lake and non-lake environments***."

And methods (Line 135)

"*More abrupt thaw processes affecting permafrost C dynamics and talik formation such as the effect of thermokarst or other thaw related landscape dynamics changes in wetland or lake distribution are not accounted for ( see Riley et al (2011) for more discussion).*"

and discussion (Line 590):

"*Other factors that are necessary for more accurate simulation of soil hydrology and carbon cycling but not considered in CLM4.5 and analogous models include: enhanced spatial resolution in complex topography and in discontinuous permafrost, sophisticated vegetation community definitions including trait based representation, shifts in vegetation community, lateral flow representation,* ***thermokarst activity and other thaw-related changes to the ground surface****, surface slope and aspect, soil heterogeneity, and potentially several other factors (see Jorgenson and Osterkamp (2005) for discussion of some of the many complexities to be considered).*"

Vegetation distribution

As discussed extensively in the presented paper, the Arctic land carbon balance is determined by changes in the net flux of vegetation carbon uptake and respiration losses. To what extent is CLM4.5 able capturing high latitude vegetation distribution/patterns? Some short discussion on simulated high latitude vegetation in CLM4.5 would be interesting to include.

The use of static vegetation distributions and simple vegetation community definitions is a major source of uncertainty in CLM4.5. We have acknowledged this model weakness in the response above.

Cumulated C fluxes

I guess you want to discuss C source numbers as PgC per year? (L387). Please specify in the text and in Fig.7B y-axis label. How does the cumulative C source from 2010- 2300 of 11.6 PgC relate to shown C release rates in Fig. 7B? If I interpret numbers shown correctly, these suggest much larger cumulative release. Given published work on total C release from future permafrost degradation under RCP8.5 in other studies (suggesting much larger release), this number (total C release) should be discussed in the context of existing estimates.

We thank the reviewer for correctly pointing out our y-axis error and suggesting to compare our cumulative releases to existing estimates. We corrected the y-axis error by moving the C source numbers from Fig. 7B to Fig. 8 for more in depth analysis (Figure 7 below; Figure 8 provided in Reviewer 1 responses). Our cumulated emission is 120 Pg C which is more in line with existing estimates. We summarize in the discussion as follows (Line 596):

*"Our simulations show increasing C emissions over time across the talik region (Fig. 1B), as cumulative NBP becomes increasingly negative (NBP < 0 equals a net C source), reaching a net source of 140 Pg C by 2300 (Fig. 8, crosses), consistent with previous estimates of net C balance across the larger pan-Arctic region from CLM4.5 (~160 Pg C, Koven et al., 2015; Lawrence et al., 2015). Ecosystems which transition from net C sinks to net C sources represent less than half the total talik area (6.8 of 14.5 Million km², Fig. 7A), but account for most (~85%) of the cumulative emissions, reaching 10 Pg C in 2100, 70 Pg C in 2200, and 120 Pg C by 2300 (Fig. 8, solid black). Removing the effect of vegetation C gain (~20 Pg C in 2100 and 40 Pg C in 2200 and 2300 according Koven et al., 2015), we estimate a cumulative permafrost emission for C source transition regions of 30 Pg C in 2100, 110 Pg C in 2200, and 160 Pg C in 2300. These numbers are on the low end but consistent with estimates of permafrost C emissions summarized by Schuur et al. (2015), which range from 37-174 Pg C by 2100 and 100-400 Pg C by 2300."*

[Figure]

**Figure 7**. Projected decade when permafrost regions shift to long-term C sources over the period 2010-2300, and relation to talik onset, soil C, and fire emissions. (A) Map of the decade of transition to C source, reflected in the color code, showing earlier transitions in cold northern permafrost. (B) The area of land that transitions peaks in the late 21st century, and is driven by regions where the C source leads talik onset (dashed). (C) The decadal time lag from talik onset to C source transition shows positive lags in warm southern permafrost (C source lags talik) and negative lags in cold norther permafrost (C source leads talik). (D) Histogram shows trimodal distribution of permafrost area as a function of decadal time lag, with negative lags related to high soil organic matter (green bars and map in E), and large positive lags related to fires (red bars and map in F) but delayed by high productivity. See text for details. These results assume a

Representative Pathway 8.5 warming scenario through 2100 and an Extended Concentration Pathway 8.5 through 2300.

Specific comments

Results presented in this study are inferred for the RCP8.5 scenario. This should be made more clear in the text (when discussing future evolution of permafrost and carbon fluxes), and in some of the figure legends.

We have added RCP8.5 and ECP8.5 information throughout the text:

L263 Section 3.1:

*"Our simulations show widespread talik formation throughout Siberia and northern North America over the period 2010-2300 (Fig. 1B), impacting ~14.5 million km$^2$ of land in NHLs (55°-80°N) assuming a freeze/thaw threshold of -0.5°C **and RCP8.5 and ECP8.5 warming scenarios**."*

L432 Section 3.3:

*"The cumulative pan-Arctic C source increases slowly over the 21$^{st}$ century, reaching 10 Pg C by 2100 with RCP8.5 warming, then increases more rapidly to 70 Pg C by 2200 and 120 Pg by 2300 **with sustained ECP8.5 warming (Fig. 8, solid black)**."*

We also included information in the first talik figure (Fig. 1) and carbon figure (Fig. 7):

**"These results assume a Representative Pathway 8.5 warming scenario through 2100 and an Extended Concentration Pathway 8.5 through 2300."**

Please check for consistent/correct use of NBP sink/source definition (e.g. L410) Fig.8: Did you intend to put the dashed horizontal NBP threshold line at -25 gCm-2yr-1 - in accordance with your definition?

The definition of NBP has been corrected in the text and in the figures.

I wonder whether a discussion of a bi-modal distribution (Fig. 7D) seems more likely than a tri-modal distribution.

We thank the reviewer for this suggestion and understand the motivation, since 2 clearly defined peaks exist on either side of the talik onset lag = 0 line. However, we argue that a tri-modal distribution is appropriate here since each peak falls into a distinct category associated with a unique process: High SOM (left peak), High Fires (right peak), and low SOM and low fires (middle peak). We have added the following discussion on L507 to justify the trimodal distribution:

**"The decadal time lag between talik onset and C source transition is more normally distributed in the remaining region, represented by the residual grey bars visible in Fig. 7D, which occurs predominantly in cold northern permafrost in northwest Siberia where low SOM (< 100 Kg m-2) and fire emission (< 25 g C m$^{-2}$ yr$^{-1}$) prevail. This region has a mean lag of 1 decade from talik onset to C source, with high standard deviation of lags (± 8 decades) reflecting a skewed distribution of GPP; low productivity in cold permafrost (GPP = 385 g C m$^{-2}$ yr$^{-1}$) increases the likelihood that soil thaw will lead to C source transition prior to talik onset, and high productivity in warm permafrost GPP = 1111 g C m$^{-2}$ yr$^{-1}$) increasing the**

*likelihood of a transition after talik onset. Cumulative C emissions from this region are on the low end (27 Pg C by 2300; Fig. 8, blue) due to low soil C (SOM = 59 kg C m$^{-2}$). "*

Maybe a shifting of some figures (e.g. Fig 10., Fig.11) to an appendix section would be good?

Based on comments from both reviewers that Figures 10 and 11 do not add much to the paper, they have been removed, and replaced with more details describing the results of these figures

L338/339 you mean positive trends?

Yes, this has been corrected

L 365 and following Given the small (statistically insignificant?) trend at Drughina, probably discussing "unchanged" conditions (instead of discussing a "change in sign") is more appropriate.

Thanks for the excellent suggestions. We have modified the text as follows (L410):

*"Thaw projections in northern Siberia indicate an **unchanged** trend and continued stability of permafrost through the early 22$^{st}$ century, followed by a shift to accelerated soil thaw in the early 2120, marked by onset of deep soil thaw late in the cold season."*

L 362 and following Please check colour specifications (in my printed version lines are yellow instead of orange, "blue" and "cyan" are used to refer to the same line, ...)

Color specifications have been corrected.

L494 define "NF" - or better avoid abbreviation

NF has been expanded to non-frozen

Fig.12 legend how were error bars inferred / what do they describe? Spelling

We have included a description of error bars in figure 12, which represent the multi-decadal standard deviation of seasonal and C source transition time series from 2040 – 2270.

L29 IS -> is

Corrected, thank you

L458 form -> from

This paragraph has been removed

L470 = -> -

Corrected, thank you

**List of Changes**

- L29: "Widespread talik at depth is projected across most of the NHL permafrost region (14 million km$^2$) *by 2300, 6.2 million km$^2$ of which is projected to become a long term C source,* **emitting 10 Pg C by 2100, 50 Pg C by 2200, and 120 Pg C by 2300, with few signs of slowing.** *Roughly half of the projected C source region occurs in predominantly warm sub-Arctic permafrost following talik onset.* **This region emits only 20 Pg C by 2300, but the CLM4.5 estimate may be biased low by not accounting for deep C in yedoma.** *Accelerated decomposition of deep soil C following talik onset shifts the ecosystem C balance away from surface dominant processes (photosynthesis and litter respiration), but sink-to-source transition dates are delayed by 20-200 years by high ecosystem productivity, such that talik peaks early (~2050s, borehole data suggests sooner) and C source transition peaks late (~2150-2200). The remaining C source region is in cold northern Arctic permafrost, which shifts to a net source early (late 21$^{st}$ century),* **emitting 80 Pg C by 2300,** *and prior to talik formation due to the high decomposition rates of shallow, young C in organic rich soils coupled with low productivity.*"

- Line 87: "***More abrupt processes such as thermokarst lake initialization can also lead to rapid thaw through pronounced sub-lake talik formation (Jorgenson and Osterkamp, 2005).*** *These processes can initiate formation of a talik zone (perennially thawed sub-surface soils) during active layer adjustment to new thermal regimes (Jorgenson et al., 2010)* ***in lake and non-lake environments.***"

- L91: "*Talik as well as longer, deeper active layer thaw stimulate respiration of soil C (Romanovsky and Osterkamp, 2000; Lawrence et al., 2008), making the* ~***1035 Pg soil organic carbon in near surface permafrost (0-3 m) and ~350 Pg soil organic carbon in deep permafrost (> 3 m) vulnerable to decomposition (Hugelius et al., 2014; Jackson et al., 2017).***"

- Line 135: "*More abrupt thaw processes affecting permafrost C dynamics and talik formation such as the effect of thermokarst or other thaw related landscape dynamics changes in wetland or lake distribution are not accounted for ( see Riley et al (2011) for more discussion).*"

- L139: "**CLM is spunup to C equilibrium for the year 1850 by repeatedly cycling through 20 years of pre-industrial climate forcing with CO$_2$ and N-deposition set at 1850 levels. C initialization is achieved via slow mixing by cryoturbation between the seasonally thawed active layers and deeper permafrost layers (Koven et al., 2009). Including vertically resolved processes leads to a sign change in the projected high-latitude C response to warming, from net C gains driven by increased vegetation productivity to net C losses from enhanced SOM decomposition (Koven et al., 2011).** *The soil grid includes 30 vertical levels that has a high-resolution exponential grid in the interval 0–0.5 m and fixed 20-cm layer thickness in the range of 0.5–3.5 m to maintain resolution through the base of the active layer and upper permafrost, and reverts to exponentially increasing layer thickness in the range 3.5–45 m to allow for large thermal inertia at depth.* **Soil C turnover in CLM4.5 is based on a vertical discretization of first-order multipool SOM dynamics (Koven et al., 2013; Oleson et al., 2013) where decomposition rates as a function of soil depth are controlled by a parameter Zτ (Koven et al., 2015; Lawrence et al., 2015). This depth control of decomposition represents the net**

*impacts of unresolved depth dependent processes. In this study, we utilize Zτ=10 m, which yields a weak additional depth dependence of decomposition beyond the environmental controls and, as discussed and evaluated relative to Zτ=1m and Zτ=0.5m in Koven et al (2015), results in CLM permafrost-domain soil C stocks that are in closest agreement (1582 Pg for Zτ=0.5 m, 1331 Pg for Zτ=1m, and 1032 Pg for Zτ=10m) with observed estimates (1060 Pg C to 3 m depth; Hugelius et al 2013). This reduction in initial C is due to higher decomposition rates at depth during the model initialization period. There is no C below 3.5 m, so additional thaw below 3 meters has a small impact on the C cycle. We note that the relationship applied in CLM4.5, which implies multiplicative impacts of limitations to decomposition, is commonly applied in land biogeochemical models, but is quite uncertain*."

- L165: "*We use an anomaly forcing method to repeatedly force CLM4.5 with observed meteorological from the CRUNCEP dataset for the period 1996–2005 (data available at dods.ipsl.jussieu.fr/igcmg/IGCM/BC/OOL/OL/CRU-NCEP/) and monthly anomalies added based on a single ensemble member from a CCSM4 Representative Concentration Pathway 8.5 (RCP8.5) simulation for the years 2006-2100 **and Extended Concentration Pathway 8.5 (ECP8.5) for the years 2100-2300. Land air temperature for the period 2006-2300, shown in Fig. 1A., is projected to increase steadily in our simulation, with a slight decrease in the rate of warming***"

- L185: "*The C source transition represents a shift of ecosystem C balance from a neutral or weak C sink to a long-term source **as C balance shifts to increasing dominance of C source processes including permafrost thaw and fires** (Koven, Lawrence and Riley, 2015).*"

- L231: "***To assess observed thaw trends from 1955-1990, we analyze individual sites which report** at least 10 months yr$^{-1}$ of reported monthly mean soil temperature at each layer, and 55 months across the 5 layers (out of 60 possible layer-months per year). Based on these requirements, we find that 6 of 9 sites yield at least 6 years of data over multiple decades, and are well suited for examining historical thaw trends. **For comparison of observed trends to historical and projected trends from 1950-2300, we analyze clusters of sites by combining the 9 sites into 3 groups based on approximate locations, and calculate** observed trends using the inter-site average at each location.*

- *L238: **We use 2 sites in northern Siberia (67°N, 144°E), 6 sites in southwest Siberia (61°N, 115°E), and 1 site in southeast Siberia (59°N, 131°E).** Site information is shown in more detail in Table 1.*"

- L245: "*Mould Bay is a continuous permafrost tundra site with measurements at 63 depths from 0 - 3 m. **Mould Bay has almost no organic layer (about 2 cm) and then sandy silt with high thermal conductivity**. Barrow is a continuous permafrost tundra site with measurements at 35 depths from 0 - 15 m. **The soil at Barrow is represented by silt with a bit of mix with some organics and almost no organic layer on top. Conductivity of the upper layer is ~1 W mK$^{-1}$ for unfrozen and ≥ 2 W mK$^{-1}$ for frozen soil**. Gakona is a continuous permafrost forest tundra site with measurements at 36 depths from 0 - 30 m. **Gakona has a thick organic layer of moss (0 to 5 cm), dead moss (from 5 to 13 cm), and peat (from 13 to 50 cm), then silty clay at depth**.*"

- L263: "*Our simulations show widespread talik formation throughout Siberia and northern North America over the period 2010-2300 (Fig. 1B), impacting ~14.5 million km$^2$ of land in*

*NHLs (55°-80°N) assuming a freeze/thaw threshold of -0.5°C **and RCP8.5 and ECP8.5 warming scenarios**."*

- L307: "***Our simulations show a similar, but very slight,** drying pattern in shallow layers in the 4 decades prior to talik onset **(1.3% loss of soil moisture over 0-1 m depth;** Fig. 2D), **accounting for about half of** total water storage **loss in** the column. **More** significant changes in water balance **occur** following talik onset, including **more** rapid **drying in shallow layers (~10% over 4 decades) and in the column (~16%), and a substantial** increase in sub-surface drainage, as discussed below*."

- L323: "*In the 3 decades leading up to talik onset, we find **gradual deepening of the thawed layer** to 3-4 m and penetration of thaw period into Jan-Feb*."

- L332: "*We note that bedrock soil is not hydrologically active in CLM4.5, and thus the rate of thawing and drainage in response to permafrost thaw may be **overestimated** in deeper CLM4.5 layers near bedrock due to reduced heat capacity*."

- L355: "*Talik onset in CLM4.5 is variable in **the region containing Gakona (southeast Alaska)** with earliest onset by mid-century (~2050s, Fig. 1A); however, our comparison to **borehole temperature data at Gakona** suggests that simulated thaw rates **in southwest Alaska and across pan-Arctic regions with** similar permafrost temperatures are underestimated, and that earliest onset may occur sooner than predicted*."

- L359: "*Overall, we find that simulated patterns of permafrost thermal state change are consistent with available observations, but that the exact thaw rates are uncertain. **Although there are many possible explanations for differences in observed and simulated thaw rates, we can attribute high observed thaw rates in part to a combination of (1) relatively dry upper soil at Gakona and Mould Bay, and (2) low surface organic layer and high conductivity of the Barrow and Mould Bay soils.** We keep these uncertainties in mind as we examine patterns of change and talik formation simulated into 2300*."

- L383: "*We focus first on site-specific long-term **historical** trends by analyzing the **6** Siberian borehole sites which recorded at least **5** years of **temperature** data spanning multiple decades: Drughina, Lensk, Macha, **Oimyakon**, Uchur, and Chaingda*."

- L385: "*Records at these locations show an **increase** in thaw volume with an average positive trend of 0.19 m months yr$^{-1}$ from 1955 – 1990 (Table 1, Fig 5). All sites except Drughina show **positive** trends*"

- L390: "*Further examination indicates that active layer thickness at Drughina actually decreased to 0.8 meters from 1989-1990 compared to 1.2 meters in the 1970s **(data not shown)**. Drughina also shows smaller average thaw volume magnitude compared to other sites, consistent with shallower thaw. Together, these findings indicate that active layer thickness is decreasing at Drughina*."

- **L395: "*There is considerable spatial variability in thaw volume and trends, but in general thaw trends increase from west (0.18 m months yr-1) to east (0.51 m months yr-1). Talik forms at several sites, at different times between 1957 and 1990 (shown by vertical dashed lines on Fig. 5), with earlier talik to the west consistent with higher mean initial thaw volumes.*"**

- **L405: "*We recompute observed thaw trends at regional clusters using combined records at the 2 sites in northern Siberia (blue), 6 sites in southwest Siberia (orange), and 1 site in southeast Siberia (brown, Table 1) and compare to historical and projected thaw volume trends in CLM4.5 (Fig. 6). Northern locations show a consistent pattern of low thaw*"**

*volume ($< 10$ m month yr$^{-1}$) and negligible thaw trend ($\sim 0$ m month yr$^{-1}$) in the historical simulations and observed record from 1950-2000. Thaw projections in northern Siberia indicate continued stability of permafrost through the early 22$^{st}$ century, followed by a shift to accelerated soil thaw in the early 2120, marked by onset of deep soil thaw late in the cold season.*

- L410: *"Thaw projections in northern Siberia indicate an **unchanged** trend and continued stability of permafrost through the early 22$^{st}$ century, followed by a shift to accelerated soil thaw in the early 2120, marked by onset of deep soil thaw late in the cold season."*

- *L413: Southern locations show a systematic underestimate of mean thaw volume ($< 20$ m month yr$^{-1}$) compared to observations ($\sim 40$ m month yr$^{-1}$) from 1950-2000. Simulated thaw trends are negligible prior to 2000, but these likely represent an underestimate given low simulated thaw volumes and significant positive observed trends in both southeast and southwest Siberia beginning in the 1960s following talik onset (Fig. 5). Thaw projections show more abrupt shifts in thaw volume in the early 21$^{st}$ century in the southwest ($\sim 2025$) and in the mid 21$^{st}$ century ($\sim 2050$) in the southeast. The strong discrepancy between observed and simulated thaw and talik onset in southern Siberia warrants close monitoring and continued investigation of this region through sustained borehole measurements and additional model realizations of potential future warming.***

- L424: "Fig. 7A plots the decade in which NHL ecosystems are projected to transition to long-term C *sources over the next 3 centuries (2010-2300). A total of 6.8 million km$^2$ of land is projected to transition, peaking in the late 21$^{st}$ century, with most regions transitioning prior to 2150 (4.8 million km$^2$ or 70%, Fig. 7B, solid black).* **C source transitions which occur in the permafrost zone, accounting for 6.2 million km$^2$ of land (91% of all C source transitions), also form talik at some time from 2006-2300 (Fig. 7C). The remaining C source transitions (0.6 million km$^2$, or 9%) occur outside the permafrost zone, primarily in eastern Europe.**"

- *L431: "Net C emissions from C source transition regions are a substantial fraction of the total NHL C budget over the next 3 centuries (Fig. 8). The cumulative pan-Arctic C source increases slowly over the 21$^{st}$ century, reaching 10 Pg C by 2100 with RCP8.5 warming, then increases more rapidly to 70 Pg C by 2200 and 120 Pg by 2300 with sustained ECP8.5 warming (Fig. 8, solid black). This pan-Arctic source represents 86% of cumulative emissions in 2300 from the larger NHL talik region (crosses), despite the 2 fold smaller land area, and exceeds the talik region through 2200 due to mitigating widespread vegetation C gains (Koven et al., 2015). Cumulative emissions over all NHL land regions (diamonds, $> 55N$) increase in similar fashion to the talik region, reaching 120 Pg C by 2200 and 220 by 2300, with no sign of slowing."*

- *L440: "The geographic pattern of C sink-to-source transition date is reversed compared to that of talik formation, with earlier transitions at higher latitudes (the processes driving these patterns are discussed in detail below). Overall, the lag relationship between talik onset and C source transition exhibits a tri-modal distribution (Fig. 7D), with peaks at negative time lag (C source leads talik onset, Median Lag = -5 to -6 decades), neutral time lag (C source synchronized with talik onset; Median Lag = -2 to 1 decade), and positive time lag (C source lags talik; Median Lag = 12 decades; red shading in Fig. 7C). Roughly half of these regions (3.2 million km$^2$) show neutral or positive time lag (lag $\geq 0$). This pattern, characteristic of the sub-Arctic ($< 65°N$), represents the vast majority of C source transitions after 2150 (Fig.*

7B, dotted), **but only accounts for 17% of cumulative emissions (20 Pg C by 2300, Fig. 8, dotted)**. *The remaining regions (3.0 million km$^2$) in the Arctic and high Arctic (> 65°N) show negative time lag and account for most of late 21$^{st}$ century sources (Fig. 7B, dashed)* **and cumulative emissions (95 Pg C by 2300, or 79%; Fig. 8, dashed)**. *C sources in regions not identified as talik (0.63 million km$^2$) either show talik presence at the start of our simulation, or are projected to transition in the absence of permafrost or in regions of severely degraded permafrost (Fig. 7C, dash dotted).* **This region contributes only 5 Pg C (4%) of cumulative C emissions in 2300."**

- L458: "**In these regions**, *thaw volume is low (< 50 m months yr$^{-1}$) and shows a weak relationship to NBP (NBP* **decreases** *much faster than thaw volume) prior to C source onset (indicated by large green circle in Fig. 8A)."*

- L470: "*A broader analysis of soil thaw statistics over all regions and periods indicates* **that most C source transitions (~2.3 million km$^2$, or 77% of land where C source leads talik) occur at active layer depths below 3 m and thaw season penetration into November**."

- L479: "*The total area of land in which SOM exceeds 100 kg C m$^{-2}$ represents 2/3 of all land where C sources lead talik onset (2.0 million km$^2$), and peaks at a negative time lag of -5 to -6 decades (Fig. 7D, green bars), which perfectly aligns with the peak distribution of negative time lags.* **Cumulative C emissions from regions of SOM > 100 kg C m$^{-2}$ are also 2/3 of total C emissions (80 Pg C; Fig 8, green)."**

- L489: "*and talik formation occurs when these regions are weak sinks (NBP* **>** *0 g C m$^{-2}$ yr$^{-1}$)."*

- L490: "*In general,* **C source onset under high thaw volume indicates** *these regions are more sensitive to C emissions from deep soil thaw."*

- L495: "*The regions where fire C emissions exceed 25 g C m$^{-2}$ yr$^{-1}$, representing our threshold for C source transition, are exclusively boreal ecosystems, account for 1/3 of all land with negative lags (~1.1 million km$^2$), and align perfectly with the peak distribution of positive time lags (Fig. 7D, red bars)* **and cumulative C emissions (20 Pg C in 2300, Fig. 8, red)."**

- L507: **"The decadal time lag between talik onset and C source transition is more normally distributed in the remaining region, represented by the residual grey bars visible in Fig. 7D, which occurs predominantly in cold northern permafrost in northwest Siberia where low SOM (< 100 Kg m-2) and fire emission (< 25 g C m$^{-2}$ yr$^{-1}$) prevail. This region has a mean lag of 1 decade from talik onset to C source, with high standard deviation of lags (± 8 decades) reflecting a skewed distribution of GPP; low productivity in cold permafrost (GPP = 385 g C m$^{-2}$ yr$^{-1}$) increases the likelihood that soil thaw will lead to C source transition prior to talik onset, and high productivity in warm permafrost GPP = 1111 g C m$^{-2}$ yr$^{-1}$) increasing the likelihood of a transition after talik onset. Cumulative C emissions from this region are on the low end (27 Pg C by 2300; Fig. 8, blue) due to low soil C (SOM = 59 kg C m$^{-2}$). "**

- L527: "*GPP and combined respiration* **increase by ~15%** *per decade for each permafrost regime surrounding the decade of C source transition with peak fluxes in the growing season (Fig. 11 A - D)."*

- L541: **"The trend in the respiration difference in warm and cold permafrost, which increase by similar amounts (~100 g C m$^{-2}$ yr$^{-1}$), thus reflects an increasing dominance of respiration from younger and older soil C pools, respectively. These trends are identical to the corresponding NBP trends, which decrease by 100 g C m$^{-2}$ yr$^{-1}$ over the same period**

- *from neutral to net source (Fig. 12 G – H), such that the differences between GPP and respiration driving the NBP trends are explained almost entirely by the increasing fraction of soil vs litter respiration."*
- **L586:** *"Controlled experiments demonstrating the sensitivity of talik to parameters that control soil drying such ice impedance or baseflow scalars (e.g. Lawrence et al., 2015), and the effect of organic content and mineral soil texture (Lawrence and Slater, 2008), could provide key insight on soil thermal dynamics in frozen or partially frozen conditions. Other factors affecting soil hydrology and carbon cycling not considered in our CLM4.5 simulations include high spatial resolution in discontinuous permafrost, shifts in vegetation community, lateral flow representation, thermokarst activity and other thaw-related changes to the ground surface, surface slope and aspect, soil heterogeneity, and potentially several other factors (see Jorgenson and Osterkamp (2005) for discussion of some of the many complexities to be considered)."*

- L596: ***"Our simulations show increasing C emissions over time across the talik region (Fig. 1B), as cumulative NBP becomes increasingly negative (NBP < 0 equals a net C source), reaching a net source of 140 Pg C by 2300 (Fig. 8, crosses), consistent with previous estimates of net C balance across the larger pan-Arctic region from CLM4.5 (~160 Pg C, Koven et al., 2015; Lawrence et al., 2015). Ecosystems which transition from net C sinks to net C sources represent less than half the total talik area (6.8 of 14.5 Million km$^2$, Fig. 7A), but account for most (~85%) of the cumulative emissions, reaching 10 Pg C in 2100, 70 Pg C in 2200, and 120 Pg C by 2300 (Fig. 8, solid black). Removing the effect of vegetation C gain (~20 Pg C in 2100 and 40 Pg C in 2200 and 2300 according Koven et al., 2015), we estimate a cumulative permafrost emission for C source transition regions of 30 Pg C in 2100, 110 Pg C in 2200, and 160 Pg C in 2300. These numbers are on the low end but consistent with estimates of permafrost C emissions summarized by Schuur et al. (2015), which range from 37-174 Pg C by 2100 and 100-400 Pg C by 2300."***

- L608: "***About half of this region (3.2 million km$^2$)*** *shows a pattern of accelerated soil C respiration following talik onset, which shifts the surface C balance of photosynthetic uptake and litter respiration from net C sinks to long term net sources **totaling 20 Pg C** across 3.2 million km$^2$ of NHL land by 2300."*

- Line 626: ***"Our estimate of C emissions following talik onset (~20 Pg C) is low compared to the cumulative emissions from all long term C source transitions (120 Pg C), but likely strongly underestimated. Soil C is not permitted below 3.5 m in CLM4.5, or in most analogous models, such that potential decomposition of the ~350 Pg soil organic C in deep permafrost (> 3 m) is not accounted for (Hugelius et al., 2014; Jackson et al., 2017). This is significant for our simulations, which show frequent talik formation and accelerating thaw volumes below 3 m (e.g., Fig. 3). We therefore caution the reader in the interpretation of the timing and magnitude of permafrost C emissions following talik onset in our simulations, which represent a lower bound of potential emissions based on the current formulation of CLM4.5."***

- L635: *"We identify an equally large region of land in the high Arctic, representing ~3.0 million km$^2$, which is projected to transition to a long term C source much sooner than the sub-Arctic in the absence of talik, **and emit 5 times as much carbon by 2300 (~95 Pg C)**."*

- L688: "*6.8 million km² of land impacted in Siberia and North America will produce an integrated C source of **90 Pg C by 2100 and 120 Pg C by 2200**.*"
- L692: "*Our main results emphasize **an increasingly important impact of NHL cold season warming on** earlier spring thaw, longer non-frozen seasons, and increased depth and seasonal duration of soil thaw.*"

*Figure Changes:*

- We also included information in the first talik figure (Fig. 1) and carbon figure (Fig. 7): "***These results assume a Representative Pathway 8.5 warming scenario through 2100 and an Extended Concentration Pathway 8.5 through 2300.***"

- The definition of NBP has been corrected in the text and in the figures.

- Revised Figure 8 (now Figure 9) with arrows indicating C source or sink for clarification:

- Corrected y-axis error on Fig. 7 by moving the C source numbers from Fig. 7B to Fig. 8 for more in depth analysis (Figure 7 below; Figure 8 provided in Reviewer 1 responses). Added Figure 8

- Removed Figures 10 and 11

- Color specifications have been corrected in all figures.

- New Fig. 11: We have included a description of error bars in figure 12, which represent the multi-decadal standard deviation of seasonal and C source transition time series from 2040 – 2270.

- Removed the thaw onset symbols from Figure 6
- We also include an additional column in Table 2 for soil characteristics:

**Tracked Changes**

[revised manuscript text omitted]